# Multitrait analyses identify genetic variants associated with aortic valve function and aortic stenosis risk

Shinwan Kany[1,2,3,4], Joel T. Rämö [1,5,6], Cody Hou[1,7], Sean J. Jurgens [1,8], Shaan Khurshid[1,4,9,10], Victor Nauffal [1,11], Jonathan W. Cunningham[1,11], Emily S. Lau[1,9], Satoshi Koyama [1], FinnGen*, Jennifer E. Ho [1,12], Jeffrey E. Olgin[13], Sammy Elmariah[13], Aarno Palotie[6], Mark E. Lindsay [1,4,9,14,15], Patrick T. Ellinor [1,4,9,10,14] & James P. Pirruccello [1,13,16] ✉

The genetic influences on normal aortic valve function and their impact on aortic stenosis risk are of substantial interest. We used deep learning to measure peak velocity, mean gradient and aortic valve area from magnetic resonance imaging and conducted genome-wide association studies (GWAS) in 59,571 participants in the UK Biobank. Incorporating the aortic valve measurement GWAS with aortic stenosis GWAS using multitrait analysis of GWAS (MTAG), we identified 166 distinct loci (134 with aortic valve traits, 134 with aortic stenosis and 166 unique loci across all GWAS), including *PCSK9* and *LDLR*. The MTAG aortic stenosis PGS was associated with aortic stenosis in All of Us (hazard ratio (HR) = 3.32 for top 5% versus all others, $P = 8.8 \times 10^{-22}$) and Mass General Brigham Biobank (HR = 2.76, $P = 7.8 \times 10^{-15}$). Using Mendelian randomization, we found evidence supporting a potential causal role for Lp(a) and LDL on aortic valve function. These findings have implications for the early pathogenesis of aortic stenosis and suggest modifiable pathways as targets for preventive therapy.

Aortic stenos, the pathological narrowing of the aortic valve orifice, is associated with a high burden of morbidity and mortality, and affected over 12 million people in 2017 (ref. 1). In addition to the lack of preventive therapy, no established medical treatment apart from surgery or percutaneous valve replacement is available for aortic stenosis, and such interventions are typically reserved for severe disease[2,3]. Recent efforts have identified common genetic variants associated with the clinical diagnosis of aortic stenosis[4,5], with the largest effort to date yielding 32 genomic loci[6,7].

Analysis of quantitative endophenotypes for disease such as aortic diameter for aortic aneurysm[8,9] or left ventricular ejection fraction for heart failure[10,11] is a powerful approach for genetic discovery in healthy study populations. For example, a 2023 genome-wide association study

(GWAS) of thoracic aortic aneurysm and dissections, involving 8,626 cases and 453,043 controls, reported 21 genetic loci associated with the phenotype[12]. However, a 2022 GWAS of ascending aortic diameter in only 38,694 participants in the UK Biobank found 82 loci, including 19 of the 21 loci later identified in the 2023 disease-based GWAS[8], emphasizing the sample efficiency of quantitative endophenotypes for genetic discovery.

Phase-contrast cardiovascular magnetic resonance imaging (cMRI) provides velocity information to reveal blood flow patterns in the aorta, which allows for inference about aortic valve function and disease[13]. We recently developed a deep-learning-based model to estimate these clinically relevant aortic valve measurements from velocity-encoded cMRI data in the UK Biobank[14]. In this study, we aimed

**Fig. 1 | Study overview. a**, Deep learning was used to create segmentation masks of the ascending aorta in the aortic flow image series in the UK Biobank. **b**, The masks were used to create velocity maps that were used to construct three cMRI-derived phenotypes (AVA, mean gradient and peak velocity). The orange color denotes female participants and the blue color denotes male participants. **c**, The traits were then used to conduct GWAS of common variants. The summary statistics were further used to create MTAG of AVA, mean gradient and peak velocity with a disease-based aortic stenosis GWAS. **d**, Gene-set enrichment and analyses for tissue enrichment, as well as cell-specific analysis for the thoracic aorta and left ventricle, were performed. **e**, Further analyses were performed using Mendelian randomization and polygenic scores of lipid-based traits and other common risk factors with the cMRI-derived traits. Additionally, polygenic risk scores were constructed for each phenotype. Polygenic scores from GWAS before MTAG were used to predict incident aortic stenosis in the external FinnGen cohort. **f**, MTAG-adjusted scores were applied to the external All of Us cohort and the MGB Biobank. cMRI images are reproduced by kind permission of UK Biobank. Medical images were used from Servier Medical Art under Creative Commons-BY 4.0 license.

to use cMRI-derived aortic velocity-based traits to elucidate the common genetic variation underlying aortic valve function and its potential associations with aortic stenosis.

## Results

### Deep learning to measure aortic valve endophenotypes

We studied the following three measurements relevant to aortic valve function in systole: aortic valve area (AVA; cm$^2$), peak velocity (m s$^{-1}$) and mean gradient (mm Hg; Fig. 1). These phenotypes were derived from velocity-encoded cMRI near the aortic valve in the UK Biobank using the deep-learning model that we have described in detail previously[14]. In short, we developed a U-Net-based deep-learning model constructed in PyTorch (v2.1.0)[15] using a 'ConvNext-small' encoder that had been pretrained with natural images from ImageNet[16,17] to localize the aorta and permit extraction of flow measurements from velocity-encoded images in UK Biobank (Supplementary Methods). Beginning with the 62,902 participants with deep-learning-based flow measurements that passed model quality control (QC), a total of 3,331 participants were excluded for pre-existing cardiovascular disease or genetic QC, leaving 59,571 participants contributing to the GWAS of peak velocity and mean gradient and 59,569 contributing to the GWAS of AVA (Table 1 and Supplementary Fig. 1). Disease definitions are provided in Supplementary Table 1.

### Common variants related to aortic valve function

To understand the genetic basis of aortic valve function from common variation, we conducted GWAS of 59,571 participants using REGENIE v2.2.4 (Supplementary Tables 2–4). For all three aortic valve traits, we observed a total of 90 loci (43 for AVA, 27 for peak velocity and 20 for mean gradient), of which 61 unique loci were associated

**Table 1 | Participant characteristics of the UK Biobank cohort with cMRI**

| Characteristics | Women | Men | All |
|---|---|---|---|
| n | 31,159 | 28,412 | 59,571 |
| Age at time of cMRI | 65.3 (7.65) | 66.5 (7.85) | 65.9 (7.77) |
| Height (cm) | 163 (6.24) | 176 (6.65) | 169 (9.24) |
| Weight (kg) | 69.0 (13.2) | 83.5 (13.3) | 75.9 (15.1) |
| SBP (mmHg) | 139 (20.2) | 144 (17.7) | 141 (19.2) |
| DBP (mmHg) | 77.6 (10.1) | 80.8 (9.92) | 79.1 (10.2) |
| BMI (kg m$^{-2}$) | 26.1 (4.78) | 27.0 (3.91) | 26.5 (4.41) |
| AVA (cm$^2$) | 2.50 (0.403) | 3.16 (0.545) | 2.81 (0.578) |
| Peak velocity (m s$^{-1}$) | 1.16 (0.250) | 1.17 (0.289) | 1.16 (0.269) |
| Mean gradient (mmHg) | 2.66 (1.23) | 2.82 (1.57) | 2.73 (1.41) |
| Moderate aortic stenosis | 219 (1%) | 122 (0%) | 341 (1%) |
| Severe aortic stenosis | 25 (0%) | 21 (0%) | 46 (0%) |

SBP, systolic blood pressue; DBP, diastolic blood pressure; BMI, body mass index; AVA, aortic valve area.

with at least one trait at a genome-wide significance threshold of $P < 5 \times 10^{-8}$ (Supplementary Table 4 and Supplementary Fig. 2). Shared at genome-wide significance across all three traits were genetic loci near LPA, PDE3A, HMGA2, *CDK8*, KCNRG/DLEU1, *GOSR2*, *CTAGE1* and *MN1*. Loci near DLEU1, HMGA2 and *GOSR2* have previously been linked to the diameter of the aortic root[18]. A locus near *CTAGE1* has been previously

described for aortic stenosis as well as abdominal aortic aneurysm[19,20], whereas *LPA* is a well-known gene associated with aortic stenosis in previous GWAS[6,19]. *PDE3A* and *CDK8* have all been previously linked to cardiovascular disease or physiology[21–23], although not specifically to aortic valve disease, whereas the link between *MN1* and cardiovascular disease remains largely unexplored. We also observed two signals on the X chromosome, in loci near *NDP* (for the gradient-based measures) and near *TSPAN6* (for AVA), both of which have not been described for aortic stenosis or aortic valve disease before. In a sensitivity analysis (Supplementary Methods) excluding participants with aortic valve measurements consistent with moderate (341 participants) or severe (46 participants) aortic stenosis, the number of associated loci increased from 61 to 64. This included the loss of four loci near *OR5D13*, *ELN*, *TMEM170A* and *PSTPIP1*, and the gain of seven loci near ANK2, MAD2L1, MYBPC3, CSH2, PLAU, TBC1D12 and *PIGU)*, all of which were within one order of magnitude of the $P < 5 \times 10^{-8}$ significance threshold in the main analysis (Supplementary Table 5).

In total, 6 of 23 loci in the recent GWAS discussed in refs. 6,19 previously associated with aortic stenosis (*LPA*, *ACTR2*, *CTAGE1*, *TEX41*, *TMEM170A* and *APLP/WNT4*) were re-identified here in association with peak velocity, mean gradient or AVA, while 1 locus for bicuspid aortic valve specifically (*GATA4*) was re-identified. A study in ref. 7 published a meta-analysis of calcific aortic stenosis, including bicuspid valve cases, and reported 32 loci at genome-wide significance, of which 9 were replicated in at least one of the deep-learning-derived continuous traits (ACTR2, ZEB2, LPA, FERD3L, SURF6/MYMK, PDE3A, HMGA2, TMEM170A and SLC44A2/LDLR). The full GWAS findings are represented in detail in the Supplementary Results and Supplementary Table 4.

### Common variants related to aortic stenosis

We conducted a meta-analysis of disease-based GWAS of aortic stenosis using METAL, including data from ref. 19 (13,765 cases and 640,102 controls), from ref. 6 (14,451 cases and 398,544 controls), from all UK Biobank participants who did not undergo cMRI (5,038 cases and 412,301 controls) and from the freeze 12 of FinnGen (12,398 cases and 487,930 controls). The details in ref. 19 and UK Biobank analyses had sample overlap, which was accounted for in our analyses. The baseline characteristics of all biobank participants included are provided in Supplementary Tables 6 and 7. For this disease-based GWAS meta-analysis, we observed 92 loci at genome-wide significance, including 20 of 23 previously described loci for calcific aortic stenosis (Supplementary Tables 4 and 8). We also confirmed 29 of 32 loci from the meta-analysis discussed in ref. 7, which was not included in our meta-analysis before multitrait analysis of GWAS (MTAG) for genetic discovery (but has a partial sample overlap; Supplementary Table 9).

Additionally, we observed newly associated loci near *LDLR* ($\beta = 0.031$, $P = 1.2 \times 10^{-13}$) and a missense variant in *PCSK9* ($\beta = 0.089$, $P = 2.4 \times 10^{-11}$), both of which are loci harboring genes that regulate blood levels of apolipoprotein B-rich lipids through the low-density lipoprotein (LDL) receptor. Furthermore, in addition to the previously implicated genes in lipid metabolism and aortic stenosis (FADS2/MYRF ($\beta = 0.047$, $P = 1.2 \times 10^{-32}$) and SORT1/PSRC1 ($\beta = -0.033$, P = $1.6 \times 10^{-20}$)), we identified variants near *SORL1* ($\beta = -0.024$, $P = 4.8 \times 10^{-8}$), which encodes the sortilin-related receptor[24]. We observed several other loci associated with lipid metabolism, such as *ALPL*, *SCARB1*, *LPL* and *ANGPTL4*. Lipoprotein lipase is encoded by *LPL* and involved in the lipolysis of triglycerides in lipoproteins[25]. One of the key regulators of LPL is angiopoietin-like protein 4 (ANGPTL4) that inhibits lipoprotein lipase similarly to ANGPTL3 (refs. 26,27).

Among the previously replicated loci for aortic stenosis is interleukin-6 (*IL6*; $\beta = 0.051$, $P = 9.2 \times 10^{-45}$). We also observed a variant near IL6 receptor (*IL6R*; $\beta = 0.025$, $P = 7.6 \times 10^{-11}$), encoding the IL6R.

We additionally observed loci with key regulators of phosphate homeostasis, such as previously reported *ALPL* ($\beta = 0.031$, $P = 4.2 \times 10^{-18}$)

and, here for the first time in connection to aortic stenosis, *FGF23* ($\beta = -0.028$, $P = 3.2 \times 10^{-13}$)[28].

We observed similar results in UK Biobank and FinnGen when (1) limiting our definition of aortic stenosis cases to those who also had a procedure code for aortic valve intervention or (2) excluding those with a diagnosis of coronary artery disease before aortic stenosis to account for a possible ascertainment bias (Supplementary Table 10). A polygenic score (PGS) based on GWAS of aortic valve traits was predictive of aortic stenosis in FinnGen. Being in the top 5% for genetically predicted mean gradient led to a hazard ratio (HR) of 1.44 for aortic stenosis (95% confidence interval (CI) = 1.35–1.54, $P = 1.1 \times 10^{-27}$) compared to the bottom 95% (Supplementary Table 11).

### MTAG

Having established that quantitative aortic valve measurements were heritable biomarkers for aortic valve stenosis, we pursued MTAG to incorporate information from these related traits[29]. This analysis included the three aortic valve traits and the aortic stenosis meta-analysis. This yielded an effective sample size of $n = 96,385$ for peak velocity, $n = 98,645$ for mean gradient, $n = 77,183$ for AVA and $n = 205,483$ for aortic stenosis. The maxFDR, the upper bound for the false discovery rate (FDR), was 0.015 for peak velocity, 0.014 for mean gradient, 0.012 for AVA and 0.003 for aortic stenosis.

Using this approach, the number of loci associated with at least one of the three aortic valve traits at genome-wide significance increased from 90 to 273 loci overall (90 for AVA, 88 for mean gradient and 95 for peak velocity) and from 61 unique loci to 134 unique loci between all three traits (Supplementary Table 12 and Fig. 2), while the number of aortic stenosis loci increased from 92 in the METAL meta-analysis to 134 post-MTAG (Fig. 3). When considered altogether, 166 distinct loci were identified across the aortic valve endophenotypes and aortic stenosis GWASes within the MTAG framework (Supplementary Table 12). Of these, 32 loci were associated only with aortic stenosis but did not reach genome-wide significance for any of the aortic valve endophenotypes, including *PCSK9* and *IL6R*. However, these loci had substantial subthreshold signals, as seen in Supplementary Table 8. A sensitivity analysis leaving out the mean gradient (which is highly correlated with peak velocity) showed similar yield in loci (Supplementary Results).

Of the 166 distinct MTAG loci, 102 were associated with aortic stenosis and at least one valve measurement with $P < 5 \times 10^{-8}$ for both (Supplementary Fig. 3, raw non-MTAG P values and effect estimates shown for all MTAG loci in Fig. 4).

### Genetic discovery of 134 aortic stenosis loci after MTAG

Among the 134 independent loci for aortic stenosis (Supplementary Table 12), we re-identified all of the 23 previously reported loci for calcific aortic stenosis, with the sole exception of a locus near *HMGB1* from ref. 19 and 29 of the 32 loci reported from ref. 7 (Supplementary Table 9). Among the 53 loci that were only observed for aortic stenosis after MTAG, we uncovered an additional locus near *APOE* ($-0.024$, $P = 4.8 \times 10^{-9}$) related to lipid metabolism. Additionally, other genes related to inflammation pathways, such as *TRAF1* ($\beta = 0.048$, $P = 3.3 \times 10^{-9}$)[30], *TRIM59* ($\beta = 0.021$, $P = 3.7 \times 10^{-8}$)[31] and *UBE2L3* ($\beta = 0.025$, $P = 2.0 \times 10^{-8}$)[32] emerged after MTAG. Another group of loci was previously implicated in GWAS of aortic root diameter from cMRI or echocardiography images, such as *KCNRG/DLEU1*, *CACNA1C*, *FGGY*[18,33] and *SMAD3*, which is known for its role in aortic syndromes[34]. We also discovered variants near genes more closely related to valve pathology. For instance, both *ERG* ($\beta = -0.107$, $P = 4.3 \times 10^{-9}$)[35] and *VGLL4* ($\beta = -0.019$, $P = 2.0 \times 10^{-8}$)[36] were previously associated with valve morphogenesis. We also observed loci near *KLF2*, *PRDM16* and *PKN2* (refs. 37–39), which have been linked to flow-mediated responses in vascular smooth muscle cells. Of the 134 loci identified in the MTAG-augmented analysis of aortic stenosis, 46 loci were also identified among the 241 loci associated with coronary artery disease discussed in ref. 40 (Supplemental Results).

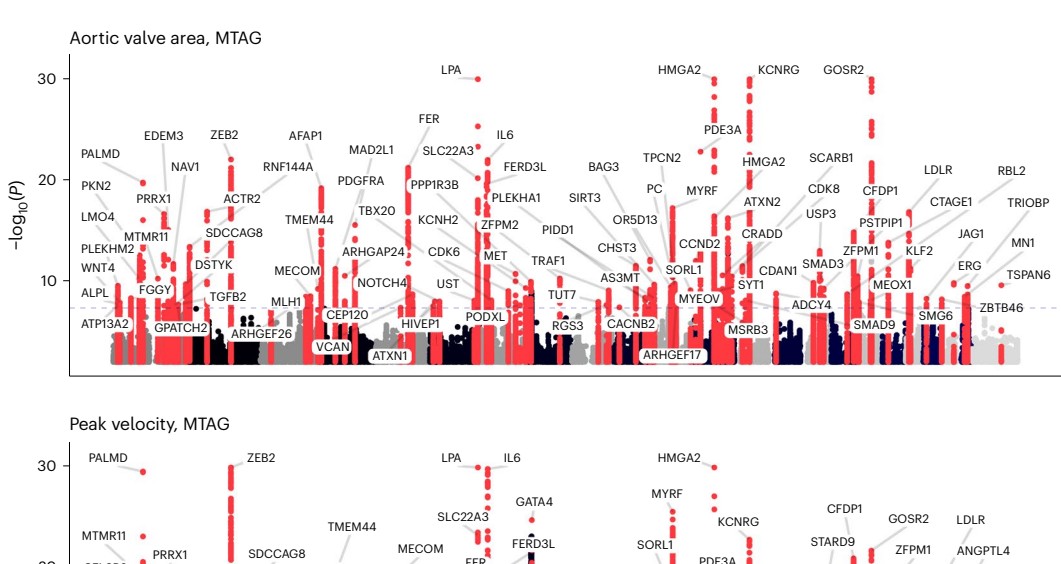

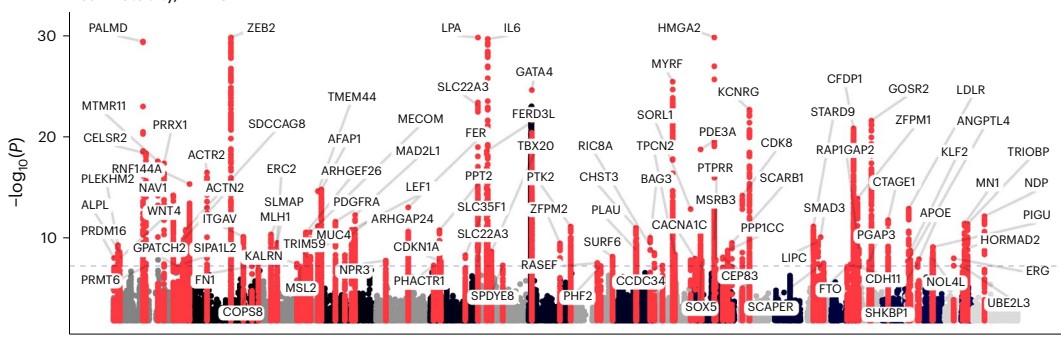

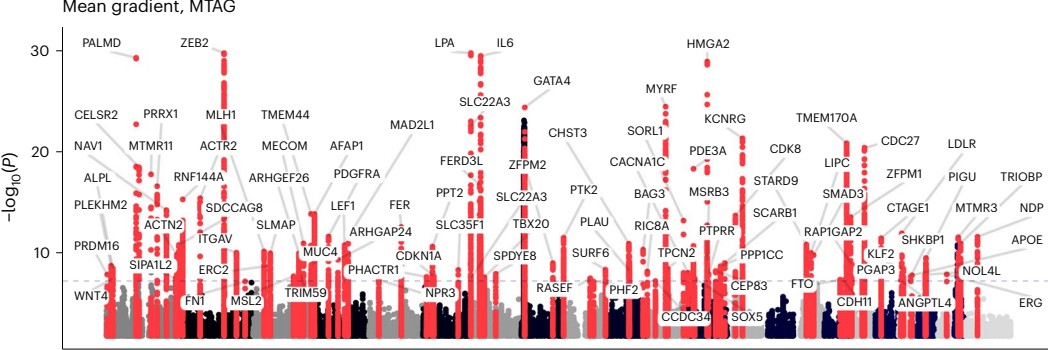

**Fig. 2 | Manhattan plot of GWAS results after MTAG.** Genetic associations for AVA, peak velocity and mean gradient after MTAG. Loci with $P < 5 \times 10^{-8}$ are colored red and labeled with the name of the gene in closest proximity to the strongest associated variant. MTAG uses a generalization of IVW meta-analysis and provides updated effect sizes, effective sample sizes and $P$ values for the summary statistics for each trait. For visualization purposes, the Manhattan plot is truncated at a $-\log_{10}(P) = 30$. $X$ axis, chromosomal position; $y$ axis, $-\log_{10}(P)$.

## Polygenic scores predict aortic stenosis risk

We used the MTAG-augmented GWAS to produce 1.1-million variant polygenic scores that were tested for association with aortic stenosis diagnosed after enrollment in the All of Us biobank (Methods) using Cox regression. All four polygenic scores were significantly associated with incident aortic stenosis among 496 incident cases and 243,954 controls. The strongest in terms of effect size was the aortic stenosis score (HR = 1.64 per s.d., 95% CI = 1.50–1.78, $P = 8.7 \times 10^{-30}$), whereas the weakest was the mean gradient score (HR = 1.53 per s.d., 95% CI = 1.40–1.66, $P = 1.1 \times 10^{-22}$; Supplementary Table 13).

When participants were stratified into the top 5% for each score versus the remaining 95%, the mean gradient PGS was strongly associated with aortic stenosis (HR = 2.61, 95% CI = 2.00–3.40, $P = 1.1 \times 10^{-11}$). Using the aortic stenosis MTAG PGS, participants in the top 5% had a more than threefold increase in risk for incident aortic stenosis (HR = 3.32, 95% CI = 2.60–4.4.24, $P = 8.8 \times 10^{-22}$; Fig. 5).

We further applied the MTAG-derived polygenic scores in the healthcare-based Mass General Brigham (MGB) Biobank with 680 cases and 42,328 controls in Cox proportional hazard models. All four

polygenic scores were significantly and directionally consistent in their association with aortic stenosis (HR = 1.56–1.61 per s.d. of PGS, $P = 3.5 \times 10^{-17}$ to $6.3 \times 10^{-36}$; Supplementary Table 14). In the stratified analysis comparing the top 5% of participants (bottom 5% for AVA) with the remaining participants, the aortic stenosis-based score was significantly associated with aortic stenosis (HR = 2.76, $P = 7.8 \times 10^{-15}$). The PGS for mean gradient (HR = 2.79, $P = 4.3 \times 10^{-15}$) and the score for peak velocity (HR = 3.02, $P = 6.3 \times 10^{-19}$) had effect estimates similar to those of the aortic stenosis-based score.

## MAGMA, tissue and cell-type enrichment analyses

Statistical gene-set prioritization was performed for each phenotype by applying the MAGMA framework to the MTAG-augmented GWAS summary statistics[41,42]. Top pathways achieving FDR $P < 0.05$–and associated with both aortic valve measurements and aortic stenosis–included those for coronary artery disease, lipid particle composition and suppressor of mothers against decapentaplegic (SMAD) protein complexes (Supplementary Figs. 4 and 5 and Supplementary Table 15).

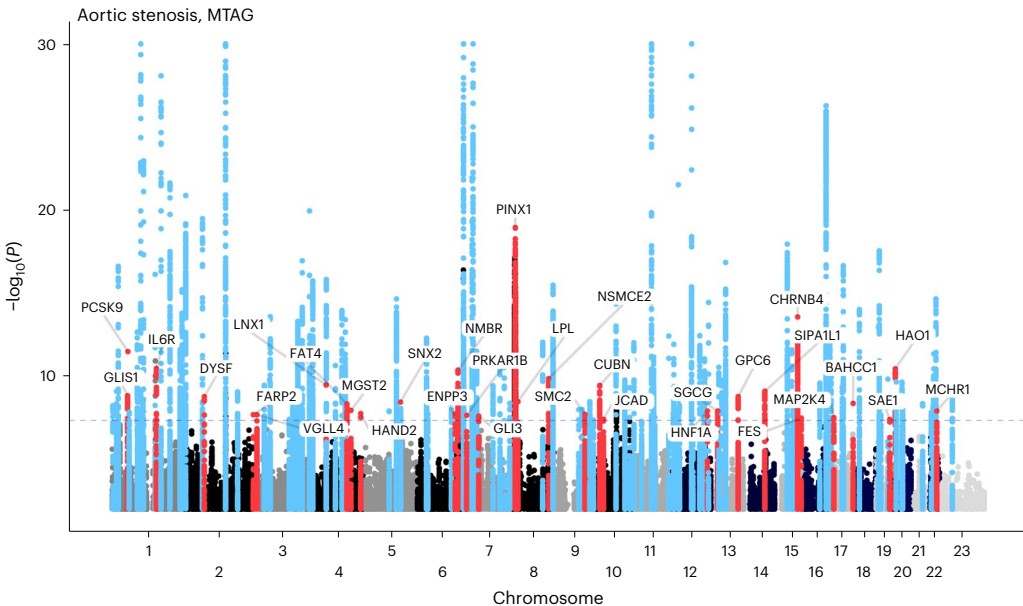

**Fig. 3 | GWAS for aortic stenosis after MTAG.** Genetic associations for aortic stenosis after MTAG. The color is blue when the locus was associated at $P < 5 \times 10^{-8}$. The color is red when the locus was associated at $P < 5 \times 10^{-8}$ with aortic stenosis but not aortic valve measurements after MTAG; the loci are annotated with the name of the gene in closest proximity to the strongest associated variant. MTAG uses a generalization of IVW meta-analysis and provides updated effect sizes, effective sample sizes and $P$ values for the summary statistics. For visualization purposes, the Manhattan plot is truncated at a $-\log_{10}(P) = 30$. $X$ axis, chromosomal position; $y$ axis, $-\log_{10}(P)$.

Examining gene expression within human tissues using MAGMA and GTEx (v8)[41,43], we found a significant enrichment of expression of genes identified in the GWAS within three arterial tissues (aorta, tibial artery and coronary artery), as well as esophageal tissues and nominal enrichment within left ventricular tissue for aortic flow traits but not for aortic stenosis (Supplementary Fig. 6). We then assessed cell-type specificity using single-nucleus RNA sequencing (scRNA-seq) data from the left ventricle, the aorta and the aortic valve.

In an aortic valve scRNA-seq dataset from a hyperlipidemic mouse model from ref. 44, the loci observed in our study were associated with valvular interstitial cells and valvular endothelial cells but not leukocytes (Supplementary Fig. 7). We also examined cell types in the neighboring tissues (the ascending aorta and the left ventricle). In ascending aortic scRNA-seq data, vascular smooth muscle cells, pericytes and fibroblast cell types overexpressed genes linked to peak velocity, mean gradient, AVA and aortic stenosis (Supplementary Fig. 8)[8]. Analysis of scRNA-seq from left ventricular cardiomyocytes revealed significant enrichment of expression of genes identified in all four GWAS within 'activated fibroblasts', 'fibroblast I' and 'fibroblast II', as well as other cell populations from ref. 45. The lowest expression and the only cell types not reaching nominal significance in gene expression from all four GWAS were 'cardiomyocyte III' and 'epicardial' cell populations (Supplementary Fig. 9).

### Causal evidence for lipoproteins in aortic valve dysfunction

Given the identification of loci involved in lipid biology, we pursued further evaluation with PGS and Mendelian randomization. In total, 1.1 million variant PGS for ApoA, ApoB, triglycerides and Lp(a) were produced in UK Biobank participants whose data did not contribute to the cMRI-based GWAS and tested against the aortic valve measurements. The scores explained between 10.7% (triglycerides) and 28.5% (Lp(a)) of the variance of their respective phenotypes within the cMRI participants (Supplementary Table 16). PGS of ApoB and Lp(a) were significantly associated with aortic valve traits and aortic stenosis, while ApoA PGS was not (Supplementary Table 17).

In the two-sample Mendelian randomization analysis, a similar pattern was observed. For example, ApoB was associated with decreased AVA ($\beta = -0.09$, $P = 2.2 \times 10^{-11}$) and increased peak velocity ($\beta = 0.0.8$, $P = 5.4 \times 10^{-9}$), mean gradient ($\beta = 0.08$, $P = 2.4 \times 10^{-9}$; Supplementary Table 18) and risk for AS (odds ratio (OR) = 1.12, $P = 5.5 \times 10^{-13}$; using the pre-MTAG meta-analysis; Supplementary Table 19) using the inverse variance weighted (IVW) method. All associations remained significant in a sensitivity analysis using MR Egger ($P = 7.8 \times 10^{-5}$ for AVA to $7.8 \times 10^{-7}$ for aortic stenosis). Lp(a) was also associated with smaller AVA ($\beta = -0.06$, $P = 1.1 \times 10^{-10}$; Supplementary Table 18) and greater aortic stenosis risk (OR = 1.10, $P = 1.5 \times 10^{-15}$; Supplementary Table 19). There was no association with ApoA using IVW ($P = 0.66-0.93$) or MR Egger ($P = 0.77-0.95$).

### Genetic evidence for phosphate in aortic valve dysfunction

Serum calcium and phosphate levels have long been associated with aortic stenosis in observational studies[46]. Given the association between variants at the *FGF23* locus and peak velocity, we performed Mendelian randomization for blood levels of phosphate, calcium and vitamin D as exposures against aortic valve measurements and aortic stenosis (using the pre-MTAG METAL meta-analysis)[47] (Supplementary Tables 18 and 19). A 202-variant genetic instrument for phosphate was nominally significantly associated with AVA ($\beta = -0.07$, $P = 1.4 \times 10^{-2}$) and was directionally concordant but not significantly associated with peak velocity ($\beta = 0.04$, $P = 6.9 \times 10^{-2}$) or mean gradient ($\beta = 0.04$, $P = 5.9 \times 10^{-2}$). We observed a genetically predicted increase in risk for aortic stenosis per s.d. of phosphate (OR = 1.22, $P = 1.5 \times 10^{-11}$). We did not observe robust genetic evidence for a causal role of calcium or vitamin D in the risk for aortic stenosis or influence on aortic valve function (Supplementary Tables 18 and 19).

### Genetic evidence for blood pressure in aortic stenosis

Observational studies have linked common cardiovascular risk factors with aortic stenosis risk[48]. We applied Mendelian randomization to seek causal evidence for these common risk factors on aortic valve dysfunction. The outcome GWAS for aortic stenosis was the METAL meta-analysis, excluding the UK Biobank, and no MTAG was used to avoid a substantial sample overlap with the exposure GWAS. We observed that 1 s.d. of increase in systolic blood pressure (SBP) was associated

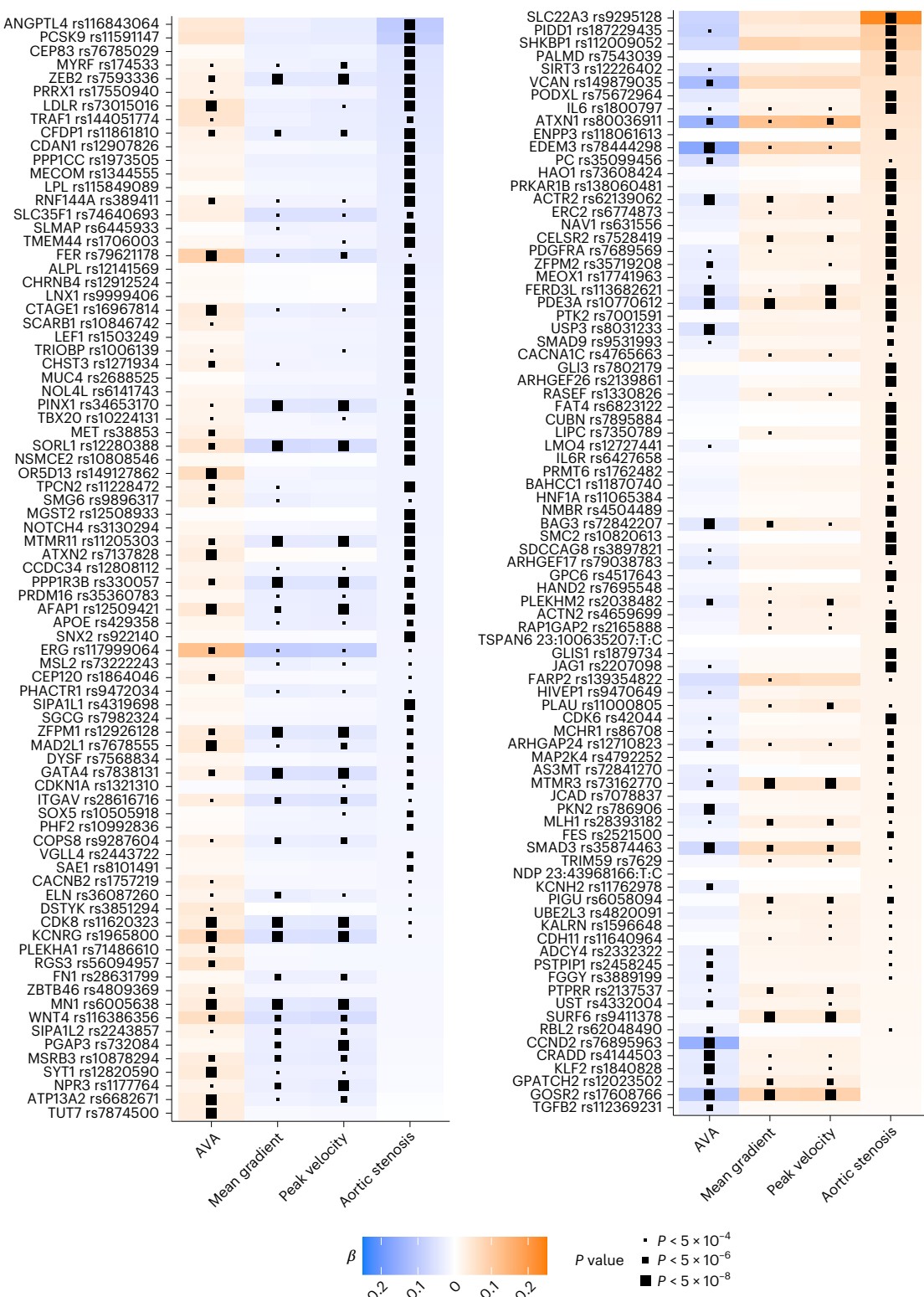

**Fig. 4 | Variant effect alignment at MTAG loci.** Depiction of the loci significant for at least one trait in MTAG analysis. Effect estimates and *P* values are taken from the original (non-MTAG) analysis. Effect direction is depicted with respect to the reference allele. Variants with a risk-decreasing reference allele for aortic stenosis are depicted in the left panel. Those with a risk-increasing reference allele are depicted in the right panel. Within each panel, variants are sorted by effect size for aortic stenosis. Effect sizes have units of s.d. for AVA, peak velocity, mean gradient and log odds for aortic stenosis. Variants are represented by the nearest gene name and the variant identifier. Multiple variants at one locus are listed when the lead variant differs across phenotypes.

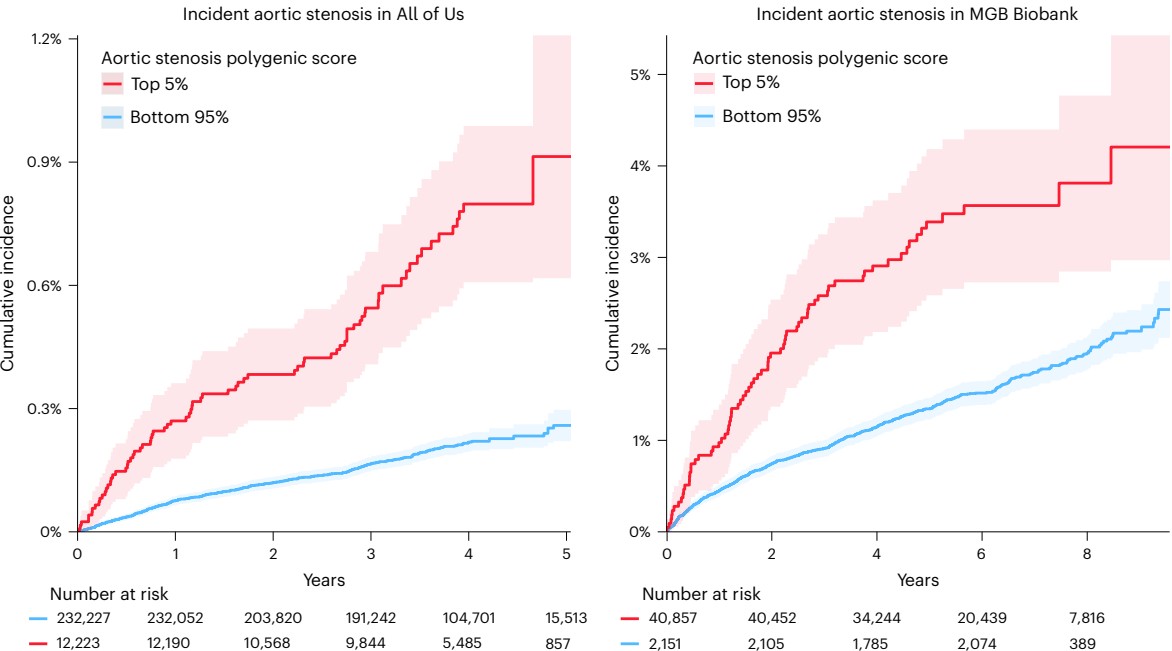

**Fig. 5 | Cumulative incidence of aortic stenosis in All of Us stratified by an MTAG polygenic score.** Kaplan–Meier cumulative incidence of aortic stenosis in 496 cases and 243,954 controls and 95% CI in All of Us (496 cases and 243,954 controls, left) and MGB Biobank (680 cases and 42,328 controls, right). Red indicates the top 5% for the aortic stenosis polygenic score. Blue indicates all other participants. The plot is terminated at the 95th percentile of follow-up time (5.0 and 9.0 years). For All of Us, the HR for the top 5% of participants was 3.32 ($P = 8.8 \times 10^{-22}$) compared to the remaining 95% of participants. For MGB Biobank, the HR for the top 5% of participants was 2.76 ($P = 7.8 \times 10^{-15}$) compared to the remaining 95% of participants.

with higher peak velocity ($\beta = 0.26$, $P = 1.1 \times 10^{-15}$) and greater risk for aortic stenosis (OR = 1.28, $P = 8.6 \times 10^{-12}$; Supplementary Tables 18 and 19). A similar observation was seen for body mass index (BMI) with a higher mean gradient ($\beta = 0.16$, $P = 3.4 \times 10^{-19}$) and risk for aortic stenosis (OR = 1.17, $P = 9.1 \times 10^{-18}$).

## Discussion

In this study, we analyzed aortic valve function and disease diagnoses to comprehensively evaluate the genetic basis of aortic stenosis. First, we conducted a large-scale genetic analysis of three deep-learning-derived measurements of aortic valve function from cMRI in 59,571 participants, identifying 61 distinct loci. We additionally conducted a meta-analysis of aortic stenosis in over 40,000 cases and 1.5 million controls, identifying 91 loci. We then conducted a multitrait analysis incorporating both continuous aortic valve measures and the disease-based GWAS, identifying 166 genetic loci (134 for aortic valve function and 134 for aortic stenosis).

These findings demonstrate the power of jointly analyzing endophenotypes and their downstream disease outcomes. Of the 23 loci previously identified in refs. 6,19, we replicated all but one locus (near *HMGB1*) while expanding the loci for aortic stenosis by 111 further loci, including loci near *PCSK9* and *LDLR*. We also replicate 29 of 32 loci in the recent meta-analysis of ref. 7, which also included bicuspid aortic valve.

Our findings suggest that risk for aortic stenosis is conferred at least in part through the same genetic mechanisms that drive normal variation in aortic valve function in the healthy population. Through sensitivity analyses excluding the 0.65% of participants with any measurement consistent with moderate-to-severe aortic stenosis, we found little change in the genetic signal from aortic valve measurements. We conclude that the genetic associations with aortic valve function were driven by normal variation in these healthy participants, rather than by those at the extremes. The genetic correlation between these measures in healthy people and the aortic stenosis GWAS meta-analysis was concordant with that interpretation—the rg with aortic stenosis was 0.64 for the gradient-based measures and 0.50 for AVA.

Remarkably, more than one-third of the genome-wide significant loci discovered in our analysis have also been associated with coronary artery disease at genome-wide significance. The most readily interpretable contributions are those from lipoprotein-related loci, including those near *ANGPTL4*, *APOE*, *LDLR*, *LIPC*, *LPA*, *LPL*, *PCSK9*, *SCARB1* and *SORT1*, an observation further supported by the association between polygenic scores for lipoproteins and variation in aortic valve measurements. Furthermore, with Mendelian randomization, we observed evidence consistent with a causal effect of higher levels of ApoB, triglycerides and Lp(a), but not ApoA-containing particles, on greater peak velocity and mean gradient across the aortic valve. Because genetically driven exposure to cholesterol is lifelong, these findings suggest that a comprehensive lipid-lowering strategy targeting all ApoB lipids (including Lp(a)) at very early stages could impact the pathogenesis or progression of calcific aortic stenosis. The overlap with coronary artery disease was also evident in the GWAS of continuous aortic valve traits, in which nearly 30% of the 61 loci had previously been reported in a GWAS of coronary artery disease[40]. Critically, the imaging traits were obtained for research purposes, so their genetic link to coronary artery disease is not influenced by imaging obtained due to coronary artery disease (that is, confounding by indication). In a sensitivity analysis of the GWAS, excluding individuals diagnosed with coronary artery disease before their aortic stenosis diagnosis, our observations remained largely consistent, suggesting that ascertainment bias is unlikely to explain our findings.

The genetic contributions to normal variation in peak velocity and mean gradient were predictive of aortic stenosis diagnoses in the following three external cohorts: FinnGen (for the PGS from the aortic valve function GWASes before MTAG, because FinnGen data were used in the MTAG analyses) and MGB and All of Us (for the MTAG-augmented PGS). Additionally, in All of Us, a fully external biobank not used in the GWAS or the PGS derivation, each s.d. increase in the MTAG-based aortic stenosis PGS conferred an HR of 1.64 for incident aortic stenosis. Compared to the remaining 95% of the population, participants in the top 5% of aortic stenosis PGS had a ~3.3-fold increased risk, warranting

further evaluation for clinical utility as part of a comprehensive screening algorithm in the future. The best-performing disease-based PGS for aortic stenosis from a recent study based on the Million Veteran Program showed an OR of 1.41–1.44 per s.d. for aortic stenosis[49], but differences in study populations and analytic design preclude a direct comparison of performance.

While several GWAS loci point to specific factors associated with, for example, lipid loci and liver biology (*LPL* and *ALPL*), the tissue-specific analysis suggested that the predominant information contained within the GWAS is salient to the structures in and around the aortic valve itself. Using scRNA-seq to further study both heart and arterial tissue, we observed that fibroblast cell types were prominently associated with aortic valve function and aortic stenosis. Endocardial and endothelial cells of the heart, but less so endothelial cells of the aorta, were also associated.

In addition to ApoB-containing lipoproteins, we also observed evidence for a causal effect of phosphate metabolism on aortic valvular function and aortic stenosis. We observed that for a lead variant near *FGF23*, rs10744645, the C allele was associated with a greater risk of aortic stenosis. Compared to the lipoprotein association pattern, which was concordant between risk for aortic stenosis and normal variation in aortic valve function, here we observed a distinct pattern—in the valve function GWAS, there was no evidence for an association between rs10744645 and AVA, peak velocity or mean gradient. This variant has previously been shown to be significantly associated with phosphate levels; the C allele was associated with 0.07 s.d. greater serum phosphate. It is in strong linkage disequilibrium ($R^2 = 0.87$) with missense variant rs7955866, which is thought to impede FGF23 protein degradation and thereby increase serum phosphate levels[28]. Formalized in a Mendelian randomization analysis, we again observed an increased risk for aortic stenosis, but no robust evidence supporting a causal effect of greater phosphate on higher aortic valve gradients. These results suggest that blood phosphate levels may have a causal role in disease progression rather than in the original pathogenesis of valve disease.

Because our GWAS before MTAG showed considerable power for genetic discovery, we aimed to leverage the summary statistics to explore potential causal contributions of common cardiovascular risk factors to aortic stenosis risk. Beyond the well-known link between dyslipidemia and aortic stenosis, observational studies have reported an association of hypertension and diabetes with a higher risk for aortic stenosis[48,50]. Here we show genetic evidence in support of a causal relationship between higher SBP and aortic stenosis risk, which also extends to normal variation below what is considered clinically significant aortic valve disease. We also observed that genetically predicted higher hemoglobin A1c was associated with smaller AVA, but the effect size for aortic stenosis was modest and not robust to sensitivity analysis. Our results suggest that optimal blood pressure control may be crucial in preventing aortic valve disease.

Several inflammation-linked loci were also identified. For example, the *IL6R* locus strengthens the link between inflammation and aortic stenosis beyond the previously identified *IL6* locus. We also found loci near *TRIM59* and *UBE2L3*, both of which are involved in inflammatory pathways protecting against inflammation until depletion of the proteins they encode[32,51]. We further report the *TRAF1* locus, which encodes for tumor necrosis factor receptor-associated factor 1 and has been shown to be pro-inflammatory in atherosclerosis[30]. Previous studies postulated that several disease processes contribute to aortic stenosis, including pro-fibrotic and pro-inflammatory processes[52]. The findings in the present analysis further support the role of inflammation in the pathogenesis of aortic stenosis.

Several other loci are of particular interest due to the biology that they highlight. For example, the aortic valve is unique in the high shear stress to which it is exposed. Supporting this notion, one locus associated with peak velocity harbors *KLF2*, which encodes a transcription factor that has been shown to sense shear stress and protect against vascular calcification[38]. Another locus was *PKN2* (for AVA), which encodes protein kinase N2, activated by flow through the mechanoreceptor Piezo-1 to regulate vascular tone[37]. Evidence of shear stress contributing to inflammation is provided by a study that showed that transcatheter aortic valve replacement reduces monocyte activation in peripheral blood via the Piezo-1 receptor[53].

## Limitations

Our study should be viewed in the context of its several limitations. It is based on the UK Biobank, whose participants tend to be healthier and not entirely representative of the general population in the United Kingdom[54]. Moreover, because of immortal time bias, the cMRI cohort is likely even healthier than other UK Biobank participants. While analyses were conducted across three major biobanks, the vast majority of participants had European ancestry, limiting generalizability to diverse populations. While MTAG is a tool that has been shown to be robust for genetic discovery, the potential for false-positive results cannot be excluded and requires replication using other datasets. Further functional and prospective clinical validation is warranted before any clinical implementation of programs to manipulate cholesterol or phosphate levels for aortic stenosis prevention in humans.

In summary, we studied normal variation in aortic valve function, identifying 134 loci associated with aortic stenosis risk and 166 with aortic valve stenosis or function. We observed strong associations between aortic stenosis risk and coronary artery disease, lipoprotein biology and phosphate handling, suggesting future avenues for research to prevent the development or progression of aortic stenosis.

## Online content

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

[1]Cardiovascular Disease Initiative, Broad Institute of MIT and Harvard, Cambridge, MA, USA. [2]Department of Cardiology, University Heart and Vascular Center Hamburg–Eppendorf, Hamburg, Germany. [3]German Center for Cardiovascular Research (DZHK), Partner Site Hamburg/Kiel/Lübeck, Hamburg, Germany. [4]Cardiovascular Research Center, Massachusetts General Hospital, Boston, MA, USA. [5]Department of Medicine, Brigham and Women's Hospital, Boston, MA, USA. [6]Institute for Molecular Medicine Finland (FIMM), Helsinki Institute of Life Science (HiLIFE), University of Helsinki, Helsinki, Finland. [7]University of Minnesota Medical School, Minneapolis, MN, USA. [8]Department of Experimental Cardiology, Amsterdam Cardiovascular Sciences, Heart Center, Amsterdam UMC location University of Amsterdam, Amsterdam, the Netherlands. [9]Cardiology Division, Massachusetts General Hospital, Boston, MA, USA. [10]Telemachus and Irene Demoulas Family Foundation Center for Cardiac Arrhythmias, Massachusetts General Hospital, Boston, MA, USA. [11]Division of Cardiovascular Medicine, Brigham and Women's Hospital, Boston, MA, USA. [12]Cardiology Division, Beth Israel Deaconess Medical Center, Boston, MA, USA. [13]Division of Cardiology, University of California, San Francisco, San Francisco, CA, USA. [14]Harvard Medical School, Boston, MA, USA. [15]Thoracic Aortic Center, Massachusetts General Hospital, Boston, MA, USA. [16]Institute for Human Genetics, University of California, San Francisco, San Francisco, CA, USA. *A list of authors and their affiliations appears at the end of the paper. ✉e-mail: james.pirruccello@ucsf.edu

### FinnGen

**Joel T. Rämö**[1,5,6] **& Aarno Palotie**[6]

A full list of members appears in the Supplementary Information.

## Methods

### FinnGen ethics statement

Patients and control participants in FinnGen provided informed consent for biobank research, based on the Finnish Biobank Act. Alternatively, separate research cohorts, collected before the Finnish Biobank Act came into effect (in September 2013) and the start of FinnGen (August 2017), were collected based on study-specific consents and later transferred to the Finnish biobanks after approval by Fimea (Finnish Medicines Agency), the National Supervisory Authority for Welfare and Health. Recruitment protocols followed the biobank protocols approved by Fimea. The Coordinating Ethics Committee of the Hospital District of Helsinki and Uusimaa (HUS) issued a statement number HUS/990/2017 for the FinnGen study.

The FinnGen study is approved by Finnish Institute for Health and Welfare (permits THL/2031/6.02.00/2017, THL/1101/5.05.00/2017, THL/341/6.02.00/2018, THL/2222/6.02.00/2018, THL/283/6.02.00/2019, THL/1721/5.05.00/2019 and THL/1524/5.05.00/2020), Digital and population data service agency (permits VRK43431/2017-3, VRK/6909/2018-3 and VRK/4415/2019-3), the Social Insurance Institution (permits KELA 58/522/2017, KELA 131/522/2018, KELA 70/522/2019, KELA 98/522/2019, KELA 134/522/2019, KELA 138/522/2019, KELA 2/522/2020 and KELA 16/522/2020), Findata (permits THL/2364/14.02/2020, THL/4055/14.06.00/2020, THL/3433/14.06.00/2020, THL/4432/14.06/2020, THL/5189/14.06/2020, THL/5894/14.06.00/2020, THL/6619/14.06.00/2020, THL/209/14.06.00/2021, THL/688/14.06.00/2021, THL/1284/14.06.00/2021, THL/1965/14.06.00/2021, THL/5546/14.02.00/2020, THL/2658/14.06.00/2021 and THL/4235/14.06.00/2021), Statistics Finland (permits TK-53-1041-17 and TK/143/07.03.00/2020 (earlier TK-53-90-20), TK/1735/07.03.00/2021 and TK/3112/07.03.00/2021) and Finnish Registry for Kidney Diseases permission/extract from the meeting minutes on 4 July 2019.

The biobank access decisions for FinnGen samples and data used in FinnGen Data Freeze 12 include THL Biobank BB2017_55, BB2017_111, BB2018_19, BB_2018_34, BB_2018_67, BB2018_71, BB2019_7, BB2019_8, BB2019_26, BB2020_1 and BB2021_65; Finnish Red Cross Blood Service Biobank 7.12.2017; Helsinki Biobank HUS/359/2017, HUS/248/2020, HUS/150/2022 §12, §13, §14, §15, §16, §17, §18 and §23; Auria Biobank AB17-5154 and amendment 1 (7 August 2020) and amendments BB_2021-0140, BB_2021-0156 (26 August 2021, 2 February 2022), BB_2021-0169, BB_2021-0179, BB_2021-0161, AB20-5926 and amendment 1 (April 23 2020) and its modification (22 September 22 2021); Biobank Borealis of Northern Finland 2017_1013, 2021_5010, 2021_5018, 2021_5015, 2021_5023, 2021_5017 and 2022_6001; Biobank of Eastern Finland 1186/2018 and amendment 22 §/2020, 53§/2021, 13§/2022, 14§/2022 and 15§/2022; Finnish Clinical Biobank Tampere MH0004 and amendments (21.02.2020 and 06.10.2020) §8/2021, §9/2022, §10/2022, §12/2022, §20/2022, §21/2022, §22/2022 and §23/2022; Central Finland Biobank 1-2017, and Terveystalo Biobank STB 2018001 and amendment 25 August 2020; Finnish Hematological Registry and Clinical Biobank decision 18 June 2021; Arctic biobank P0844: ARC_2021_1001.

### Ethics statement

All patients gave informed consent, and all studies were approved by the regulatory authorities responsible for the respective study. This study adheres to the Declaration of Helsinki. This project was conducted under the UK Biobank (application 41664), the All of Us cohort and FinnGen. It was considered exempt by the University of California, San Francisco Institutional Review Board (22-37715) and approved by the MGB Institutional Review Board (protocol 2019P003144). Further detailed information is provided in the Supplementary Notes.

### Study overview

In brief, UK Biobank participants with cardiac MRI were analyzed to calculate physical measurements from velocity-encoded data[14]. GWAS were performed, and the heritability and genetic correlation

of the phenotypes were calculated. GWAS of aortic stenosis was conducted in the cMRI-unrelated subset of UK Biobank and in FinnGen and meta-analyzed with two prior aortic stenosis GWAS[6,19] results using METAL[55]. Those summary statistics were integrated with those from the velocity-encoded data using MTAG[29]. Polygenic scores were produced from the MTAG-enhanced summary statistics and applied in the All of Us cohort and MGB Biobank to test for their relationship with aortic stenosis. Additional GWAS of lipid traits were conducted in the cMRI-unrelated subset of UK Biobank to enable causal inference of lipid measurements on the cMRI-derived phenotypes using Mendelian randomization.

### MRI in UK Biobank

The UK Biobank is a prospective, general population-based cohort study that enrolled ~500,000 individuals in the UK between the ages of 40 and 69 years from 2006 to 2010. Informed consent was obtained from all participants. Comprehensive phenotyping, including questionnaires about family history, physical traits, lifestyle factors, laboratory values and imaging, was obtained for each participant. Inpatient electronic health records from Hospital Episode Statistics (England), Patient Episode Database (Wales) and Scottish Morbidity Records (Scotland) and National Health Service death registries are linked to the cohort[56,57]. The phenotypes in this study were generated from the cardiac MRI using deep learning[14].

### Statistical analysis

All statistical tests were two-tailed, and genome-wide significance was defined as $P < 5 \times 10^{-8}$. Statistical analyses were performed with R (v4.2.2) unless otherwise stated, with additional software detailed in the relevant subsections. Association tests were conducted and adjusted as outlined in each specific subsection.

### Definitions of diseases and outcomes in the UK Biobank

Definitions of diseases and outcomes are provided in Supplementary Table 1. Generally, we used self-reported data, as well as International Classification of Diseases (ICD) codes (ICD-9 and ICD-10), and procedural codes (OPCS-3 and OPCS-4) from National Health Service registries and the inpatient data from HES. We censored follow-up time on 31 October 2022 (for diagnostic and procedural codes) or 30 November 2022 (for mortality). Participants who had withdrawn consent from the UK Biobank before 25 October 2023 were excluded.

### Genotyping and imputation in the UK Biobank

For the UK Biobank, participants were genotyped with either the UK Biobank Axiom or UK BiLEVE arrays, and imputation was conducted with the UK10K + 1000 Genomes Phase III and Haplotype Reference Consortium panel[56]. Both arrays were used to capture short insertions and deletions as well as single-nucleotide polymorphisms (SNPs) with a shared marker content of 95%. QC for these genotyped variants included removal of variants with minor allele frequency (MAF) less than 1%, genotyping call rate less than 90% or Hardy–Weinberg violation within the GWAS participants at $P < = 1 \times 10^{-15}$.

Sample-level and QC included removal of those without imputed data, sample-level genotype missingness greater than or equal to 2%, aneuploidy of sex chromosomes, outliers for heterozygosity and excessive third-degree relatives as centrally computed by the UK Biobank[56]. For the genetic analysis, participants with the following cardiovascular diseases before the data of the cMRI were excluded: aortic valve surgery, thoracic aortic disease, bicuspid aortic valve, Marfan syndrome, congenital heart disease, anterior myocardial infarction or heart failure (Supplementary Table 1).

### GWAS

We conducted GWAS of three cMRI-derived measurements, including peak velocity, mean gradient and AVA. Before conducting each GWAS, a

rank-based inverse normal transformation was applied to the residuals of the measurements after accounting for the GWAS covariates[58]. The SNP heritability was assessed using BOLT-REML (v2.3.4)[59].

REGENIE v2.2.4 was used to conduct GWAS for each phenotype. REGENIE is a machine-learning-based approach that uses a two-step whole-genome regression to account for population structure and sample relatedness[60]. The analysis was adjusted for sex, age and age² at the time of cMRI, the genotyping array, the cMRI scanner's unique identifier and the first ten principal components of genetic ancestry that were centrally computed by UK Biobank[56].

In the first step, a prediction of individual trait values is generated using genetic data—specifically, the autosomal panel of genotyped variants remaining after the variant-level QC is used by REGENIE to calculate the leave-one-chromosome-out polygenic scores. Those scores are used in step 2, where REGENIE computed the effect size and association $P$ value for each of the centrally imputed SNPs from UK Biobank[56].

## Aortic stenosis GWAS

In the UK Biobank, using data from participants who did not undergo cMRI and were not within three degrees of relatedness to those who did, we conducted a GWAS of aortic stenosis diagnosed before death or censoring at the end of the follow-up period. REGENIE (v2.2.4) was used to conduct GWAS, which was adjusted for sex, age and age² at the end of follow-up time, the genotyping array and the first ten principal components of genetic ancestry that were centrally computed by UK Biobank[56]. In total, 5,038 cases and 412,301 controls contributed to the GWAS. We performed additional sensitivity analyses (1) limiting the definition of aortic stenosis to those with a diagnosis code of aortic stenosis and a procedure code for aortic valve intervention and (2) excluding those with a diagnosis of coronary artery disease before diagnosis of aortic stenosis.

## METAL aortic stenosis GWAS meta-analysis

Aortic stenosis GWAS summary statistics from the UK Biobank GWAS, the FinnGen Data Freeze 12 GWAS (Supplementary Note), the MVP GWAS, mentioned in ref. 6, and GWAS, mentioned in ref. 19, were meta-analyzed using METAL (v2020-05-05)[55]. METAL was run with effective-sample-size weighting and sample-overlap correction because the GWAS discussed in ref. 19 also incorporated UK Biobank data. As the GWAS summary statistics, mentioned in ref. 19, did not contain allele frequencies, the ALFA European ancestry allele frequency (https://www.ncbi.nlm.nih.gov/snp/docs/gsr/alfa/ALFA_20230706150541/) was assigned to each variant. For each cohort, the effective sample size was computed as recommended by the METAL authors—namely, twice the harmonic mean of cases and controls. Variants that were only present in MVP and not found in the other cohorts were removed before downstream analyses.

## MTAG

MTAG is a computational method used to jointly analyze GWAS of correlated traits, increasing statistical power to detect genetic associations[29]. The method is a generalization of IVW meta-analysis and provides updated effect sizes, effective sample sizes and $P$ values for the summary statistics for each trait. We used MTAG (v1.0.8) with its default settings in a four-way analysis that incorporated the REGENIE summary statistics for mean gradient, peak velocity and AVA, as well as the aortic stenosis GWAS meta-analysis. The MTAG method allows for the calculation of the 'maxFDR', the upper bound for the FDR under MTAG assumptions, which we also computed for this four-way analysis.

## MAGMA gene-set analysis

We used MAGMA (v1.09b) with its default settings, with the default gene and SNP annotations keyed to GRCh37 (refs. 41,42,61). These are based on a European ancestry panel from the 1000 Genomes Project,

which were precomputed by the MAGMA authors. Genes and gene sets were considered significant based on a per-trait FDR < 0.05 using $p$.adjust in R.

## Tissue and cell-type enrichment with MAGMA

Preparation of tissue and cell-type-specific gene sets is detailed in the Supplementary Methods. In brief, gene sets were constructed from GTEx v8 and from single-nucleus gene expression studies from aortic, left ventricular and aortic valve datasets. These gene sets were tested for association with each of the GWAS phenotypes using MAGMA (v1.09b) with default settings. GSEA software (v4.3.2) was used to characterize the identities of cell clusters where no author-provided cell cluster label was available[62].

## Lipid trait GWAS

We used REGENIE (v2.2.4) to conduct GWAS of the following circulating lipids taken at the time of UK Biobank enrollment: Lp(a), LDL, ApoB, ApoA and triglycerides. Participants were excluded if they underwent cMRI or were within three degrees of relatedness to participants who underwent cMRI. This left 342,663 participants with ApoA, 374,374 with ApoB, 300,666 with Lp(a) and 375,614 with LDL measurements for each GWAS. The lipid traits were transformed using a rank-based inverse normal transformation after residualization for the GWAS covariates, which included sex, age and age² at the time of follow-up, the genotyping array and the first ten principal components of genetic ancestry that were centrally computed by UK Biobank[56].

## Polygenic score derivation

Polygenic scores for the computed cMRI-derived phenotypes were constructed using the generated GWAS summary statistics using PRScs (v1.1.0) and the 'EUR' UK Biobank linkage disequilibrium panel precomputed by the PRScs authors[63]. This model is a Bayesian approach using continuous shrinkage of SNP weights and was run in 'auto' mode on a per-chromosome basis with the default settings. The 'auto' mode does not need validation data for tuning because it sets a standard half-Cauchy prior on the global shrinkage parameter to learn the global scaling parameter from provided data. The precomputed LD panel calculates the polygenic scores from 1.1 million SNPs from HapMap3 (ref. 64).

## Polygenic scores from MTAG

Polygenic scores for the MTAG-enhanced summary statistics were constructed using PRScs and the 'EUR' UK Biobank linkage disequilibrium panel precomputed by the PRScs authors[63]. This model is a Bayesian approach using continuous shrinkage of SNP weights and was run in 'auto' mode on a per-chromosome basis with the default settings. The 'auto' mode does not require validation data for tuning because it sets a standard half-Cauchy prior on the global shrinkage parameter to learn the global scaling parameter from provided data. The MTAG-augmented scores are calculated using the precomputed LD panel from 1.1 million SNPs from HapMap3 (ref. 64). The disease definition of aortic stenosis as an outcome for the PGS analysis in MGB is provided in Supplementary Table 20.

## Polygenic scores from Lipid trait GWAS

Using the lipid trait GWAS, we constructed polygenic scores using PRScs and the 'EUR' UK Biobank linkage disequilibrium panel precomputed by the PRScs authors[63]. These models were run in 'auto' mode on a per-chromosome basis. The precomputed LD panel calculates the polygenic scores from 1.1 million SNPs from HapMap3.

## Testing lipid polygenic scores

The constructed lipid-based scores were scaled to a mean of 0 and a s.d. of 1 and used as the independent variable in linear regression models adjusted for sex, age and age² at time of cMRI, the MRI serial number, the genotyping array and the first five principal components

of ancestry. The three aortic valve traits from those participants who underwent cMRI were similarly scaled and used as the dependent variable in the regression models. To test the strength of the lipid polygenic scores in predicting lipids, we calculated similar regression models with the scaled lipid polygenic scores as the independent variable and scaled measurements of circulating lipids taken at enrollment as the dependent variable.

## Mendelian randomization for aortic valve function and aortic stenosis

We used the TwoSampleMR (v0.4.9) package in R to perform Mendelian randomization. The approach followed common two-sample MR methods and the TwoSampleMR package defaults unless stated otherwise[65].

We constructed the instruments for Lp (a), LDL, ApoB, ApoA and triglycerides using GWAS generated from participants who did not undergo cMRI, and adjusted for statin usage. Additionally, we conducted GWAS of UK Biobank participants who had not undergone cMRI to construct instruments for height, weight, BMI, SBP and diastolic blood pressure; markers of kidney function (cystatin c and creatine); calcium, phosphate and vitamin D levels; IGF-1; measures of diabetic conditions (glycated hemoglobin A1c and blood glucose); and weekly American standard drinks and smoking in pack years. These GWAS were performed using REGENIE (v2.2.4). Participants were excluded if they underwent cMRI or were within three degrees of relatedness to participants who underwent cMRI. Blood-based continuous biomarker traits were transformed using a rank-based inverse normal transformation after residualization for the GWAS covariates, which included sex, age and age$^2$ at the time of follow-up, the genotyping array and the first ten principal components of genetic ancestry that were centrally computed by UK Biobank[56].

The lead SNPs ($P < 5 \times 10^{-8}$) were then clumped with a 10,000 kb window and $R^2 = 0.001$ based on in-sample linkage disequilibrium. The primary analysis was performed with the IVW method. Sensitivity analyses included MR Egger, which allows for pleiotropy, and weighted median, which assumes that at least half of the instruments are valid and is robust to outliers[66], weighted mode and simple mode. We also tested for horizontal pleiotropy with the MR Egger intercept and heterogeneity with Cochran's Q statistic. The effect sizes are provided as coefficients ($\beta$) per 1 s.d. of exposure.

### Reporting summary
Further information on research design is available in the Nature Portfolio Reporting Summary linked to this article.

## Data availability
Aortic valve measurements have been returned to the UK Biobank for access by researchers with UK Biobank access. Summary statistics for the GWAS meta-analyses have been deposited into the GWAS catalog (https://www.ebi.ac.uk/gwas/) under accessions GCST90651070, GCST90651071, GCST90651072, GCST90651073, GCST90651074, GCST90651075, GCST90651076 and GCST90651077, as well as to Zenodo (https://doi.org/10.5281/zenodo.14025285 (ref. 67)). The polygenic score weights are available on Zenodo (https://doi.org/10.5281/zenodo.15069071 (ref. 68)) and in the PGS Catalog (https://www.pgscatalog.org/) with publication ID PGP000747 and score ID PGS005254-5257. Individual-level data from Finnish biobanks can be accessed through the Fingenious services (https://site.fingenious.fi/en/) managed by FINBB. Finnish Health register data can be applied from Findata (https://findata.fi/en/data/). The All of Us biobank access is currently granted to researchers at academic, nonprofit and certain for-profit health institutions, with plans to expand. Researchers register through the All of Us Researcher Workbench (including identity verification and training) and must comply with Data Use Policies. Research projects using these data are publicly listed in the All of Us Research Projects Directory. Data from the MGB Biobank are available to researchers affiliated with MGB who have approval. It is currently not possible to access the MGB Biobank without an MGB affiliation. Further information regarding access can be obtained by emailing biobank@partners.org. MVP aortic stenosis summary statistics are available from dbGaP under accession phs001672. The Chen 2023 aortic stenosis summary statistics are available at https://zenodo.org/records/7829401 (ref. 69).

## Code availability
We used publicly available software to conduct analyses, details of which are listed in the Methods and the Supplementary Notes. The deep-learning semantic segmentation model is available for download at https://doi.org/10.5281/zenodo.14544762 (ref. 70).

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

## Acknowledgements
This work was supported by the National Institutes of Health (NIH; K08HL159346 and R01HL178603 to J.P.P.). S. Kany was supported by the Walter Benjamin Fellowship from the Deutsche

Forschungsgemeinschaft (521832260). J.T.R. was supported by a research fellowship from the Sigrid Jusélius Foundation. S.J.J. is supported by the Junior Clinical Scientist Fellowship from the Dutch Heart Foundation (grant 03-007-2022-0035). S.E. receives research grants from the NIH (R01 HL151838) and the Patient-Centered Outcomes Research Institute. A.P. was supported by the Academy of Finland Centre of Excellence in Complex Disease Genetics (grants 312074 and 336824). This work was supported by the Fondation Leducq (14CVD01) and by grants from the NIH (1RO1HL092577 and K24HL105780 to P.T.E.) and (R01HL168889, R01HL160003 and K24HL153669 to J.E.H.). V.N. was supported by the Sperling Family Fellowship. S. Khurshid was supported by grants from NIH (K23HL169839) and the American Heart Association (23CDA1050571). E.S.L. was supported by grants from NIH (K23-HL159243, R01HL180648) and the American Heart Association (853922). This work was supported by a grant from the American Heart Association Strategically Focused Research Networks to P.T.E. M.E.L. was supported by the Fredman Fellowship for Aortic Disease and the Toomey Fund for Aortic Dissection Research. We acknowledge the staff and participants of the UK Biobank, All of Us, MVP and FinnGen. Full acknowledgement is provided in the Supplementary Notes.

## Author contributions

J.P.P., S. Kany and P.T.E. conceived of the study. S. Kany, J.P.P., V.N. and J.W.C. annotated images used in the models of the studies. J.P.P. trained the deep-learning models. S. Kany and J.P.P. conducted QC and designed the postprocessing of data. S. Khurshid performed analyses for the MGB replication. S.J.J. performed the PGS analysis in MGB. J.T.R. performed GWAS and PGS analysis in FinnGen. J.P.P. conducted bioinformatic analyses in the UK Biobank and multitrait analyses. J.P.P., S. Kany and P.T.E. wrote the paper. J.T.R., C.H., S.J.J., S. Khurshid, V.N., J.W.C., E.S.L., S. Koyama, J.E.H., J.E.O., S.E., A.P. and M.E.L. contributed to the analysis plan or provided critical revisions.

## Competing interests

J.E.H. has received past research support from Bayer AG focused on machine learning and cardiovascular disease. E.S.L. has received previous honoraria and consulting fees from Roche Diagnostics and Astellas Pharma. S. Khurshid receives sponsored research support from Bayer AG. S.E. is supported by a grant from Edwards Lifesciences, Abbott Vascular and Medtronic; has received consulting fees from Edwards Lifesciences and holds equity in Prospect Health. P.T.E. is supported by a grant from Bayer AG to the Broad Institute focused on the genetics and therapeutics of cardiovascular diseases and has also served on advisory boards or consulted for Bayer AG, Quest Diagnostics, MyoKardia and Novartis. The other authors declare no competing interests.

## Additional information

**Correspondence and requests for materials** should be addressed to James P. Pirruccello.

# Reporting Summary

## Statistics

For all statistical analyses, confirm that the following items are present in the figure legend, table legend, main text, or Methods section.

| n/a | Confirmed | |
|---|---|---|
| ☐ | ☒ | The exact sample size (*n*) for each experimental group/condition, given as a discrete number and unit of measurement |
| ☐ | ☒ | A statement on whether measurements were taken from distinct samples or whether the same sample was measured repeatedly |
| ☐ | ☒ | The statistical test(s) used AND whether they are one- or two-sided<br>*Only common tests should be described solely by name; describe more complex techniques in the Methods section.* |
| ☐ | ☒ | A description of all covariates tested |
| ☐ | ☒ | A description of any assumptions or corrections, such as tests of normality and adjustment for multiple comparisons |
| ☐ | ☒ | A full description of the statistical parameters including central tendency (e.g. means) or other basic estimates (e.g. regression coefficient) AND variation (e.g. standard deviation) or associated estimates of uncertainty (e.g. confidence intervals) |
| ☐ | ☒ | For null hypothesis testing, the test statistic (e.g. *F*, *t*, *r*) with confidence intervals, effect sizes, degrees of freedom and *P* value noted<br>*Give P values as exact values whenever suitable.* |
| ☒ | ☐ | For Bayesian analysis, information on the choice of priors and Markov chain Monte Carlo settings |
| ☒ | ☐ | For hierarchical and complex designs, identification of the appropriate level for tests and full reporting of outcomes |
| ☐ | ☒ | Estimates of effect sizes (e.g. Cohen's *d*, Pearson's *r*), indicating how they were calculated |

*Our web collection on statistics for biologists contains articles on many of the points above.*

## Software and code

Policy information about availability of computer code

| Data collection | PyTorch v.2.1.0<br>TraceOverlay v0.1.0 |
|---|---|
| Data analysis | R4.2.2<br>BOLT-REML v2.3.4<br>REGENIE  v2.2.4<br>METAL v2020-05-05<br>MAGMA version 1.09b<br>MTAG 1.0.8<br>GSEA v4.3.2<br>PRScs v1.1.0<br>Two SampleMR v.0.4.9 |

For manuscripts utilizing custom algorithms or software that are central to the research but not yet described in published literature, software must be made available to editors and reviewers. We strongly encourage code deposition in a community repository (e.g. GitHub). See the Nature Portfolio guidelines for submitting code & software for further information.

## Data

Policy information about availability of data

All manuscripts must include a data availability statement. This statement should provide the following information, where applicable:
- Accession codes, unique identifiers, or web links for publicly available datasets
- A description of any restrictions on data availability
- For clinical datasets or third party data, please ensure that the statement adheres to our policy

Aortic valve measurements have been returned to the UK Biobank for access by researchers with UK Biobank access. Summary statistics for the GWAS meta-analyses have been deposited into the GWAS Catalog as well as to Zenodo (DOI 10.5281/zenodo.14025285(Kany and Pirruccello 2024)). The polygenic score weights are available on Zenodo (DOI 10.5281/zenodo.15069071(Kany and Pirruccello 2025)) and in the PGS Catalog. Individual-level data from Finnish biobanks can be accessed through the Fingenious® services (https://site.fingenious.fi/en/) managed by FINBB. Finnish Health register data can be applied from Findata (https://findata.fi/en/data/). The All of Us biobank access is currently only granted to researchers at academic, non-profit, and certain for-profit health institutions, with plans to expand. Researchers register through the All of Us Researcher Workbench (including identity verification and training) and must comply with Data Use Policies. Research projects using these data are publicly listed in the All of Us Research Projects Directory. Data from the MGB Biobank are available to researchers affiliated with MGB who have approval. It is currently not possible to access the MGB Biobank without a MGB affiliation. Further information regarding access can be obtained through email to biobank@partners.org. MVP aortic stenosis summary statistics are available from dbGaP under accession #phs001672. The Chen 2023 aortic stenosis summary statistics are available at https://zenodo.org/records/7829401 .

## Research involving human participants, their data, or biological material

Policy information about studies with human participants or human data. See also policy information about sex, gender (identity/presentation), and sexual orientation and race, ethnicity and racism.

| | |
|---|---|
| Reporting on sex and gender | Sex was determined by genetic and self-reported sex and particpants retained for analysis if both matched. |
| Reporting on race, ethnicity, or other socially relevant groupings | No special consideration on race and ethnicity in this work. The first 5 principal components of genetic ancestry were used for adjusting regression analyses. |
| Population characteristics | Informed consent was obtained from all participants. Comprehensive phenotyping including questionnaires about family history, physical traits, life-style factors, laboratory values and imaging was obtained for each participant. Inpatient electronic health records from Hospital Episode Statistics (England), Patient Episode Database (Wales) and Scottish Morbidity Records (Scotland) as well as National Health Service death registries are linked to the cohort.The imaging substudy of the UK Biobank is planned to perform 1.5 Tesla cardiac MRI in ca. 100,000 participants with ca. 65,000 studies in individual participants available as of the time of manuscript preparation.For the UK Biobank, participants were genotyped with either the UK Biobank Axiom or UK Bi LEVE arrays and imputation conducted with the UKI0K+ 1000 Genomes phase Ill and Haplotype Reference Consortium panel.<br>Out of the 59,571 participants in the UK Biobank, 52% were female, the mean age at the time of MRI was 65.9 years and mean body mass index 26.5 kg/m^2. Mean systolic blood pressure was 141mmHg, and a total of 341 had moderate aortic stenosis and 46 had severe aortic stenosis at baseline. |
| Recruitment | The UK Biobank is a prospective, general population-based cohort study that enrolled ~500,000 individuals in the UK between the ages 40-69 years from 2006-2010. |
| Ethics oversight | This project was conducted under UK Biobank application #41664, in the All of Us cohort as well as FinnGen. It was considered exempt by the UCSF IRB (#22-37715), and approved by the MGB institutional review board (IRB; protocol 2019P003144). |

Note that full information on the approval of the study protocol must also be provided in the manuscript.

# Field-specific reporting

Please select the one below that is the best fit for your research. If you are not sure, read the appropriate sections before making your selection.

☒ Life sciences  ☐ Behavioural & social sciences  ☐ Ecological, evolutionary & environmental sciences

For a reference copy of the document with all sections, see nature.com/documents/nr-reporting-summary-flat.pdf

# Life sciences study design

All studies must disclose on these points even when the disclosure is negative.

| | |
|---|---|
| Sample size | All available participants with cardiac imaging at the time of the study were included.  62,902 participants with deep -earning based flow measurements. A total of 3,331 participants were excluded for pre-existing cardiovascular disease or genetic quality control, leaving 59,571 participants contributing to the GWAS of peak velocity and mean gradient, and 59,569 contributing to GWAS of AVA. |

| Data exclusions | Participants were excluded if they failed the genetic quality control or had preexisting cardiovascular disease. |
| Replication | The GWAS of aortic valve function itself could not be replicated due to the lack of a similar cohort with cardiac imaging and lack of indication bias. The GWAS were used to create polygenic risk scores that were replicated in FinnGen and All of Us. |
| Randomization | This was a genetic study using continuous traits of a observational cohort study. |
| Blinding | Since there is no treatment investigated here, blinding does not apply. |

# Reporting for specific materials, systems and methods

We require information from authors about some types of materials, experimental systems and methods used in many studies. Here, indicate whether each material, system or method listed is relevant to your study. If you are not sure if a list item applies to your research, read the appropriate section before selecting a response.

## Materials & experimental systems

| n/a | Involved in the study |
|---|---|
| ☒ | Antibodies |
| ☒ | Eukaryotic cell lines |
| ☒ | Palaeontology and archaeology |
| ☒ | Animals and other organisms |
| ☒ | Clinical data |
| ☒ | Dual use research of concern |
| ☒ | Plants |

## Methods

| n/a | Involved in the study |
|---|---|
| ☒ | ChIP-seq |
| ☒ | Flow cytometry |
| ☒ | MRI-based neuroimaging |

## Plants

| Seed stocks | N/a, this is not a plant study. |
| Novel plant genotypes | N/a, this is not a plant study. |
| Authentication | N/a, this is not a plant study. |

