## [Peer Review File · Nature Genetics]

Multi-trait analyses identify genetic variants associated with aortic valve function and aortic stenosis risk

Corresponding Author: Dr James Pirruccello

Version 0:

Decision Letter:

18th Oct 2023

Dear Dr Pirruccello,

Your Article, "The genetic determinants of aortic valve function link 81 loci to aortic stenosis risk" has now been seen by 2 referees. You will see from their comments below that while they find your work of interest, some important points are raised. We are interested in the possibility of publishing your study in Nature Genetics, but would like to consider your response to these concerns in the form of a revised manuscript before we make a final decision on publication.

To guide the scope of the revisions, the editors discuss the referee reports in detail within the team with a view to identifying key priorities that should be addressed in revision. *In this case, we think both referees have provided constructive reviews aimed at strengthening the analyses and improving the presentation, and we particularly ask that you address their technical comments as thoroughly as possible with appropriate revisions. In addition, please provide further evidence on the robustness of the methodology since it has not been published in a peer-reviewed journal [ref.15 – Kany et al. medRxiv (2023) doi:10.1101/2023.04.29.23289299].* We hope that you will find the prioritized set of referee points to be useful when revising your study. Please do not hesitate to get in touch if you would like to discuss these issues further.

We therefore invite you to revise your manuscript taking into account all reviewer and editor comments. Please highlight all changes in the manuscript text file. At this stage we will need you to upload a copy of the manuscript in MS Word .docx or similar editable format.

*2) If you have not done so already please begin to revise your manuscript so that it conforms to our Article format instructions, available

[here](http://www.nature.com/ng/authors/article_types/index.html).

*3) Include a revised version of any required Reporting Summary: <https://www.nature.com/documents/nr-reporting-summary.pdf>

Link Redacted

We hope to receive your revised manuscript within 3 to 6 months. If you cannot send it within this time, please let us know.

Nature Genetics is committed to improving transparency in authorship. As part of our efforts in this direction, we are now requesting that all authors identified as 'corresponding author' on published papers create and link their Open Researcher and Contributor Identifier (ORCID) with their account on the Manuscript Tracking System (MTS), prior to acceptance. ORCID helps the scientific community achieve unambiguous attribution of all scholarly contributions. You can create and link your ORCID from the home page of the MTS by clicking on 'Modify my Springer Nature account'. For more information please visit please visit www.springernature.com/orcid.

Sincerely,
Wei

Wei Li, PhD
Senior Editor
Nature Genetics
New York, NY 10004, USA
www.nature.com/ng

Reviewers' Comments:

Reviewer #1:

Remarks to the Author:

Kany, Pirruccello and colleagues report an elegant genome-wide association study and multitrait analysis (MTAG) of 3 aortic valve (AV) related traits (aortic valve area, peak velocity and mean gradient; N=44,780 from UKB) as well as aortic stenosis (AS; 2 cohorts: UKB with 5038 cases; and FinnGen with 9853 cases). Using standard GWAS, 25 loci are reported for AV traits and 4 loci are reported for AS in the UKB. The authors then performed MTAG with 3 AV traits, and 2 AS GWAS (UKB+FinnGen) identifying an additional 36 loci associated with AV traits, and 50 loci associated with AS. The GWAS/MTAG is then followed by standard post-GWAS analyses, including gene-set and tissue enrichment analyses. Using Mendelian randomization, the authors show expected causal relation between lipoproteins or phosphocalcic metabolism and AS. The authors then derive polygenic scores (PGS) showing that they are associated with incident AS in All of Us, and prevalent AS in the Mass General Brigham Biobank.

Other GWAS of AS have been recently reported with similar or larger sample sizes. The main novel aspects of the present work are 1) the GWAS of AV traits in the general population, 2) leveraging such GWAS of AV traits to uncover loci associated with AS by using MTAG, and 3) the derivation of AV PGS, and their association with incident AS.

The methodology is sound and the results are generally well reported (with some exceptions, see minor comments below).

I have a few concerns/suggestions:

1) The authors should maximize statistical power of the AS GWAS. Why wasn't a GWAS meta-analysis performed for UKB+FinnGen followed by MTAG? This approach would maximize locus discovery of the GWAS. It would also be useful to leverage other published AS GWAS: e.g. PMID 37038246 (~11,000 cases excluding UKB), PMID 36802703 (14,451 cases from Million Veteran Program). A MTAG using a larger AS sample size may result in more robust findings, potentially reducing false-discovery rate for AS.

2) MTAG has been shown to be robust for locus discovery in other diseases and some cardiovascular traits/disease. Using MTAG is definitely very elegant to maximize power to discover loci associated with AS (although use for AV traits is less impactful). Nonetheless, since this is a first attempt to use MTAG in AS, it should be interpreted with caution regarding potential for false positive results. The authors do report the maxFDR as slightly high (~0.06). Can they attempt to replicate some of the loci in external cohorts (see comment #1). These cohorts/studies should be leveraged for locus discovery and/or replication, assuming the data can be made available. In addition, assuming that not all loci would be replicated, acknowledgement of potential for false-positive findings should be made in the limitations section.

3) A table with demographic and clinical data should be reported for all cohorts included in the AS GWAS and/or PGS

analyses, stratified by case status. Data on severity/actionability of AS should be reported (including AS grade if available, and AS-related procedures; see comments 4-5 below).

4) As expected from the Biobank study design, the AS case definition is imperfect. Ideally, a definition only including clinically actionable AS should be used (e.g. moderate or severe AS). Can a sensitivity analysis restricted to cases with moderate/severe AS be performed (e.g. defining cases as AS + aortic valve intervention) ?

5) Reporting of AS (including mild AS) in biobanks is subject to ascertainment bias. Patients with other cardiovascular/cardiometabolic diseases will be more likely to have cardiac imaging performed. This may potentially introduce bias in cross-trait analyses. For example, although there are many mechanistic and epidemiological studies linking AS to coronary artery disease (CAD), the observed association of AS loci with CAD may in part be mediated by such ascertainment bias, whereby patients with CAD are more likely to undergo cardiac imaging and be diagnosed with AS. Can the authors discuss and address such potential ascertainment bias?

6) The phenotyping of AV traits using CMR-based machine learning models is original and was developed by the lead authors. Since the methodology has not yet been published in a peer-reviewed journal (at the time of my revision), the authors should provide some data on the robustness of the approach and how it correlates with standard (echocardiography-derived) measures of AV function. Referral to the MedRxiv manuscript may be insufficient I believe since this is a key aspect of the study.

Minor comments, mostly to improve clarity:

- 1) Line 109, this sentence is unclear: "all marginal association P value within one order of magnitude of the $P < 5 \times 10^{-8}$ significance threshold" Please rephrase
- 2) Line 137, this sentence is unclear: "including 19 of the 26 loci identified in the unadjusted analysis (81 loci in total)"
- 3) Lines 144-145, the sentence on overlap with CAD is interesting but needs some context (also see major comment 5 above)
- 4) Line 148-150, a more complete description of prior AS loci should be provided here. How many AS loci were identified in previous studies? Of those, how many are validated in the present study? A supplementary table can be provided.
- 5) Line 159, what is the relevance of CHRN4 association with tobacco?
- 6) Line 160, all effect sizes ≥ 0.1 . Effect size in which trait?
- 7) Line 163, the sentence starting with ERG seems out of context?
- 8) Add titles and legend/abbreviations to the supplementary tables

Reviewer #2:

Remarks to the Author:

Summary

This paper reports new loci for aortic stenosis (AS) risk. The authors used deep learning to measure peak velocity, mean gradient and aortic valve area from cardiac magnetic resonance (CMR) imaging measurements from participants in the imaging study of UK Biobank (UKB). GWAS was performed on these traits, and then multi-trait analysis of GWAS (MTAG) was used to incorporate data from AV measures and AS diagnosis from both UKB and FinnGen. Polygenic scores were associated with AS and MR was used to test for causal relationships of lipid measures with interesting results from Lp(a) and LDL on AV function and elevated phosphate on AS.

Comments to Authors

This is an interesting paper with discovery of new loci contributing to AV function and AS risk, with nice downstream analyses using the new results for reporting potentially causal relationship. I have a few comments for the authors to consider:

- Abstract – you are reporting a lot of results from all analyses performed in your paper, I suggest some additional information on the methods and datasets for the PRS and MR work and motivation. You could reduce the text on gene reporting and processes as it is not clear which genes map to which trait in the current reporting
- Introduction – you could expand the literature quoted for quantitative phenotypes?
- Results _ Figure 1 – suggest as 6 sub figures, they are labelled as a-f, or prepare as one flow figure?
- Results - The genetic correlation between mean gradient and peak velocity was 0.986, why did you perform a GWAS for each trait, what was the overlap in findings
- Results – you indicate 23 loci you found for CMR valve traits were not in a prior GWAS of AS, what were the p values in a look up, was there some statistical support?
- Results – MTAG – for this analysis your input was 5 traits, I was interested in this approach and the robustness of the results. Two of the CMR phenotypes have a correlation of 0.986 and you also included 2 datasets for the same trait AS. Can you justify this approach and if you performed MTAG with one of the highly correlated CMR trait and a meta-analysis dataset for AS – are the results consistent. The number of loci discovered using this method dramatically increases loci yield so it would be good to have some sensitivity analyses reported as the subsequent analyses are based on these findings and there is no formal replication.
- Results – the PRS and MR work is interesting and nicely motivated by the results from the loci you discovered. As mentioned above, the rationale for the lipids and also the serum calcium and phosphate focus should be justified in the

abstract as per here.

- Results – mapping back to abstract again, I suggest reporting of the PRS in the two US cohorts earlier in the abstract and the paper for prediction of AS, this follows better from the AS/endophenotype discoveries before exploration of pathways, genes and MR.
- Table 1 – title needs to indicate which resource
- Table 2- not readable – would be good as one of the suppl tables, and simplified as a main table – maybe put on 2 pages, it is useful needs chr build, so legend.
- Figure 3 – this also needs to be branded as MTAG result
- Methods – multi -ancestry GWAS – is there any heterogeneity across ancestries?

Minor comments

- Missing page numbering, please do this as standard.
- Suppl. Tables – there are no legends, these need to be added for knowledge of the column headers at least as many abbreviations, the reviewer needs to work these out which can lead to mis-interpretation

Version 1:

Decision Letter:

Our ref: NG-A63301R

1st Nov 2024

Dear Dr. Pirruccello,

Thank you for submitting your revised manuscript "The genetic determinants of aortic valve function link 166 loci to aortic stenosis risk" (NG-A63301R). It has now been seen by the original referees and their comments are below. The reviewers find that the paper has improved in revision, and therefore we'll be happy in principle to publish it in Nature Genetics, pending minor revisions to satisfy the referees' final requests and to comply with our editorial and formatting guidelines.

Sincerely,
Wei

Wei Li, PhD
Senior Editor
Nature Genetics
www.nature.com/ng

Reviewer #1 (Remarks to the Author):

The authors have properly addressed my major concerns, and the manuscript now represents a major contribution to the field of genetics of aortic valve function and disease.

The following remaining comments are minor:

1) Title: The genetic determinants of aortic valve function link 166 loci to aortic stenosis risk

The authors are correct that the MTAG identifies 166 loci associated with either AV function and/or AS. However, the title claiming that 166 loci are linked to AS risk is not supported by the presented data. If the authors want to keep this title, they need to provide evidence that all 166 loci are linked to AS risk. At minimum, lead variants at all 166 loci would need to be significantly associated with AS in the AS GWAS METAL meta-analysis (not MTAG) using a bonferroni corrected threshold ($P < 0.05/166$) or similar correction for multiple "targeted" testing.

2) Line 203: "...remaining loci had still had...". Please rephrase.

3) Line 227: "All five polygenic scores were significantly associated ...". It is unclear what are these 5 scores, and Table S5 only lists 4 scores.

4) It is unclear why there are supplemental tables in two different sequences (A-M, and 1-6). This is a little confusing. For each supplemental table, please provide a list of abbreviations used for clarity.

- 5) Figure 4, specify if the beta are those of the MTAG or GWAS. If beta differs between GWAS and MTAG, it is preferable to report that of the GWAS as these are less biased by the other correlated traits.
- 6) Figure 5, provide the numbers at risk below the KM curves (at time 0 and each year of follow-up). I also suggest to add the results from MGB as a second panel (using the same PRS). Provide the HR with 95%CI on the figure and/or legend.
- 7) Figure S4, Units (mmHg, cm²) are not needed since only P values are presented. If the authors decide to include the units, please check if the unit for peak velocity should be cm/s or m/s.
- 8) Lines 356-358 "In comparison, the best-performing disease-based PGS for AS from a recent study based on MVP showed a ~1.4 increase in odds for AS." This sentence is confusing. The effect size in the Small et al refers to that per SD increase in PRS which is similar to what is reported by the authors using their PRS (1.6). The sentence placed here gives the impression that the "new" PRS developed by the authors performs better which is not supported by the presented data. Please remove sentence OR rephrase OR provide the performance of the Small et al PRS in AllOfUs to directly compare the "published" with the "new" PRS.

Reviewer #2 (Remarks to the Author):

Thank you to the authors for their extensive revision of their manuscript. With the increase in sample size and application of MTAG, there has been a substantial uplift in loci and performance of PRS and the reporting of new biological mechanisms. It is great the methodology is also now included in this manuscript. In response to reviewer 1 – many sensitivity analyses have been performed which is reassuring and supports the main observations.

Thank you for carefully addressing each of my comments and with the new analyses providing new figures and tables. Figure 1 is now very clear and the abstract very informative and clear.

We thank the Editors and Reviewers for the crucial comments and valuable suggestions to improve our manuscript. In response to these comments, we have performed a comprehensive major revision of the study, that includes updated GWAS, expanded downstream analyses as well as contextualization of our new findings:

- We now updated our deep learning model that further refines the edges around the aortic blood pool and, in turn, have updated our sample size from 44,870 participants to 59,571 participants that contribute to GWAS of continuous aortic valve traits. The updated sample size now yields 90 loci at genome-wide significance of which 61 were unique loci associated with at least one trait (up from 26 loci in the previous version).
- We now describe the machine learning approach to segmentation in detail in the supplementary methods.
- We now perform a meta-analysis of aortic stenosis cases in FinnGen, UK Biobank and published summary statistics from Chen et al. (Yu Chen et al. 2023) and Small et al. (Small et al. 2023) before MTAG, yielding 92 loci at genome-wide significance. Previously, both FinnGen and UK Biobank were separately used as input for MTAG.
- The updated MTAG analysis incorporating the larger sample size from the continuous traits and the extended meta-analysis now yields 166 loci across all traits. This approach leads to a maxFDR, the upper bound for the false discovery rate, of 0.015 for peak velocity, 0.014 for mean gradient, 0.012 for aortic valve area, and 0.003 for aortic stenosis.
- These GWAS were then used to construct new polygenic scores that were tested in *All of Us* and Mass General Brigham Biobank for the association with incident aortic stenosis. Using the polygenic score based on the AS GWAS after MTAG, participants in the top 5% of polygenic risk have a more than three-fold increase in risk for incident AS in *All of Us* (HR 3.32, 95% CI 2.60-4.4.24, $P=8.8 \times 10^{-22}$).
- We used these extended GWAS findings to then re-analyse all MAGMA, tissue and cell-type enrichment analyses. This included single cell datasets from aortic valves, aortic tissue, left ventricle as well as adjacent tissue such as esophageal tissue.
- Due to our findings that included many loci associated with risk factors such as lipid metabolism, phosphate and calcification pathways, we now perform Mendelian randomization analysis of common risk factors such as BMI, systolic and diastolic blood pressure, phosphate and lipids among others for the association with aortic stenosis or aortic valve dysfunction.

Reviewer #1:

Remarks to the Author:

Kany, Pirruccello and colleagues report an elegant genome-wide association study and multitrait analysis (MTAG) of 3 aortic valve (AV) related traits (aortic valve area, peak velocity and mean gradient; N=44,780 from UKB) as well as aortic stenosis (AS; 2 cohorts: UKB with 5038 cases; and FinnGen with 9853 cases). Using standard GWAS, 25 loci are reported for AV traits and 4 loci are reported for AS in the UKB. The authors then performed MTAG with 3 AV traits, and 2 AS GWAS (UKB+FinnGen) identifying an additional 36 loci associated with AV traits, and 50 loci associated with AS. The GWAS/MTAG is then followed by standard post-GWAS analyses, including gene-set and tissue enrichment analyses. Using Mendelian randomization, the authors show expected causal relation between lipoproteins or phosphocalcic metabolism and AS. The authors then derive polygenic scores (PGS) showing that they are associated with incident AS in All of Us, and prevalent AS in the Mass General Brigham Biobank.

Other GWAS of AS have been recently reported with similar or larger sample sizes. The main novel aspects of the present work are 1) the GWAS of AV traits in the general population, 2) leveraging such GWAS of AV traits to uncover loci associated with AS by using MTAG, and 3) the derivation of AV PGS, and their association with incident AS.

Author response:

Thank you for your interest in our work. We appreciate the thoughtful review and the kind words.

The methodology is sound and the results are generally well reported (with some exceptions, see minor comments below).

I have a few concerns/suggestions:

1) The authors should maximize statistical power of the AS GWAS. Why wasn't a GWAS meta-analysis performed for UKB+FinnGen followed by MTAG? This approach would maximize locus discovery of the GWAS. It would also be useful to leverage other published AS GWAS: e.g. PMID 37038246 (~11,000 cases excluding UKB), PMID 36802703 (14,451 cases from Million Veteran Program). A MTAG using a larger AS sample size may result in more robust findings, potentially reducing false-discovery rate for AS.

Author response:

Thank you, this is a great idea. We have now conducted a meta-analysis of disease-based GWAS of AS using METAL and accounting for sample overlap, including data from Chen et al. (13,765 cases; 640,102 controls)(Yu Chen et al. 2023), from Small et al. (14,451 cases; 398,544 controls)(Small et al. 2023), from all UK Biobank participants who did not undergo cMRI (5,038 cases; 412,301 controls), and from the freeze 12 of FinnGen (12,398 cases;

487,930 controls). This meta-analysis yielded 92 loci at genome-wide significance which we then followed with MTAG with the continuous AV traits. This MTAG analysis yielded 134 loci for AS and 134 loci for the continuous AV traits (not a typo; by chance the count is 134 for each). The majority are shared by AS and the AV traits, so that the total number of distinct loci is 166 across all traits.

As anticipated by the Reviewer, the maxFDR, the upper bound for the false discovery rate, improved to 0.015 for peak velocity, 0.014 for mean gradient, 0.012 for aortic valve area, 0.003 for AS.

Changes to the Manuscript:

The **Results section** now states:

Common variants related to aortic stenosis

We conducted a meta-analysis of disease-based GWAS of AS using METAL, including data from Chen et al. (13,765 cases; 640,102 controls)(Yu Chen et al. 2023), from Small et al. (14,451 cases; 398,544 controls)(Small et al. 2023), from all UK Biobank participants who did not undergo cMRI (5,038 cases; 412,301 controls), and from the freeze 12 of FinnGen (12,398 cases; 487,930 controls). The Chen, et al, and UK Biobank analyses had sample overlap, which was accounted for in our analyses. The baseline characteristics of all biobank participants included are provided in Supplementary Tables 1-2. For this disease-based GWAS meta-analysis, we observed 92 loci at genome-wide significance, including the 20 out of 23 previously described loci for calcific AS (Table 2, Supplementary Table D). We also confirmed 29 out of 32 loci from the meta-analysis by Theriault et al.(Thériault et al. 2024) which was not included in our meta-analysis before MTAG for genetic discovery (but has a partial sample overlap) (Supplementary Table G).

Additionally, we observed newly associated loci near LDLR ($\beta=0.031$, $P=1.2 \times 10^{-13}$) and a missense variant in PCSK9 ($\beta=0.089$, $P=2.4 \times 10^{-11}$), both of which are loci harboring genes that regulate blood levels of apolipoprotein B-rich lipids through the LDL receptor. Furthermore, in addition to the previously implicated genes in lipid metabolism and AS (FADS2/MYRF [$\beta=0.047$, $P=1.2 \times 10^{-32}$] and SORT1/PSRC1 [$\beta=-0.033$, $P=1.6 \times 10^{-20}$]), we identified variants near SORL1 ($\beta=-0.024$, $P=4.8 \times 10^{-08}$) which encodes the sortilin-related receptor (Rogaeva et al. 2007). SORL1 is a receptor for ApoE but has also been linked to IL6 signaling(Carlo 2013; Larsen and Petersen 2017). We additionally observed several other loci associated with lipid metabolism such as ALPL, SCARB1, LPL and ANGPTL4. Both latter genes are important in triglyceride metabolism. Lipoprotein lipase is encoded by LPL and involved in the lipolysis of triglycerides in lipoproteins(Li et al. 2014). One of the key regulators of LPL is angiopoietin-like protein 4 (ANGPTL4) that inhibits lipoprotein lipase similarly to ANGPTL3(Musunuru et al. 2010; Köster et al. 2005).

Among the previously described and replicated loci for AS is IL6 ($\beta=0.051$, $P=9.2 \times 10^{-45}$). In addition to this well-known locus, we also observed a variant near IL6R ($\beta=0.025$, $P=7.6 \times 10^{-11}$), encoding the IL6 receptor.

We additionally observed loci with key regulators of phosphate homeostasis, such as the previously reported *ALPL* ($\beta=0.031$, $P=4.2 \times 10^{-18}$) and, here for the first time observed in connection to AS, *FGF23* ($\beta=-0.028$, $P=3.2 \times 10^{-13}$) (ADHR Consortium 2000).

We observed similar results in UK Biobank and FinnGen when a) limiting our definition of AS cases to those who also had a procedure code for aortic valve intervention or b) excluding those with a diagnosis of coronary artery disease before AS to account for a possible ascertainment bias (Supplementary Table 3).

Of the 61 loci identified in the analysis of AV traits, 19 loci were also identified among the 241 loci associated with coronary artery disease in Aragam, et al (Aragam et al. 2022) (see Supplemental Notes). A PGS based on GWAS of aortic valve traits was predictive of AS in FinnGen. Being in the top 5% for genetically predicted mean gradient led to an HR of 1.44 for AS (95% CI 1.35-1.54, $P=1.1 \times 10^{-27}$) compared to the bottom 95% (see Supplementary Results).

Multitrait analysis of GWAS identifies 166 loci for aortic valve function and aortic stenosis

Having established that quantitative aortic valve measurements were heritable biomarkers for aortic valve stenosis, we pursued multi-trait analysis of GWAS (MTAG) to incorporate information from these related traits (Turley et al. 2018). This analysis included the three AV traits and the AS meta-analysis. This yielded an effective sample size of $N=96,385$ for peak velocity, $N=98,645$ for mean gradient, $N=77,183$ for AVA, and $N=205,483$ for AS. The maxFDR, the upper bound for the false discovery rate, was 0.015 for peak velocity, 0.014 for mean gradient, 0.012 for aortic valve area, and 0.003 for AS.

Using this approach, the number of loci associated with at least one of the three aortic valve traits at genome-wide significance increased from 90 to 273 loci overall (90 for AVA, 88 for mean gradient and 95 for peak velocity) and from 61 unique loci to 134 unique loci between all three traits (Supplemental Table F, Figure 3) while the number of AS loci increased from 92 in the METAL meta-analysis to 134 post-MTAG (Figure 4). When considered altogether, 166 distinct loci were identified across the aortic valve endophenotypes and AS GWASes within the MTAG framework (Supplementary Table F). Out of these, 32 loci were associated only with AS but did not reach genome-wide significance for any of the aortic valve endophenotypes, including *PCSK9* and *IL6R*. However, these loci had substantial sub-threshold signals as seen in Table 2. A sensitivity analysis leaving out mean gradient (which is highly correlated with peak velocity) showed similar yield in loci (see Supplementary Results).

While 93 MTAG loci were overlapping between at least one valve measurement GWAS and one AS GWAS with $P < 5 \times 10^{-8}$ for both, the remaining loci had still had a $P < 5 \times 10^{-5}$ (Figure 4).

2) MTAG has been shown to be robust for locus discovery in other diseases and some cardiovascular traits/disease. Using MTAG is definitely very elegant to maximize power to discover loci associated with AS (although use for AV traits is less impactful). Nonetheless, since this is a first attempt to use MTAG in AS, it should be interpreted with caution regarding potential for false positive results. The authors do report the maxFDR as slightly high (~0.06). Can they attempt to replicate some of the loci in external cohorts

(see comment #1). These cohorts/studies should be leveraged for locus discovery and/or replication, assuming the data can be made available. In addition, assuming that not all loci would be replicated, acknowledgement of potential for false-positive findings should be made in the limitations section.

Author response:

We fully agree with the Reviewer's suggestion. In response to the first point raised by the Reviewer as well as this point, we elected to use most publicly available datasets for genetic discovery in a meta-analysis using METAL, followed by MTAG with the continuous traits.

In our updated analysis, we calculate the highest maxFDR with 0.012 for AVA, 0.015 for peak velocity, 0.014 for mean gradient, and 0.003 for aortic stenosis. We additionally acknowledge the potential for false-positive findings in the limitation sections.

Change in manuscript:

The **Results section** now states:

Multitrait analysis of GWAS identifies 166 loci for aortic valve function and aortic stenosis

Having established that quantitative aortic valve measurements were heritable biomarkers for aortic valve stenosis, we pursued multi-trait analysis of GWAS (MTAG) to incorporate information from these related traits (Turley et al. 2018). This analysis included the three AV traits and the AS meta-analysis. This yielded an effective sample size of N=96,385 for peak velocity, N=98,645 for mean gradient, N=77,183 for AVA, and N=205,483 for AS. The maxFDR, the upper bound for the false discovery rate, was 0.015 for peak velocity, 0.014 for mean gradient, 0.012 for aortic valve area, and 0.003 for AS.

Using this approach, the number of loci associated with at least one of the three aortic valve traits at genome-wide significance increased from 90 to 273 loci overall (90 for AVA, 88 for mean gradient and 95 for peak velocity) and from 61 unique loci to 134 unique loci between all three traits (**Supplemental Table F, Figure 3**) while the number of AS loci increased from 92 in the METAL meta-analysis to 134 post-MTAG (Figure 4). When considered altogether, 166 distinct loci were identified across the aortic valve endophenotypes and AS GWASes within the MTAG framework (**Supplementary Table F**). Out of these, 32 loci were associated only with AS but did not reach genome-wide significance for any of the aortic valve endophenotypes, including PCSK9 and IL6R. However, these loci had substantial sub-threshold signals as seen in **Table 2**. A sensitivity analysis leaving out mean gradient (which is highly correlated with peak velocity) showed similar yield in loci (see **Supplementary Results**).

While 93 MTAG loci were overlapping between at least one valve measurement GWAS and one AS GWAS with $P < 5 \times 10^{-08}$ for both, the remaining loci had still had a $P < 5 \times 10^{-5}$ (**Figure 4**).

The **Discussion>Limitations** section now states:

While MTAG is a tool that has been shown to be robust for genetic discovery, the potential for false-positive results cannot be excluded and is in need for replication using other datasets. Further functional and prospective, clinical validation is warranted before any clinical implementation of programs to manipulate cholesterol or phosphate levels for AS prevention in humans.

The **Methods section** now states:

METAL aortic stenosis GWAS meta-analysis

AS GWAS summary statistics from the UK Biobank GWAS, the FinnGen Data Freeze 12 GWAS (Supplementary Note), the Small 2023 MVP GWAS (Small et al. 2023), and the Chen 2023 GWAS (Yu Chen et al. 2023) were meta-analyzed using METAL v2020-05-05 (Willer, Li, and Abecasis 2010). METAL was run with effective-sample-size weighting and sample-overlap correction because the Chen 2023 GWAS also incorporated UK Biobank data. Because the Chen 2023 GWAS summary statistics did not contain allele frequencies, the ALFA European ancestry allele frequency (from https://www.ncbi.nlm.nih.gov/snp/docs/gsr/alfa/ALFA_20230706150541/) was assigned to each variant. For each cohort, the effective sample size was computed as recommended by the METAL authors: namely, twice the harmonic mean of cases and controls. Variants that were only present in MVP and not found in the other cohorts were removed prior to downstream analyses.

The **Supplementary notes>Supplementary Figure 1** now shows the new study design:

UK Biobank cohort with cardiac MRI

Disease-based cohorts used for MTAG and polygenic risk scores

3) A table with demographic and clinical data should be reported for all cohorts included in the AS GWAS and/or PGS analyses, stratified by case status. Data on severity/actionability of AS should be reported (including AS grade if available, and AS-related procedures; see comments 4-5 below).

Author response:

We have included baseline characteristics stratified by case status for the overall UK Biobank, the UK Biobank MRI cohort, and FinnGen. AS grading was not possible to ascertain using ICD billing codes but is possible for the MRI UK Biobank cohort using our deep learning model which

we have included in table 1. Unfortunately, we do not have individual-level data of both PMID 37038246 and PMID 36802703.

Changes to the Manuscript:

The new **Table 1** now shows:

Table 1: Participant characteristics of the UK Biobank cohort with magnetic resonance imaging

	Women	Men	All
N	31159	28412	59571
Age at time of MRI	65.3 (7.65)	66.5 (7.85)	65.9 (7.77)
Height (cm)	163 (6.24)	176 (6.65)	169 (9.24)
Weight (kg)	69.0 (13.2)	83.5 (13.3)	75.9 (15.1)
Systolic blood pressure (mmHg)	139 (20.2)	144 (17.7)	141 (19.2)
Diastolic blood pressure (mmHg)	77.6 (10.1)	80.8 (9.92)	79.1 (10.2)
Body mass index (kg/m ²)	26.1 (4.78)	27.0 (3.91)	26.5 (4.41)
Aortic valve area (cm ²)	2.50 (0.403)	3.16 (0.545)	2.81 (0.578)
Peak velocity (m/sec)	1.16 (0.250)	1.17 (0.289)	1.16 (0.269)
Mean gradient (mmHg)	2.66 (1.23)	2.82 (1.57)	2.73 (1.41)
Moderate aortic stenosis	219 (1 %)	122 (0 %)	341 (1 %)
Severe aortic stenosis	25 (0 %)	21 (0 %)	46 (0 %)

Supplementary Tables 1 and 2 now show:

Table 1: Baseline Characteristics UK Biobank AS cohort

	AS Case	Control	All
N	5,038	41,2301	417,339
Age at enrollment	63.3 (5.6)	57.1 (8.1)	57.2 (8.1)
Age at censoring	75.1 (5.8)	69.4 (8.1)	69.4 (8.1)
Female	1,833 (36.4%)	225,402 (54.7%)	227,235 (54.4)
Body mass index (kg/m ²)	29.7 (5.5)	27.5 (4.8)	27.5 (4.8)
Hypertension	4,117 (81.7%)	162,226 (39.3%)	166,343 (39.9%)

Type 2 Diabetes	1,418 (28.1%)	34,859 (8.5%)	36,277 (8.7%)
Atrial Fibrillation	2,182 (43.3%)	30,251 (7.3%)	32,433 (7.8%)
Coronary Artery Disease	3,149 (62.5%)	43,961 (10.7%)	47,110 (11.3%)
AV intervention	2,086 (41.4%)	509 (0.1%)	2,595 (0.6%)

Table 2: Baseline Characteristics FinnGen AS cohort

	AS Case	Control	All
N	12,398	487,930	500,328
Age at DNA sampling	68.77 (12.22)	52.71 (17.83)	53.11 (17.89)
Age at end of follow-up	78.39 (10.20)	60.37 (17.96)	60.81 (18.03)
Female	4,684 (37.8)	277,370 (56.8)	282,054 (56.4)
Body mass index (kg/m ²)	28.11 (4.99)	27.32 (5.54)	27.35 (5.53)
Hypertension	8,928 (72.0)	145,695 (29.9)	154,623 (30.9)
Type 2 Diabetes	4,716 (38.0)	86,278 (17.7)	90,994 (18.2)
Atrial Fibrillation	5,673 (45.8)	57,854 (11.9)	63,527 (12.7)
Coronary Artery Disease	4,960 (40.0)	51,685 (10.6)	56,645 (11.3)
Aortic Stenosis	12,398 (100.0)	0 (0.0)	12,398 (2.5)
AV Intervention	2,833 (22.9)	453 (0.1)	3,286 (0.7)

4) As expected from the Biobank study design, the AS case definition is imperfect. Ideally, a definition only including clinically actionable AS should be used (e.g. moderate or severe AS). Can a sensitivity analysis restricted to cases with moderate/severe AS be performed (e.g. defining cases as AS + aortic valve intervention) ?

Author response:

This is an interesting point. To address the potential influence of AS definition, we have now performed sensitivity analyses in both UK Biobank and FinnGen by using an AS definition that mandates an ICD code for AS but also a procedural code for aortic valve surgery.

In this surgery + ICD code definition (N cases = 2,053), we observe the same 4 loci in UK Biobank that we observe when only using the ICD code for AS (N cases = 3,413). In FinnGen, we observe 4 loci compared with 34 using the main analysis which is explained by the large drop in cases from 12,398 to 2,833 using the more strict definition. These four loci are replicating 3 (PALMD, ZEB3 and LPA) observed in the UK Biobank analysis.

A more minor comment that we did want to make is that there are softer degrees of actionability that are already in play in the clinic. The only available *therapy* is, as the Reviewer rightly points out, either a surgical or catheter-based aortic valve replacement. Currently, this is limited to symptomatic individuals with severe AS or asymptomatic individuals with a decrement in left ventricular function. But if we consider the stage at which surveillance is recommended to be clinically relevant, then even mild AS is recommended for a follow-up every 2-3 years by the current ACC/AHA guidelines on the management of valvular heart disease (Otto et al. 2021).

Changes to the Manuscript:

The **Supplementary Results** section now states:

Sensitivity analysis using different definitions of aortic stenosis cases

To address the potential influence of the AS definition, we have now performed sensitivity analyses in both UK Biobank and FinnGen by using an AS definition that mandates an ICD code for AS but also a procedural code for aortic valve surgery. In this surgery + ICD code definition (N cases = 2,053), we observe the same 4 loci in UK Biobank that we observe when only using the ICD code for AS (N cases = 3,413). In FinnGen, we observe 4 loci compared with 34 using the main analysis which is explained by the large drop in cases from 12,398 to 2,833 using the more strict definition. These four loci are replicating 3 (PALMD, ZEB3 and LPA) observed in the UK Biobank analysis (**Supplementary Table 3**).

Additionally, we investigated whether indication bias for ascertainment of AS diagnoses used for the case/control meta-analysis could majorly influence our results. To this extent, we have now performed GWAS sensitivity analyses in UK Biobank and FinnGen excluding AS cases that were preceded by a diagnosis of CAD. In UK Biobank (N cases = 3,413), we retain the same 4 loci as in the main analysis. In FinnGen, we observed 23 loci (N cases = 9,584) when excluding those with CAD before AS diagnosis compared with 34 loci (N cases = 12,398) using the main definition (**Supplementary Table 3**).

Supplementary Table 3 now shows:

Table 3: Sensitivity analyses in UK Biobank and FinnGen

Cohort	UK Biobank, all	UK Biobank, No CAD	UK Biobank, Surgical AS	FinnGen, all	FinnGen, No CAD	FinnGen, Surgical AS
N Cases	5,038	3,413	2,053	12,398	9,584	2,833
N Controls	412,301	398,814	414,743	487,930	487,930	487,930
N Lead SNPs	4	4	4	34	23	4
Nearest Genes	PALMD, ZEB2, LPA, IL6	PALMD, ZEB2, LPA, IL6/STEA P1B	PALMD, ZEB2, LPA, IL6	LMO4, PALMD, ADAMTSL4, PRRX1, NAV1, DSTYK, RNF144A, ZEB2, TMEM44, AC058822.1, FER, SNCAIP, TSBP1, NMBR, LPA, IL6, FERD3L, FADS1/FADS2,	ALPL, PALMD, PRRX1, NAV1, RNF144A, ZEB2, TMEM44, AC058822.1/PD GFRA, FER, TSBP1, NMBR, LPA, DGKB, IL6, FADS1/FADS2, MYEOV, PDE32,	PALMD, RNF144A, ZEB2, LPA

				MYEOV, FGF23, PDE3A, PLXNC1, SMAD9, STARD9, SPG11, CHRN4, GLIS2, CFDP1, MEOX1, CA10, BAHCC1, TSPAN16, HORMAD2, RPS6KA3	STARD9, TRIM69, BCAR1, MEOX1, BAHCC1, TSPAN16	
--	--	--	--	---	---	--

5) Reporting of AS (including mild AS) in biobanks is subject to ascertainment bias. Patients with other cardiovascular/cardiometabolic diseases will be more likely to have cardiac imaging performed. This may potentially introduce bias in cross-trait analyses. For example, although there are many mechanistic and epidemiological studies linking AS to coronary artery disease (CAD), the observed association of AS loci with CAD may in part be mediated by such ascertainment bias, whereby patients with CAD are more likely to undergo cardiac imaging and be diagnosed with AS. Can the authors discuss and address such potential ascertainment bias?

Author response:

We share the Reviewer's concerns about ascertainment bias, which was part of the initial motivation for the arm of our approach that takes advantage of imaging-based phenotypes obtained for research across large samples irrespective of indication (rather than based on clinical indications).

Ascertainment bias likely plays a role in the diagnosis of aortic stenosis. The two most probable reasons for aortic stenosis diagnosis are either clinical work-up because of symptoms or through imaging for other indications. Critically, in the UK Biobank, imaging is performed for research purposes and therefore free of indication bias.

Consequently, we are able to assess whether the association of aortic stenosis loci with CAD is primarily due to ascertainment bias by checking the results for the three MRI-derived AV traits.

First, we sought to confirm that our findings were not driven only by the extremes of AV dysfunction—i.e., to confirm that normal variation in AV hemodynamics, even after excluding the extremes, would yield comparable results. We have performed a sensitivity analysis (see **Supplementary Notes**) excluding participants with aortic valve measurements consistent with moderate (341 participants) or severe (46 participants) AS. In these analyses, the number of associated loci actually increased from 61 to 64, including a loss of four loci near *OR5D13*, *ELN*, *TMEM170A* and *PSTPIP1*, and a gain of seven loci (near *ANK2*, *MAD2L1*, *MYBPC3*, *CSH2*, *PLAU*, *TBC1D12*, *PIGU*), all of which were within one order of magnitude of the $P < 5 \times 10^{-8}$ significance threshold (**Supplementary Table E**). This data supports that our findings in the AV traits are driven by variation within the normal range.

With respect to the more pressing concern of indication bias for ascertainment of AS diagnoses used for the case/control meta-analysis, we have now performed GWAS sensitivity analyses in UK Biobank and FinnGen excluding AS cases that were preceded by a diagnosis of CAD. In UK Biobank (N cases = 3,413), we retain the same 4 loci as in the main analysis. In FinnGen, we observed 23 loci (N cases = 9,584) when excluding those with CAD before AS diagnosis compared with 34 loci (N cases = 12,398) using the main definition. We submit that a reduction in effective sample size might be the driving factor for the observed results.

That being said, we acknowledge that this is not definitive evidence nor does it exclude potential ascertainment bias in the main results and have therefore added this limitation.

Changes to the Manuscript:

The **Results>Common variant related to aortic valve function** section now states:

In a sensitivity analysis (see Supplementary Notes) excluding participants with aortic valve measurements consistent with moderate (341 participants) or severe (46 participants) AS, the number of associated loci increased from 61 to 64, including a loss of four loci near OR5D13, ELN, TMEM170A and PSTPIP1, and a gain of seven loci (near ANK2, MAD2L1, MYBPC3, CSH2, PLAU, TBC1D12, PIGU), all of which were within one order of magnitude of the $P < 5 \times 10^{-8}$ significance threshold in the main analysis(**Supplementary Table E**).

The **Discussion** section now states:

Importantly, our findings underscore that these effects are evident in valvular function well before the onset of clinical AS. Critically, the imaging traits were obtained for research purposes, and so their genetic link to coronary artery disease is not influenced by imaging that was obtained due to coronary artery disease (i.e., confounding by indication). In a sensitivity analysis of the AS diagnosis-based GWAS that excluded those with a diagnosis of coronary artery disease occurring before their AS diagnosis, our observations remained largely consistent which suggests that ascertainment bias might not fully explain our findings. Further, these observations provide a genetic foundation for the notion that subclinical variation in aortic valve function may have utility as an early marker of atherosclerosis.

The **Supplementary Results** section now states:

Sensitivity analysis using different definitions of aortic stenosis cases

To address the potential influence of the AS definition, we have now performed sensitivity analyses in both UK Biobank and FinnGen by using an AS definition that mandates an ICD code for AS but also a procedural code for aortic valve surgery. In this surgery + ICD code definition (N cases = 2,053), we observe the same 4 loci in UK Biobank that we observe when only using the ICD code for AS (N cases = 3,413). In FinnGen, we observe 4 loci compared with 34 using the main analysis which is explained by the large drop in cases from 12,398 to 2,833

using the more strict definition. These four loci are replicating 3 (PALMD, ZEB3 and LPA) observed in the UK Biobank analysis (**Supplementary Table 3**).

Additionally, we investigated whether indication bias for ascertainment of AS diagnoses used for the case/control meta-analysis could majorly influence our results. To this extent, we have now performed GWAS sensitivity analyses in UK Biobank and FinnGen excluding AS cases that were preceded by a diagnosis of CAD. In UK Biobank (N cases = 3,413), we retain the same 4 loci as in the main analysis. In FinnGen, we observed 23 loci (N cases = 9,584) when excluding those with CAD before AS diagnosis compared with 34 loci (N cases = 12,398) using the main definition (**Supplementary Table 3**).

Supplementary Table 3 now shows:

Table 3: Sensitivity analyses in UK Biobank and FinnGen

Cohort	UK Biobank, all	UK Biobank, No CAD	UK Biobank, Surgical AS	FinnGen, all	FinnGen, No CAD	FinnGen, Surgical AS
N Cases	5,038	3,413	2,053	12,398	9,584	2,833
N Controls	412,301	398,814	414,743	487,930	487,930	487,930
N Lead SNPs	4	4	4	34	23	4
Nearest Genes	PALMD, ZEB2, LPA, IL6	PALMD, ZEB2, LPA, IL6/STEAP1B	PALMD, ZEB2, LPA, IL6	LMO4, PALMD, ADAMTSL4, PRRX1, NAV1, DSTYK, RNF144A, ZEB2, TMEM44, AC058822.1, FER, SNCAIP, TSBP1, NMBR, LPA, IL6, FERD3L, FADS1/FADS2, MYEOV, FGF23, PDE3A, PLXNC1, SMAD9, STARD9, SPG11, CHRN4, GLIS2, CFDP1, MEOX1, CA10, BAHCC1, TSPAN16, HORMAD2, RPS6KA3	ALPL, PALMD, PRRX1, NAV1, RNF144A, ZEB2, TMEM44, AC058822.1/PDGFRA, FER, TSBP1, NMBR, LPA, DGKB, IL6, FADS1/FADS2, MYEOV, PDE32, STARD9, TRIM69, BCAR1, MEOX1, BAHCC1, TSPAN16	PALMD, RNF144A, ZEB2, LPA

6) The phenotyping of AV traits using CMR-based machine learning models is original and was developed by the lead authors. Since the methodology has not yet been published in a peer-reviewed journal (at the time of my revision), the authors should provide some data on the robustness of the approach and how it correlates with standard (echocardiography-derived) measures of AV function. Referral to the MedRxiv manuscript may be insufficient I believe since this is a key aspect of the study.

Author response:

Thank you for the important point highlighted here. We now describe the overall deep learning approach in more detail within this present manuscript.

To describe it briefly here, too: our model was developed to specifically segment the images in the UK Biobank flow series to extract the velocities in the ascending aorta. This series has paired anatomical images and velocity-encoded images (where each pixel represents the same physical location in the both paired images), so that after mapping the anatomical location we can then read out just the aortic pixel velocities. Then, these velocities are quality controlled to account for aliasing and outlier values using an extended VENC window and the 99th percentile of measurements.

The segmentation model was trained using 1000 randomly selected images from the CINE sequences, which were manually annotated by a cardiology fellow (S.Kany). An attending cardiologist (J.P.P) then reviewed all annotations and, as needed, adjusted them to precisely capture the aortic blood pool.

For the segmentation, we have benchmarked the model against 30 randomly selected images annotated separately by J.P.P. The model achieved a mean Dice score of 0.973 (95% CI 0.967 to 0.979) for the ascending aortic blood pool. The Dice score considers the overlap between the pixels of the model with the pixels of J.P.P's annotation, with 1 denoting perfect overlap.

Therefore, we believe that the model is able to adequately segment the structure of interest.

Since our quality control and post-processing was multi-layered, we have now added a method section in the Supplementary Notes>Methods section that provides more detail about the UK Biobank MRI cohort and the semantic segmentation model.

Changes to the Manuscript:

The **Supplementary Methods** section now states:

Magnetic resonance imaging in UK Biobank

The UK Biobank is a prospective, general population-based cohort study that enrolled ~500,000 individuals in the UK between the ages 40-69 years from 2006-2010(Sudlow et al. 2015). Informed consent was obtained from all participants. Comprehensive phenotyping including questionnaires about family history, physical traits, life-style factors, laboratory values and imaging was obtained for each participant. Inpatient electronic health records from Hospital

Episode Statistics (England), Patient Episode Database (Wales) and Scottish Morbidity Records (Scotland) as well as National Health Service death registries are linked to the cohort (Bycroft et al. 2018).

The imaging substudy of the UK Biobank is planned to perform 1.5 Tesla cardiac MRI in ca. 100,000 participants with ca. 65,000 studies in individual participants available as of the time of manuscript preparation (Raisi-Estabragh et al. 2021). The cardiac MRI images were captured in a 20 minute study using a Siemens 1.5 Tesla MAGNETOM Aera scanner (Siemens Healthineers, Erlangen, Germany). In this study, phase contrast flow images aimed to be placed above the sinotubular junction at end-diastole were used. Phase contrast imaging uses magnitude scans as reference scans and velocity-encoded scans to create phase contrast velocity maps with a planned standard velocity encoding (VENC) of 2 m/s (Petersen et al. 2015). Over the cardiac cycle, 30 images with a slice thickness (depth) of 6 mm and a voxel size of 1.77 x 1.77 mm with retrospective gating were acquired. Consequently, the amount of time represented by each image varied among participants (due to varying heart rates).

Generating training and test data for segmentation

A cumulative density function (CDF) was generated from the non-background, non-lung components of the manually annotated CINE images as defined by the manually traced segmentation masks. The pixel intensities were rescaled based on the inverse of this cumulative density function, similar to the approaches described by Nyúl, et al (Nyul, Udupa, and Zhang 2000), and Shinohara, et al (Shinohara et al. 2014). The Kornia library (Riba et al. 2019) was then used to augment the data during training. The model was fully unfrozen and trained for 500 epochs with PyTorch using the AdamW optimizer with the default weight decay (0.01). The training schedule was a OneCycleLR schedule, which was described by Smith and Topin for model superconvergence (Smith and Topin 2019). The loss function was a focal Dice loss with a gamma parameter of 2 (Lin et al. 2020), with the focal loss component given 95% of the weight and the Dice loss component given 5% of the weight (Kirillov et al. 2023). At each epoch, a Dice score was computed for the ascending aortic blood pool in the validation samples and averaged; if this average value was superior to any prior epoch, then the model weights were saved. The model weights from the epoch with the best validation Dice score (epoch 276) were saved for downstream use. The model was then applied to all available CINE images.

Segmentation quality control

After applying the model to all CINE images, the output segmentation masks underwent heuristic quality control for the ascending aortic blood pool using an approach that has been previously described (Pirruccello, Chaffin, et al. 2022). Images without exactly one connected component for the ascending aortic blood pool were flagged. The instantaneous frame-to-frame change in the number of pixels attributed to the ascending aortic blood pool was computed, and any study above or below 5 standard deviations from the mean shift was flagged. Images were also flagged if both systolic and diastolic phases were not detected. Any flagged image was removed from analysis. Only participants with complete studies (those having 30 images that satisfied quality control) were retained for downstream analysis.

Extracting velocity-encoded values

Velocity at each pixel was computed within the aortic blood pool. The CINE segmentation masks were overlaid on their paired velocity-encoded images, which allowed for velocity-based measurements to be computed for labeled regions. For each aortic blood pool pixel at each of 30 time points throughout the cardiac cycle, the through-plane velocity at that pixel was extracted from the paired velocity-encoded image. For each image, the VENC value was retrieved from the Siemens header (DICOM group 0x0029, element 0x1010). "Bits_stored" was uniformly defined as 12 in the DICOM metadata (DICOM group 0x0028, element 0x0101). Therefore, the pixel data encoded a range of intensity values from 0 through 4095 (i.e., $2^{12}-1$). These were remapped to velocity values with units of centimeters per second, ranging from -VENC to +VENC, using Formula 1.

Formula 1

$$\frac{2 \times VENC \times \text{pixel intensity}}{2^{\text{bits_stored}} - 1} - VENC$$

That formula yielded a through-plane velocity value for each pixel. Aggregating these values from all pixels at each frame allowed for the calculation of bulk properties such as Velocity Time Integral (VTI; necessary for computing the AVA) and forward stroke volume.

Deriving phenotypes from segmentation and velocity encoding data

Aortic diameter was computed from the CINE segmentation masks, after accounting for the physical representation of each pixel in centimeters from the DICOM metadata and computing the elliptical minor axis diameter at its largest point in systole using image moments, as previously described (Horn, Klaus, and Horn 1986; Pirruccello, Chaffin, et al. 2022).

Peak velocity was determined by identifying the 99th percentile velocity for all pixels at each time point, and retaining the maximum value at any time point during systole. The 99th percentile velocity was selected as a heuristic to reduce spuriously high peak velocity values attributable to noise. The 99th percentile velocity was used to compute the gradient across the valve (using the simplified Bernoulli equation $4v^2$) for each frame, and then the mean value of that gradient at all time points during systole was taken as the mean gradient.

Forward stroke volume was computed by summing all pixel-wise forward volumes (computed by multiplying velocity, width, height, and duration of each frame) for the aortic blood pool during systole.

Aortic valve area was computed by dividing forward stroke volume by VTI (Yap et al. 2007). The 99th percentile pixel velocity was treated as the boundary of the VTI envelope throughout the cardiac cycle in order to calculate VTI for this formula.

Minor comments, mostly to improve clarity:

1) Line 109, this sentence is unclear: "all marginal association P value within one order of magnitude of the $P < 5 \times 10^{-8}$ significance threshold" Please rephrase

Author response:

Thanks, we have rephrased. This statement was intended to refer to the loss or gain of loci at genome-wide significance in the sensitivity analysis that were close to the threshold of $5E-8$ in the main analysis.

Changes to the Manuscript:

The **Results>Common variant related to aortic valve function** section now states:

In a sensitivity analysis (see Supplementary Notes) excluding participants with aortic valve measurements consistent with moderate (341 participants) or severe (46 participants) AS, the number of associated loci increased from 61 to 64, including a loss of four loci near OR5D13, ELN, TMEM170A and PSTPIP1, and a gain of seven loci (near ANK2, MAD2L1, MYBPC3, CSH2, PLAU, TBC1D12, PIGU), all of which were within one order of magnitude of the $P < 5 \times 10^{-8}$ significance threshold in the main analysis(**Supplementary Table E**).

2) Line 137, this sentence is unclear: "including 19 of the 26 loci identified in the unadjusted analysis (81 loci in total)"

Author response:

Thanks, we have removed this phrase. The “unadjusted” referred to the GWAS of continuous traits before MTAG. The section now provides a detailed summary of our findings after MTAG.

Changes to the Manuscript:

The **Results** section now states:

Multitrait analysis of GWAS identifies 166 loci for aortic valve function and aortic stenosis

Having established that quantitative aortic valve measurements were heritable biomarkers for aortic valve stenosis, we pursued multi-trait analysis of GWAS (MTAG) to incorporate information from these related traits (Turley et al. 2018). This analysis included the three AV traits and the AS meta-analysis. This yielded an effective sample size of $N=96,385$ for peak velocity, $N=98,645$ for mean gradient, $N=77,183$ for AVA, and $N=205,483$ for AS. The maxFDR, the upper bound for the false discovery rate, was 0.015 for peak velocity, 0.014 for mean gradient, 0.012 for aortic valve area, and 0.003 for AS.

Using this approach, the number of loci associated with at least one of the three aortic valve traits at genome-wide significance increased from 90 to 273 loci overall (90 for AVA, 88 for mean gradient and 95 for peak velocity) and from 61 unique loci to 134 unique loci between all three traits (**Supplementary Table F, Figure 3**) while the number of AS loci increased from 92 in the METAL meta-analysis to 134 post-MTAG (**Figure 4**). When considered altogether, 166 distinct loci were identified across the aortic valve endophenotypes and AS GWASes within the MTAG framework (**Supplementary Table F**). Out of these, 32 loci were associated only with AS but did not reach genome-wide significance for any of the aortic valve endophenotypes, including PCSK9 and IL6R. However, these loci had substantial sub-threshold signals as seen

in **Table 2**. A sensitivity analysis leaving out mean gradient (which is highly correlated with peak velocity) showed similar yield in loci (**see Supplementary Results**). While 93 MTAG loci were overlapping between at least one valve measurement GWAS and one AS GWAS with $P < 5 \times 10^{-08}$ for both, the remaining loci had still had a $P < 5 \times 10^{-5}$ (**Figure 4**).

3) Lines 144-145, the sentence on overlap with CAD is interesting but needs some context (also see major comment 5 above)

Author response:

Thanks, we have now expanded the supplementary section of the overlap between CAD loci and AS. We also now provide a paragraph in the discussion to elaborate on our findings on the genetic similarities of CAD and AS. As raised by the Reviewer above, we also performed sensitivity analysis excluding people with a diagnosis of CAD before AS. Here, we observe similar results in UK Biobank and slightly fewer loci in FinnGen at a lower sample size.

Changes to the Manuscript:

The **Results>Common variants related to aortic stenosis** section now states:

Of the 61 loci identified in the analysis of AV traits, 19 loci were also identified among the 241 loci associated with coronary artery disease in Aragam, et al(Aragam et al. 2022)(**see Supplemental Notes**).

The **Results>Genetic discovery of 134 aortic stenosis loci after MTAG** section now states:

Of the 134 loci identified in the MTAG-augmented analysis of AS, 46 loci were also identified among the 241 loci associated with coronary artery disease in Aragam, et al(Aragam et al. 2022)(see Supplemental Notes).

The **Discussion** section now states:

The additional 40+ loci that overlap with coronary disease suggest that there is a substantial amount of additional biology to be untangled through the joint analysis of AS and coronary disease. This overlap was also evident in the GWAS of continuous aortic valve traits with almost 30% of the 61 loci also reported in a GWAS of coronary artery disease(Aragam et al. 2022). Importantly, our findings underscore that these effects are evident in valvular function well before the onset of clinical AS. Critically, the imaging traits were obtained for research purposes, and so their genetic link to coronary artery disease is not influenced by imaging that was obtained due to coronary artery disease (i.e., confounding by indication). In a sensitivity analysis of the AS diagnosis-based GWAS that excluded those with a diagnosis of coronary artery disease occurring before their AS diagnosis, our observations remained largely consistent which suggests that ascertainment bias might not fully explain our findings. Further, these observations provide a genetic foundation for the notion that subclinical variation in aortic valve function may have utility as an early marker of atherosclerosis.

The **Supplementary Results** now show:

Overlap between coronary disease and aortic valve loci

Out of the 61 loci identified in the analysis of aortic valve traits, 19 were among those loci identified in a recent GWAS of coronary artery disease by Aragam, et al (Supplementary Table 3 in Aragam, et al.)(Aragam et al. 2022). These loci included ANP32E, OTUD7B, ZEB2, MAD2L1, FER, HMGA1, LPA, FERD3L, SURF6, PDE3A, ATXN2, TBX5, SMAD3, TMEM170A, GOSR2, CTAGE1, ADAMTS10, SLC44A2 and KLF2.

Similarly, out of the 134 loci identified in the MTAG-augmented analysis, 46 were also identified among the 241 loci associated with coronary artery disease in Aragam, et al(Aragam et al. 2022). These included loci near PRDM16, PCSK9, FGGY, PSRC1/CELSR2, MTMR11/OTUD7B, IL6R, ZEB2/TEX41, KALRN/UMPS, STAG1/PPP2R3A, ARHGEF26, LNX1/PDGFRA, MAD2L1, FER, SNX2, MICA/NOTCH4, LPA, FERD3L/HDAC9, TBX20, MET/CFTR, AOC1/NOS3, LPL, NSMCE2/TRIB1, TRAF1/C5, MYMK/ABO, JCAD, SUFU/CNNM2, PDE3A, CEP83/FGD6, PPP1CC/ATXN2, HNF1A, SCARB1, LIPC, SMAD3, CHRN4/ADAMTS7, FES, TMEM170A/CFDP1, SMG6/RAP1GAP2, RPRML/GOSR3, CTAGE1, ANGPTL4, LDLR, KLF2/MYO9B, SHKBP1, APOE, ITCH/NCOA6, HORMAD2/OSM. This list represents 34% of the MTAG-augmented loci for aortic valve measurements or aortic stenosis, and 19% of all coronary disease loci discovered to date.

The **Supplementary Results** section now states:

Sensitivity analysis using different definitions of aortic stenosis cases

To address the potential influence of the AS definition, we have now performed sensitivity analyses in both UK Biobank and FinnGen by using an AS definition that mandates an ICD code for AS but also a procedural code for aortic valve surgery. In this surgery + ICD code definition (N cases = 2,053), we observe the same 4 loci in UK Biobank that we observe when only using the ICD code for AS (N cases = 3,413). In FinnGen, we observe 4 loci compared with 34 using the main analysis which is explained by the large drop in cases from 12,398 to 2,833 using the more strict definition. These four loci are replicating 3 (PALMD, ZEB3 and LPA) observed in the UK Biobank analysis (**Supplementary Table 3**).

Additionally, we investigated whether indication bias for ascertainment of AS diagnoses used for the case/control meta-analysis could majorly influence our results. To this extent, we have now performed GWAS sensitivity analyses in UK Biobank and FinnGen excluding AS cases that were preceded by a diagnosis of CAD. In UK Biobank (N cases = 3,413), we retain the same 4 loci as in the main analysis. In FinnGen, we observed 23 loci (N cases = 9,584) when excluding those with CAD before AS diagnosis compared with 34 loci (N cases = 12,398) using the main definition (**Supplementary Table 3**).

Supplementary Table 3 now shows:

Table 3: Sensitivity analyses in UK Biobank and FinnGen

Cohort	UK Biobank, all	UK Biobank, No CAD	UK Biobank, Surgical AS	FinnGen, all	FinnGen, No CAD	FinnGen, Surgical AS
N Cases	5,038	3,413	2,053	12,398	9,584	2,833
N Controls	412,301	398,814	414,743	487,930	487,930	487,930
N Lead SNPs	4	4	4	34	23	4
Nearest Genes	PALMD, ZEB2, LPA, IL6	PALMD, ZEB2, LPA, IL6/STEAP1B	PALMD, ZEB2, LPA, IL6	LMO4, PALMD, ADAMTSL4, PRRX1, NAV1, DSTYK, RNF144A, ZEB2, TMEM44, AC058822.1, FER, SNCAIP, TSBP1, NMBR, LPA, IL6, FERD3L, FADS1/FADS2, MYEOV, FGF23, PDE3A, PLXNC1, SMAD9, STARD9, SPG11, CHRN4, GLIS2, CFDP1, MEOX1, CA10, BAHCC1, TSPAN16, HORMAD2, RPS6KA3	ALPL, PALMD, PRRX1, NAV1, RNF144A, ZEB2, TMEM44, AC058822.1/PDGFRA, FER, TSBP1, NMBR, LPA, DGKB, IL6, FADS1/FADS2, MYEOV, PDE32, STARD9, TRIM69, BCAR1, MEOX1, BAHCC1, TSPAN16	PALMD, RNF144A, ZEB2, LPA

4) Line 148-150, a more complete description of prior AS loci should be provided here. How many AS loci were identified in previous studies? Of those, how many are validated in the present study? A supplementary table can be provided.

Author response:

Thanks, we have provided a supplementary table of prior AS loci reported in literature in **Supplementary Table G**. Compared with the 23 loci described by Chen et al. and Small et al., we recapture 20 loci in our disease based meta-analysis before MTAG. After MTAG, we capture all but one locus near HMGB1 from Chen et al. However, since both Chen et al. and Small et al. are used in our meta-analysis, this is not the ideal way to look at this. Fortunately for us, Theriault et al. published a new meta-analysis of calcific AS (including Bicuspid AS) that has some overlap with our meta-analysis (for instance the UK Biobank and FinnGen) but also includes independent studies. They report 32 loci at genome-wide significance. Using our deep-learning derived continuous traits, we replicate 9 loci in the GWAS of at least one of the three traits (ACTR2, ZEB2, LPA, FERD3L, SURF6/MYMK, PDE3A, HMGA2, TMEM170A, SLC44A2/LDLR)(Theriault et al. 2024). Using our disease-based meta-analysis before MTAG,

we replicate 29 out of the 32 loci reported by Theriault et al. Detailed comparisons are provided in **Supplementary Table G**.

Changes to the Manuscript:

Detailed comparisons of our findings with previous studies are now provided in **Supplementary Table G**.

The **Results>Common variants related to aortic stenosis** section now states:

For this disease-based GWAS meta-analysis, we observed 92 loci at genome-wide significance, including the 20 out of 23 previously described loci for calcific AS (**Table 2, Supplementary Table D**). We also confirmed 29 out of 32 loci from the meta-analysis by Theriault et al. (Theriault et al. 2024) which was not included in our meta-analysis before MTAG for genetic discovery (but has a partial sample overlap) (**Supplementary Table G**).

The **Results>Genetic discovery of 134 aortic stenosis loci after MTAG** section now states:

Among the 134 independent loci for AS (**Supplementary Table F**), we reidentified all out of the 23 previously reported loci for calcific AS with the sole exception of a locus near HMGB1 from Chen et al (Yu Chen et al. 2023) and 29 out of the 32 loci reported by Theriault et al (Theriault et al. 2024) (**Supplementary Table G**)

5) Line 159, what is the relevance of **CHRNA4** association with tobacco?

Author response:

This is a good point. We wanted to highlight a possible link to a risk factor of vascular disease (i.e. smoking) that might play a role in pathogenesis of aortic valve disease. The *CHRNA4* locus was notable because it harbors a conserved gene cluster encoding the beta subunits of the nicotinic acetylcholine receptor; common and rare variants in this gene have been associated with tobacco use. While we still observe this locus, based on the Reviewer's feedback we have sought to address the point more robustly with an expanded Mendelian Randomization analysis that now includes other risk factors such as high blood pressure, blood glucose but also smoking and drinking behaviors. In this more robust way to assess the association of risk factors and AS risk, we have shifted our discussion more towards high blood pressure, diabetic conditions and body weight and therefore removed the sentence to highlight *CHRNA4*.

Changes to the Manuscript:

The **Results** section now states:

Genetic evidence for blood pressure and diabetic conditions in aortic valve disease

Observational studies have linked common cardiovascular risk factors with AS risk (Yan et al. 2017). We applied Mendelian randomization to seek causal evidence for these common risk factors on aortic valve dysfunction. The outcome GWAS for AS was the METAL meta-analysis

excluding UK Biobank and no MTAG to avoid a substantial sample overlap with exposure GWAS.

We observed that 1 SD of increase in SBP was associated with higher peak velocity ($\beta=0.26$, $P=1.1 \times 10^{-15}$) and greater risk for AS (OR 1.28, $P=8.6 \times 10^{-12}$) (**Supplementary Table K-L**). A similar observation was seen for BMI with higher mean gradient ($\beta=0.16$, $P=3.4 \times 10^{-19}$) and risk for AS (OR 1.17, $P=9.1 \times 10^{-18}$). A 486-variant genetic instrument for hemoglobin A1c—a marker of diabetes risk—was associated with a smaller AVA ($\beta=-0.08$, $P=1.8 \times 10^{-07}$) and greater AS risk (OR 1.04, $P=6.0 \times 10^{-03}$), although this association was not statistically significant in the MR Egger or weighted median analyses ($P=0.72$ and 0.49 , respectively, **Supplementary Table K-L**).

6) Line 160, all effect sizes ≥ 0.1 . Effect size in which trait?

Author response:

Thanks, this was initially used to refer to any trait. However, this way of describing our results was confusing, so we removed that sentence and now provide more data about the loci in detail in the main manuscript as well as the supplementary notes.

Changes to the Manuscript:

The **Results** section now states:

Common variants related to aortic valve function

To understand the genetic basis of aortic valve function from common variation, we conducted GWAS of 59,639 participants using REGENIEv2.2.4. For all three aortic valve traits, we observed a total of 90 loci (43 for AVA, 27 for peak velocity and 20 for mean gradient), of which 61 unique loci were associated with at least one trait at a commonly used significance threshold of $P < 5 \times 10^{-8}$ (**Supplementary Table D and Supplementary Figure 3**). Shared at genome-wide significance across all three traits were genetic loci near LPA, PDE3A, HMGA2, CDK8, KCNRG/DLEU1, GOSR2, CTAGE1 and MN1. Loci near DLEU1, HMGA2 and GOSR2 have previously been linked to the diameter of the aortic root (Nekoui et al. 2022). A locus near CTAGE1 has been previously described for AS as well as abdominal aortic aneurysm (Yu Chen et al. 2023; Klarin et al. 2020), while LPA is a well-known gene associated with AS in previous GWAS (Yu Chen et al. 2023; Small et al. 2023). PDE3A and CDK8 have all been previously linked to cardiovascular disease or physiology (Ercu et al. 2022; Monzen et al. 2008; Hall et al. 2017), although not aortic valve disease specifically, whereas the link between MN1 and cardiovascular disease is largely unexplored. We also observed two signals on the X chromosome, in loci near NDP (for the gradient-based measures) and near TSPAN6 (for AVA), both of which have not been described for aortic or aortic valve disease before.

In a sensitivity analysis (see **Supplementary Notes**) excluding participants with aortic valve measurements consistent with moderate (341 participants) or severe (46 participants) AS, the number of associated loci increased from 61 to 64, including a loss of four loci near OR5D13, ELN, TMEM170A and PSTPIP1, and a gain of seven loci (near ANK2, MAD2L1, MYBPC3, CSH2, PLAU, TBC1D12, PIGU), all of which were within one order of magnitude of the $P < 5 \times 10^{-8}$ significance threshold in the main analysis (**Supplementary Table E**).

In total, 6 out of 23 loci in the recent GWAS by Chen et al. (Yu Chen et al. 2023) and Small et al. (Small et al. 2023) previously associated with AS (LPA, ACTR2, CTAGE1, TEX41, TMEM170A, APLP/WNT4) were reidentified here in association with peak velocity, mean gradient or AVA, while 1 locus for BAV specifically (GATA4) was re-identified. Theriault et al. published a meta-analysis of calcific AS which did include BAV cases and reported 32 loci at genome-wide significance, out of which 9 were replicated in at least one of the deep-learning derived continuous traits (ACTR2, ZEB2, LPA, FERD3L, SURF6/MYMK, PDE3A, HMGA2, TMEM170A, SLC44A2/LDLR) (Theriault et al. 2024). The full GWAS findings are represented in detail in the appendix (**see Supplementary results**).

Common variants related to aortic stenosis

We conducted a meta-analysis of disease-based GWAS of AS using METAL, including data from Chen et al. (13,765 cases; 640,102 controls) (Yu Chen et al. 2023), from Small et al. (14,451 cases; 398,544 controls) (Small et al. 2023), from all UK Biobank participants who did not undergo cMRI (5,038 cases; 412,301 controls), and from the freeze 12 of FinnGen (12,398 cases; 487,930 controls). The Chen, et al, and UK Biobank analyses had sample overlap, which was accounted for in our analyses. The baseline characteristics of all biobank participants included are provided in **Supplementary Tables 1-2**. For this disease-based GWAS meta-analysis, we observed 92 loci at genome-wide significance, including the 20 out of 23 previously described loci for calcific AS (Table 2, Supplementary Table D). We also confirmed 29 out of 32 loci from the meta-analysis by Theriault et al. (Theriault et al. 2024) which was not included in our meta-analysis before MTAG for genetic discovery (but has a partial sample overlap) (**Supplementary Table G**).

Additionally, we observed newly associated loci near LDLR ($\beta=0.031$, $P=1.2 \times 10^{-13}$) and a missense variant in PCSK9 ($\beta=0.089$, $P=2.4 \times 10^{-11}$), both of which are loci harboring genes that regulate blood levels of apolipoprotein B-rich lipids through the LDL receptor. Furthermore, in addition to the previously implicated genes in lipid metabolism and AS (FADS2/MYRF [$\beta=0.047$, $P=1.2 \times 10^{-32}$] and SORT1/PSRC1 [$\beta=-0.033$, $P=1.6 \times 10^{-20}$]), we identified variants near SORL1 ($\beta=-0.024$, $P=4.8 \times 10^{-08}$) which encodes the sortilin-related receptor (Rogaeva et al. 2007). SORL1 is a receptor for ApoE but has also been linked to IL6 signaling (Carlo 2013; Larsen and Petersen 2017). We additionally observed several other loci associated with lipid metabolism such as ALPL, SCARB1, LPL and ANGPTL4. Both latter genes are important in triglyceride metabolism. Lipoprotein lipase is encoded by LPL and involved in the lipolysis of triglycerides in lipoproteins (Li et al. 2014). One of the key regulators of LPL is angiopoietin-like protein 4 (ANGPTL4) that inhibits lipoprotein lipase similarly to ANGPTL3 (Musunuru et al. 2010; Köster et al. 2005).

Among the previously described and replicated loci for AS is IL6 ($\beta=0.051$, $P=9.2 \times 10^{-45}$). In addition to this well-known locus, we also observed a variant near IL6R ($\beta=0.025$, $P=7.6 \times 10^{-11}$), encoding the IL6 receptor.

We additionally observed loci with key regulators of phosphate homeostasis, such as the previously reported ALPL ($\beta=0.031$, $P=4.2 \times 10^{-18}$) and, here for the first time observed in connection to AS, FGF23 ($\beta=-0.028$, $P=3.2 \times 10^{-13}$) (ADHR Consortium 2000).

We observed similar results in UK Biobank and FinnGen when a) limiting our definition of AS cases to those who also had a procedure code for aortic valve intervention or b) excluding those

with a diagnosis of coronary artery disease before AS to account for a possible ascertainment bias (**Supplementary Table 3**).

Of the 61 loci identified in the analysis of AV traits, 19 loci were also identified among the 241 loci associated with coronary artery disease in Aragam, et al (Aragam et al. 2022) (see **Supplemental Notes**). A PGS based on GWAS of aortic valve traits was predictive of AS in FinnGen. Being in the top 5% for genetically predicted mean gradient led to an HR of 1.44 for AS (95% CI 1.35-1.54, $P=1.1 \times 10^{-27}$) compared to the bottom 95% (see **Supplementary Results**).

Multitrait analysis of GWAS identifies 166 loci for aortic valve function and aortic stenosis

Having established that quantitative aortic valve measurements were heritable biomarkers for aortic valve stenosis, we pursued multi-trait analysis of GWAS (MTAG) to incorporate information from these related traits (Turley et al. 2018). This analysis included the three AV traits and the AS meta-analysis. This yielded an effective sample size of $N=96,385$ for peak velocity, $N=98,645$ for mean gradient, $N=77,183$ for AVA, and $N=205,483$ for AS. The maxFDR, the upper bound for the false discovery rate, was 0.015 for peak velocity, 0.014 for mean gradient, 0.012 for aortic valve area, and 0.003 for AS.

Using this approach, the number of loci associated with at least one of the three aortic valve traits at genome-wide significance increased from 90 to 273 loci overall (90 for AVA, 88 for mean gradient and 95 for peak velocity) and from 61 unique loci to 134 unique loci between all three traits (**Supplementary Table F, Figure 3**) while the number of AS loci increased from 92 in the METAL meta-analysis to 134 post-MTAG (**Figure 4**). When considered altogether, 166 distinct loci were identified across the aortic valve endophenotypes and AS GWASes within the MTAG framework (**Supplementary Table F**). Out of these, 32 loci were associated only with AS but did not reach genome-wide significance for any of the aortic valve endophenotypes, including PCSK9 and IL6R. However, these loci had substantial sub-threshold signals as seen in Table 2. A sensitivity analysis leaving out mean gradient (which is highly correlated with peak velocity) showed similar yield in loci (see **Supplementary Results**).

The **Supplementary Results** section now states:

Peak Velocity and Mean Gradient

Peak velocity and AVA are the main measurements to grade severity of AS with cardiovascular imaging in clinical guidelines for valvular heart disease (Vahanian et al. 2022; Otto et al. 2021). For peak velocity we found a total of 27 loci at genome-wide significance (**Supplementary Table D and Supplementary Figure 3**).

The two loci, near LPA ($P=1.9 \times 10^{-9}$) and GATA4 ($P=3.3 \times 10^{-22}$) have been previously reported in GWAS of aortic stenosis and bicuspid valve disease (Helgadottir et al. 2018; Yang et al. 2017). Bicuspid valve disease is the most common congenital heart valve disorder and affected patients are at markedly higher risk for early-onset AS and aortic root disease (Siu and Silversides 2010; Fazel et al. 2008). Among the loci with prior associations with related

phenotypes, we identified variants near GOSR2 ($\beta= 0.090$, $P=2.9 \times 10^{-28}$), HMGA2 ($\beta= 0.058$, $P=2.4 \times 10^{-16}$) and KCNRG/DLEU1 ($\beta=0.060$, $P=3.7 \times 10^{-17}$) that were also previously reported for planimetric AVA (Córdova-Palomera et al. 2020). Additional findings included one variant near SMAD3 ($P=4.0 \times 10^{-8}$). SMAD3 is associated with familial aortic aneurysm and dissection (Lynch et al. 2011; van de Laar et al. 2011). SMAD3 is also known as a causal gene for the aortic Loeys-Dietz syndrome (MacFarlane et al. 2019). Another observation was the overlap of previously reported loci associated with coronary artery disease and those that we observed for mean gradient and peak velocity. These loci included IMPG1/ Y RNA, LPA, SORL1, SURF6, OTUD7B, CTAGE1, LRRFIP2, ABCC9, TMEM170A and FERD3L/TWIST1 (Aragam et al. 2022; Ma et al. 2022; Takahashi et al. 2010). No participants in the GWAS had peak velocity greater than the threshold to define severe aortic stenosis (4 m/s), indicating that the common variant findings were driven by normal variation in the population and not an ascertainment bias.

For the highly correlated mean gradient, we found 20 loci at genome-wide significance with 17/20 loci overlapping with peak velocity and the remaining 3 loci with sub-threshold significance. These loci for mean gradient included ABCC9, TMEM170A and ELN. Elastin haploinsufficiency has been previously associated with aortic valve malformations (Hinton et al. 2010).

Aortic Valve Area

In the GWAS for AVA we identified 43 loci at genome-wide significance including all three previously reported loci (CRADD, KCNRG/DLEU1, GOSR2) for AVA (Córdova-Palomera et al. 2020) (**Supplementary Table D and Supplementary Figure 3**). Among these loci we observed 2 loci associated with As previously, ACTR2 ($P=7.4 \times 10^{-10}$) and LPA. Rs55730499, near LPA—encoding lipoprotein (a) [Lp (a)]—was associated with AVA ($P=7.1 \times 10^{-15}$) with one of the biggest effect sizes ($\beta=0.082$) we observed for AVA. This variant was reported in previous GWAS of Lp (a) and is in perfect LD with rs10455872 that explains ~25% of variance of circulating Lp (a) levels (Wei et al. 2018).

Among genome-wide significant variants we found a variant near SMAD3 ($P=7.0 \times 10^{-9}$) but also rs1840828 near KLF2 ($P=7.4 \times 10^{-11}$) which encodes a transcription factor that has been shown to regulate hemodynamic fluid response and vascular development (J. S. Lee et al. 2006; Rasouli et al. 2018). Similarly, one loci near PKN2 ($P=8.7 \times 10^{-10}$) was observed, which encodes for protein kinase N2 that is activated by flow through the mechanoreceptor Piezo1 to regulate vascular tone (Jin et al. 2021).

Among the 31 loci we observed for AVA that were not seen for the gradient based measures, we observed further loci that were previously associated with CAD or ischemic heart disease such as EDEM3, MAD2L1, FER, HMGA1, ATXN2 or SLC44A2 (Gan et al. 2022; Erdmann et al. 2018; Hartiala et al. 2021). Additional loci among those 31 loci were associated with aortic disease or stiffness (TGFB2, CHSY1) (Lindsay et al. 2012; Pirruccello et al. 2023), cardiac fibrosis (COL8A1) (Skrbic et al. 2015), cardiac remodeling (CCND2, AKAP13) (Zhu et al. 2018; Zakhary, Moravec, and Bond 2000) or cardiac contractility (BAG3, IGFR1) (Algül et al. 2023; Franaszczyk et al. 2014). The full list of loci is provided in **Supplementary Table D** and visualized in **Supplementary Figure 3**.

7) Line 163, the sentence starting with ERG seems out of context?

Author response:

Thanks, this was indeed a clunky sentence. We have now provided more context to our loci that have been linked to valvular pathology.

Changes to the Manuscript:

The **Results>Genetic discovery of 134 aortic stenosis loci after MTAG** now states:

Interestingly, among the 53 loci that reached genome-wide significance after MTAG, we also discovered variants near genes more closely related to valve pathology. For instance, both ERG ($\beta=-0.107$, $P=4.3 \times 10^{-09}$)(Vijayaraj et al. 2012) and VGLL4 ($\beta=-0.019$, $P=2.0 \times 10^{-08}$)(Yu et al. 2019) were previously associated with valve morphogenesis.

8) Add titles and legend/abbreviations to the supplementary tables

Author response:

Thanks, this is a good point. We have now provided a manifest sheet in the excel files that provide a title for each table as well as a list of abbreviations.

Changes to the Manuscript:

We added a manifest sheet in the excel files.

Reviewer #2:

Remarks to the Author:

Summary

This paper reports new loci for aortic stenosis (AS) risk. The authors used deep learning to measure peak velocity, mean gradient and aortic valve area from cardiac magnetic resonance (CMR) imaging measurements from participants in the imaging study of UK Biobank (UKB). GWAS was performed on these traits, and then multi-trait analysis of GWAS (MTAG) was used to incorporate data from AV measures and AS diagnosis from both UKB and FinnGen. Polygenic scores were associated with AS and MR was used to test for causal relationships of lipid measures with interesting results from Lp(a) and LDL on AV function and elevated phosphate on AS.

Comments to Authors

This is an interesting paper with discovery of new loci contributing to AV function and AS risk, with nice downstream analyses using the new results for reporting potentially causal relationship. I have a few comments for the authors to consider:

Author response:

We thank the Reviewer for the interest in our study and the appreciation of our analyses and the findings.

• **Abstract – you are reporting a lot of results from all analyses performed in your paper, I suggest some additional information on the methods and datasets for the PRS and MR work and motivation. You could reduce the text on gene reporting and processes as it is not clear which genes map to which trait in the current reporting**

Author response:

Thanks, we have now expanded upon the methods and datasets used in the study and cut back on the reporting of results.

Changes to the Manuscript:

The new **Abstract** now reads:

The genetic influences on normal aortic valve (AV) function and their impact on aortic stenosis (AS) risk are of significant interest. We sought to identify common genetic determinants of normal variation in AV function, and to understand their relationship with clinical disease. We used deep learning to measure peak velocity, mean gradient, and aortic valve area from magnetic resonance imaging (MRI) and conducted genome-wide association studies (GWAS) in 59,751 participants in UK Biobank participants. Additionally, we performed a meta-analysis of disease-based AS GWAS from UK Biobank and FinnGen as well as published summary statistics. We then performed multi-trait analysis of GWAS (MTAG) to incorporate data from AV measurements and AS GWAS, and applied PRSs to construct polygenic scores (PGS). These

PGS were tested in FinnGen before MTAG and in All of Us and Mass General Brigham Biobank (MGBB) after MTAG.

Before MTAG, we observed 61 loci associated with at least one AV measurement and 92 loci associated with AS from the disease-based GWAS. The PGS for mean gradient was predictive of AS in FinnGen (HR 1.44 for the top 5% of participants vs all others, $P=1.1 \times 10^{-27}$).

Incorporating the AV measurement GWAS with AS GWAS using MTAG, we identified 166 distinct loci (134 with AV traits, 134 with AS, and 166 unique loci across all three AV traits and the AS GWAS), including PCSK9 and LDLR. Other loci were associated with lipid metabolism, atherosclerosis, inflammation, but also phosphate hemostasis (FGF23). The PGS for AS after MTAG was associated with AS in All of Us (HR 3.32 for top 5%, $P=8.8 \times 10^{-22}$) and MGBB (HR 2.76 for top 5%, $P=7.8 \times 10^{-15}$). Tissue enrichment showed higher expression of genes in cardiac and aortic tissue. Due to the observation of loci associated with risk factors, we performed Mendelian randomization analysis to examine the causal association of common risk factors with aortic stenosis. We found evidence supporting a potential causal role for Lp(a) and LDL—but not ApoA—on AV function. Elevated phosphate, systolic blood pressure and body mass index were among the risk factors with evidence for a potential causal association with AS. Using deep-learning derived AV measurements, we identified 166 genetic loci linked to AV function or AS. These findings have implications for the early pathogenesis of AS and suggest modifiable pathways as targets for preventive therapy.

• Introduction – you could expand the literature quoted for quantitative phenotypes?

Author response:

Thanks, we have now expanded the literature quoted in the introduction.

Changes to the Manuscript:

The **Introduction** section now states:

Analysis of quantitative endophenotypes for disease—such as aortic diameter for aortic aneurysm(Pirruccello, Chaffin, et al. 2022; Guo et al. 2016) or left ventricular ejection fraction for heart failure(Pirruccello, Di Achille, et al. 2022; Tadros et al. 2021)— is a powerful approach for genetic discovery in healthy study populations. For example, a 2023 genome-wide association study (GWAS) of thoracic aortic aneurysm and dissections with 8,626 cases and 453,043 controls reported 21 genetic loci associated with the phenotype(Klarin et al. 2023). However, a 2022 GWAS of ascending aortic diameter in only 38,694 participants in the UK Biobank found 82 loci, including 19 out of the 21 loci later identified in the 2023 disease-based GWAS(Pirruccello, Chaffin, et al. 2022), emphasizing the sample efficiency of quantitative endophenotypes for genetic discovery.

For AS, endophenotypes include the principal measurements used to assess the severity of disease: aortic valve area (AVA), mean gradient, and peak velocity(Baumgartner et al. 2017).

• Results _ Figure 1 – suggest as 6 sub figures, they are labelled as a-f, or prepare as one flow figure?

Author response:

Thanks, we updated Figure 1 to reflect our expanded analyses. We also have labeled them as six subfigures from a-f.

Changes to the Manuscript:

Updated **Figure 1** is now:

Figure 1: Study overview

(a) Deep learning was used to create segmentation masks of the ascending aorta in the aortic flow image series in the UK Biobank. The masks were used to create velocity maps that were (b) used to construct 3 MRI-derived phenotypes (aortic valve area, mean gradient, peak velocity). The orange colour denotes female participants and the blue colour male participants. (c) The traits were then used to conduct genomewide-association studies (GWAS) of common variants. The summary statistics were further used to create MTAG of aortic valve area, mean gradient and peak velocity with a disease-based aortic stenosis GWAS. (d) Gene-set enrichment and analyses for tissue enrichment as well as cell-specific analysis for the thoracic aorta and left ventricle were performed. (e) Further analyses were performed using Mendelian randomization and polygenic scores of lipid-based traits and other common risk factors with the MRI-derived traits. Additionally, polygenic risk scores were constructed for each phenotype. Polygenic scores from GWAS before MTAG were used to predict incident aortic stenosis in the external FinnGen cohort. (f) MTAG-adjusted scores were applied to the external *All of Us* cohort and the MGB Biobank. MRI images are reproduced by kind permission of UK Biobank ©. Medical images were used from Servier Medical Art under Creative Commons-BY 4.0 license.

• Results - The genetic correlation between mean gradient and peak velocity was 0.986, why did you perform a GWAS for each trait, what was the overlap in findings

Author response:

This is an important point. The Reviewer rightfully highlights the high genetic (and raw) correlation between both mean gradient and peak velocity. The motivation for conducting GWAS for all three measures of aortic valve function (including aortic valve area) stems from the clinical practice of using all three measures to grade aortic valve function. The relationship between mean gradient and peak velocity however depends on the shape of the velocity curve, which can vary based on stenosis severity and flow rate (Baumgartner et al. 2017). The consensus document on the grading of aortic stenosis severity by European (European Association of Cardiovascular Imaging) and American (American Society of Echocardiography) professional societies therefore recommend the reporting of both measures (Baumgartner et al. 2017). Due to this, we have opted to include both traits to reflect the subtle clinical differences. In other words, we believe this is the clinically relevant analysis plan, even though leaving out either peak velocity or mean gradient has little effect on any meta-analysis results so long as the other remains included.

Changes to the Manuscript:

The **Supplementary Results** section now states:

Sensitivity analysis of MTAG without mean gradient

We chose to include all three measures of aortic valve function (mean gradient, peak velocity and AVA) due to the clinical relevance of using all three measures when grading aortic valve function as recommended in the 2017 EACVI/ASE consensus document on the echocardiographic assessment of aortic stenosis severity (Baumgartner et al. 2017). However, due to the high genetic correlation between both mean gradient and peak velocity ($r=0.99\pm 0.003$), we performed a sensitivity analysis leaving out mean gradient when performing MTAG of disease-based AS meta-analysis and functional measures of aortic valve function. Compared with the main analysis, this sensitivity analysis yielded an effective sample size of $N=96,365$ for peak velocity (vs $N=96,385$), $N=77,007$ for AVA (vs $N=77,183$) and $N=205,426$ for AS (vs $205,483$). The difference in lead SNPs was marginal; for peak velocity MSL2 was lost, for AVA CACNA1H and LCORL were gained with no losses, and for AS, CAMC2KG and EIF3A were gained with no losses. All these loci were near the significance threshold of $P=5E-08$ in the main analysis.

• Results – you indicate 23 loci you found for CMR valve traits were not in a prior GWAS of AS, what were the p values in a look up, was there some statistical support?

Author response:

Given the changes to our analysis noted above and the larger MRI sample size now available, our findings have changed as follows: we now identify a total of 61 unique loci across all deep-learning derived continuous traits before multi-trait analysis. Crucially, we re-capture many loci observed in disease-based AS GWAS. In total, 6 out of 23 loci in the recent GWAS by Chen et al. (Yu Chen et al. 2023) and Small et al. (Small et al. 2023) previously associated with AS (*LPA*,

ACTR2, CTAGE1, TEX41, TMEM170A, APLP/WNT4) were reidentified here in association with peak velocity, mean gradient or AVA, while 1 locus for BAV specifically (*GATA4*) was re-identified. Theriault et al. published a meta-analysis of calcific AS which did include BAV cases and reported 32 loci at genome-wide significance, out of which 9 were replicated in at least one of the deep-learning derived continuous traits (*ACTR2, ZEB2, LPA, FERD3L, SURF6/MYMK, PDE3A, HMGA2, TMEM170A, SLC44A2/LDLR*)(Theriault et al. 2024). With respect to the other side of this question (i.e., what were the P values for each of our lead SNPs in the prior AS GWAS when they did not achieve $P < 5 \times 10^{-8}$ in those GWAS), this is an interesting question and we acknowledge that it's not really addressed here; given the limited time, we opted to focus on enlarging the present analyses and incorporating additional studies to maximize statistical power for meta-analysis. All summary statistics are made available which will facilitate future such analyses.

Changes to the Manuscript:

Detailed comparisons of our findings with previous studies are now provided in **Supplementary Table G**.

The **Results>Common variants related to aortic stenosis** section now states:

For this disease-based GWAS meta-analysis, we observed 92 loci at genome-wide significance, including the 20 out of 23 previously described loci for calcific AS (**Table 2, Supplementary Table D**). We also confirmed 29 out of 32 loci from the meta-analysis by Theriault et al.(Theriault et al. 2024) which was not included in our meta-analysis before MTAG for genetic discovery (but has a partial sample overlap) (**Supplementary Table G**).

The **Results>Genetic discovery of 134 aortic stenosis loci after MTAG** section now states:

Among the 134 independent loci for AS (**Supplementary Table F**), we reidentified all out of the 23 previously reported loci for calcific AS with the sole exception of a locus near *HMGB1* from Chen et al(Yu Chen et al. 2023) and 29 out of the 32 loci reported by Theriault et al(Theriault et al. 2024)(**Supplementary Table G**)

• **Results – MTAG – for this analysis your input was 5 traits, I was interested in this approach and the robustness of the results. Two of the CMR phenotypes have a correlation of 0.986 and you also included 2 datasets for the same trait AS. Can you justify this approach and if you performed MTAG with one of the highly correlated CMR trait and a meta-analysis dataset for AS – are the results consistent. The number of loci discovered using this method dramatically increases loci yield so it would be good to have some sensitivity analyses reported as the subsequent analyses are based on these findings and there is no formal replication.**

Author response:

This is an important consideration in interpreting our data.

First, in our updated analysis, we perform traditional meta-analysis of the available data from two previously published meta-analyses on aortic stenosis (Small et al. and Chen et al.), as well as FinnGen and the people in the UK Biobank not undergoing MRI using METAL. This gives us one set of summary statistics for aortic stenosis, rather than juggling one per cohort. Then, this output from the disease-based meta-analysis is used to perform MTAG with the three clinical traits used to grade aortic valve function. The resulting maxFDR are exceptionally low with 0.012 for AVA, 0.015 for peak velocity, 0.014 for mean gradient, and 0.003 for aortic stenosis. To try to harmonize the conceptual sample size across binary traits and continuous traits, we use the concept of effective sample size (defined by the METAL authors as 2 times the harmonic mean of cases and controls). The post-MTAG equivalent effective sample size with this approach is N=96,385 for peak velocity, N=98,645 for mean gradient, N=77,183 for AVA, and N=205,483 for AS.

As the Reviewer notes, two traits are very highly correlated (mean gradient and peak velocity). We acknowledge that it is prudent to assess whether choosing to include both peak velocity and mean gradient, instead of just one or the other, made a difference in our analysis. We performed a sensitivity analysis leaving out the summary statistics of mean gradient from our MTAG. Compared with the main analysis, this sensitivity analysis yielded effective sample sizes that were almost exactly the same as the numbers above from the main analysis: N=96,365 for peak velocity (vs N=96,385), N=77,007 for AVA (vs N=77,183) and N=205,426 for AS (vs 205,483). The difference in lead SNPs was trivial; for peak velocity *MSL2* was lost, for AVA *CACNA1H* and *LCORL* were gained with no losses, and for AS, *CAMC2KG* and *EIF3A* were gained with no losses. All these loci were near the significance threshold of $P=5E-08$ in the main analysis (Detailed in the **Supplementary Results**). Nevertheless, because these 3 traits are clinically standard, we continue to include them in our main analyses.

To further verify that our findings are robust, and in response to Reviewer #1, we have performed several sensitivity analyses in UK Biobank and FinnGen; The first sensitivity analysis employs a stricter definition of aortic stenosis that requires both an ICD code for aortic stenosis as well as a procedure code for aortic valve surgery. The second sensitivity analysis excludes people that received a diagnosis of CAD before aortic stenosis diagnosis to account for possible ascertainment bias. The results are detailed further above in responses to Reviewer #1.

Changes to the Manuscript:

The **Results section** now states:

Multitrait analysis of GWAS identifies 166 loci for aortic valve function and aortic stenosis

Having established that quantitative aortic valve measurements were heritable biomarkers for aortic valve stenosis, we pursued multi-trait analysis of GWAS (MTAG) to incorporate information from these related traits (Turley et al. 2018). This analysis included the three AV traits and the AS meta-analysis. This yielded an effective sample size of N=96,385 for peak

velocity, N=98,645 for mean gradient, N=77,183 for AVA, and N=205,483 for AS. The maxFDR, the upper bound for the false discovery rate, was 0.015 for peak velocity, 0.014 for mean gradient, 0.012 for aortic valve area, and 0.003 for AS.

Using this approach, the number of loci associated with at least one of the three aortic valve traits at genome-wide significance increased from 90 to 273 loci overall (90 for AVA, 88 for mean gradient and 95 for peak velocity) and from 61 unique loci to 134 unique loci between all three traits (**Supplemental Table F, Figure 3**) while the number of AS loci increased from 92 in the METAL meta-analysis to 134 post-MTAG (Figure 4). When considered altogether, 166 distinct loci were identified across the aortic valve endophenotypes and AS GWASes within the MTAG framework (**Supplementary Table F**). Out of these, 32 loci were associated only with AS but did not reach genome-wide significance for any of the aortic valve endophenotypes, including PCSK9 and IL6R. However, these loci had substantial sub-threshold signals as seen in **Table 2**. A sensitivity analysis leaving out mean gradient (which is highly correlated with peak velocity) showed similar yield in loci (see **Supplementary Results**).

While 93 MTAG loci were overlapping between at least one valve measurement GWAS and one AS GWAS with $P < 5 \times 10^{-08}$ for both, the remaining loci had still had a $P < 5 \times 10^{-5}$ (**Figure 4**).

The **Discussion>Limitations** section now states:

While MTAG is a tool that has been shown to be robust for genetic discovery, the potential for false-positive results cannot be excluded and is in need for replication using other datasets. Further functional and prospective, clinical validation is warranted before any clinical implementation of programs to manipulate cholesterol or phosphate levels for AS prevention in humans.

The **Methods section** now states:

METAL aortic stenosis GWAS meta-analysis

AS GWAS summary statistics from the UK Biobank GWAS, the FinnGen Data Freeze 12 GWAS (Supplementary Note), the Small 2023 MVP GWAS (Small et al. 2023), and the Chen 2023 GWAS (Yu Chen et al. 2023) were meta-analyzed using METAL v2020-05-05 (Willer, Li, and Abecasis 2010). METAL was run with effective-sample-size weighting and sample-overlap correction because the Chen 2023 GWAS also incorporated UK Biobank data. Because the Chen 2023 GWAS summary statistics did not contain allele frequencies, the ALFA European ancestry allele frequency (from https://www.ncbi.nlm.nih.gov/snp/docs/gsr/alfa/ALFA_20230706150541/) was assigned to each variant. For each cohort, the effective sample size was computed as recommended by the METAL authors: namely, twice the harmonic mean of cases and controls. Variants that were only present in MVP and not found in the other cohorts were removed prior to downstream analyses.

The **Supplementary Results** section now states:

Sensitivity analysis of MTAG without mean gradient

We chose to include all three measures of aortic valve function (mean gradient, peak velocity and AVA) due to the clinical relevance of using all three measures when grading aortic valve function as recommended in the 2017 EACVI/ASE consensus document on the echocardiographic assessment of aortic stenosis severity (Baumgartner et al. 2017). However, due to the high genetic correlation between both mean gradient and peak velocity ($r=0.99\pm 0.003$), we performed a sensitivity analysis leaving out mean gradient when performing MTAG of disease-based AS meta-analysis and functional measures of aortic valve function. Compared with the main analysis, this sensitivity analysis yielded an effective sample size of $N=96,365$ for peak velocity (vs $N=96,385$), $N=77,007$ for AVA (vs $N=77,183$) and $N=205,426$ for AS (vs $205,483$). The difference in lead SNPs was marginal; for peak velocity MSL2 was lost, for AVA CACNA1H and LCORL were gained with no losses, and for AS, CAMC2KG and EIF3A were gained with no losses. All these loci were near the significance threshold of $P=5E-08$ in the main analysis.

• Results – the PRS and MR work is interesting and nicely motivated by the results from the loci you discovered. As mentioned above, the rationale for the lipids and also the serum calcium and phosphate focus should be justified in the abstract as per here.

Author response:

Thanks, we have rephrased the abstract to now clearly state the motivation for the Mendelian Randomization we conducted.

Changes to the Manuscript:

The new **Abstract** now reads:

Due to the observation of loci associated with risk factors, we performed Mendelian randomization analysis to examine the causal association of common risk factors with aortic stenosis. We found evidence supporting a potential causal role for Lp(a) and LDL—but not ApoA—on AV function. Elevated phosphate, systolic blood pressure and body mass index were among the risk factors with evidence for a potential causal association with AS. Using deep-learning derived AV measurements, we identified 166 genetic loci linked to AV function or AS. These findings have implications for the early pathogenesis of AS and suggest modifiable pathways as targets for preventive therapy

• Results – mapping back to abstract again, I suggest reporting of the PRS in the two US cohorts earlier in the abstract and the paper for prediction of AS, this follows better from the AS/endophenotype discoveries before exploration of pathways, genes and MR.

Author response:

Thanks, we have rephrased the abstract to report the PRS results from the two US cohorts earlier now. We also moved the PRS results in the manuscript before the downstream analyses.

Changes to the Manuscript:

The new **Abstract** now reads:

Before MTAG, we observed 61 loci associated with at least one AV measurement and 92 loci associated with AS from the disease-based GWAS. The PGS for mean gradient was predictive of AS in FinnGen (HR 1.44 for the top 5% of participants vs all others, $P=1.1 \times 10^{-27}$). Incorporating the AV measurement GWAS with AS GWAS using MTAG, we identified 166 distinct loci (134 with AV traits, 134 with AS, and 166 unique loci across all three AV traits and the AS GWAS), including PCSK9 and LDLR. Other loci were associated with lipid metabolism, atherosclerosis, inflammation, but also phosphate hemostasis (FGF23). The PGS for AS after MTAG was associated with AS in All of Us (HR 3.32 for top 5%, $P=8.8 \times 10^{-22}$) and MGBB (HR 2.76 for top 5%, $P=7.8 \times 10^{-15}$). Tissue enrichment showed higher expression of genes in cardiac and aortic tissue. Due to the observation of loci associated with risk factors, we performed Mendelian randomization analysis to examine the causal association of common risk factors with aortic stenosis. We found evidence supporting a potential causal role for Lp(a) and LDL—but not ApoA—on AV function. Elevated phosphate, systolic blood pressure and body mass index were among the risk factors with evidence for a potential causal association with AS. Using deep-learning derived AV measurements, we identified 166 genetic loci linked to AV function or AS. These findings have implications for the early pathogenesis of AS and suggest modifiable pathways as targets for preventive therapy.

The **Results** section now states:

Polygenic scores predict aortic stenosis in All of Us and Mass General Brigham Biobank

We used the MTAG-augmented GWAS to produce 1.1-million variant polygenic scores that were tested for association with AS diagnosed after enrollment in the All of Us biobank (**Methods**) using Cox regression. All five polygenic scores were significantly associated with incident AS among 496 incident cases and 243,954 controls. The strongest in terms of effect size was the AS score which had a HR of 1.64 (95% CI 1.50-1.78, $P=8.7 \times 10^{-30}$), while the weakest was the mean gradient score which had a HR of 1.53 (95% CI 1.40-1.66, $P=1.1 \times 10^{-22}$) (**Supplementary Table 5**).

When participants were stratified into the top 5% for each score versus the remaining 95%, the polygenic prediction of mean gradient was strongly associated with AS (HR 2.61, 95% CI 2.00-3.40, $P=1.1 \times 10^{-11}$). Using the polygenic score based on the AS GWAS after MTAG, participants in the top 5% of polygenic risk have a more than three-fold increase in risk for incident AS (HR 3.32, 95% CI 2.60-4.4.24, $P=8.8 \times 10^{-22}$; **Figure 5**)

We further applied the MTAG-derived polygenic scores in the healthcare-based Mass General Brigham (MGB) Biobank with 680 cases and 42,328 controls in Cox proportional hazard models. All four polygenic scores were significantly and directionally consistent in their association with AS (HR 1.56 - 1.61 per SD of polygenic risk score, $P=3.5 \times 10^{-17}$ to 6.3×10^{-36}) (**Supplementary Table 6**). In the stratified analysis comparing the top 5% of participants (bottom 5% for AVA) with the remaining participants, the AS-based score was significantly associated with AS (HR 2.76, $P=7.8 \times 10^{-15}$) as were the functional scores. Both the score for mean gradient (HR 2.79, $P=4.3 \times 10^{-15}$) and the score for peak velocity (HR 3.02, $P=6.3 \times 10^{-19}$) outperformed the AS-based score in terms of HR for incident AS.

MAGMA, tissue and cell-type enrichment analyses identify disease-relevant pathways

Statistical gene set prioritization was performed for each phenotype by applying the MAGMA framework to the MTAG-augmented GWAS summary statistics (C. A. de Leeuw et al. 2015; C. de Leeuw, Sey, and Won 2020). Top pathways achieving FDR $P < 0.05$ —and associated with both aortic valve measurements and AS—included those for coronary artery disease, lipid particle composition, and SMAD protein complexes (**Supplementary Figures 4-5, Supplementary Table H**).

Examining gene expression within human tissues using MAGMA and GTEx v8 (C. A. de Leeuw et al. 2015; Lonsdale et al. 2013), we found a significant enrichment of expression of genes identified in the GWAS within three arterial tissues (aorta, tibial artery, and coronary artery), as well as esophageal tissues and nominal enrichment within left ventricular tissue for aortic flow traits but not for AS (**Supplementary Figure 6**). We then assessed cell-type specificity using single-nucleus RNA sequencing (scRNA-seq) data from the left ventricle, the aorta, and the aortic valve.

In an aortic valve scRNA-seq dataset from a hyperlipidemic murine model from Lee, et al, the loci observed in our study were associated with valvular interstitial cells and valvular endothelial cells but not leukocytes (**Supplementary Figure 7**) (S. H. Lee et al. 2022). We also examined cell types in the neighboring tissues (the ascending aorta and the left ventricle). In ascending aortic scRNA-seq data, vascular smooth muscle cells, pericytes, and fibroblast cell types overexpressed genes linked to peak velocity, mean gradient, AVA and AS (**Supplementary Figure 8**) (Pirruccello, Chaffin, et al. 2022). Analysis of scRNA-seq from left ventricular cardiomyocytes revealed significant enrichment of expression of genes identified in all four GWAS within “activated fibroblasts”, “fibroblast I”, “fibroblast II”, as well as other cell populations from Chaffin, et al (Chaffin et al. 2022). The lowest expression and the only cell types not reaching nominal significance in gene expression from all 4 GWAS were “cardiomyocyte III” and “Epicardial” cell populations (**Supplementary Figure 9**).

• Table 1 – title needs to indicate which resource

Author response:

Thanks, we have clarified that this table refers to the UK Biobank cohort with magnetic resonance imaging.

Changes to the Manuscript:

Table 1 now shows:

Table 1: Participant characteristics of the UK Biobank cohort with magnetic resonance imaging

	Women	Men	All
--	-------	-----	-----

N	31159	28412	59571
Age at time of MRI	65.3 (7.65)	66.5 (7.85)	65.9 (7.77)
Height (cm)	163 (6.24)	176 (6.65)	169 (9.24)
Weight (kg)	69.0 (13.2)	83.5 (13.3)	75.9 (15.1)
Systolic blood pressure (mmHg)	139 (20.2)	144 (17.7)	141 (19.2)
Diastolic blood pressure (mmHg)	77.6 (10.1)	80.8 (9.92)	79.1 (10.2)
Body mass index (kg/m ²)	26.1 (4.78)	27.0 (3.91)	26.5 (4.41)
Aortic valve area (cm ²)	2.50 (0.403)	3.16 (0.545)	2.81 (0.578)
Peak velocity (m/sec)	1.16 (0.250)	1.17 (0.289)	1.16 (0.269)
Mean gradient (mmHg)	2.66 (1.23)	2.82 (1.57)	2.73 (1.41)
Moderate aortic stenosis	219 (1 %)	122 (0 %)	341 (1 %)
Severe aortic stenosis	25 (0 %)	21 (0 %)	46 (0 %)

• **Table 2- not readable – would be good as one of the suppl tables, and simplified as a main table – maybe put on 2 pages, it is useful needs chr build, so legend.**

Author response:

Thank you. Unfortunately with the greatly expanded number of loci this table only becomes more unwieldy. We have done our best to try to make it readable (now spanning 3 pages). We think it's a useful reference and hope to keep it but, if the manuscript is ultimately accepted, we will defer to the editors and typesetters about whether it can be spread across multiple pages in a useful way or if it would just be disruptive. If the latter, we can move it to the supplement. We also extended the legend to include the chromosome build and provide the full results for each MTAG result in **Supplementary Table F**.

Changes to the Manuscript:

Table 2 now shows:

SNP	CH	POS	Gene	EA	OA	AVA			Mean gradient			Peak velocity			Aortic stenosis		
						BETA	SE	P	BETA	SE	P	BETA	SE	P	BETA	SE	P
rs11643207	16	75498793	TMEM170A / CFDP1	C	T	0.042	0.005	1.5E-15	-0.045	0.005	8.3E-22	-0.045	0.005	1.0E-21	-0.041	0.004	5.3E-27
rs12926128	16	88572000	ZFPM1	T	C	0.041	0.006	3.2E-11	-0.04	0.005	1.7E-14	-0.041	0.005	8.2E-15	-0.026	0.004	4.1E-13
rs56209296	17	2069179	SMG6	A	G	-0.034	0.007	2.1E-07	0.029	0.006	2.5E-07	0.028	0.006	6.7E-07	0.021	0.004	5.5E-09
rs2165888	17	2886122	RAP1GAP2	G	A	0.023	0.005	1.3E-05	-0.028	0.005	8.1E-10	-0.029	0.005	8.1E-10	-0.024	0.004	9.5E-11
rs4792252	17	12181798	MAP2K4	T	C	-0.02	0.006	4.2E-04	0.023	0.005	3.3E-06	0.025	0.005	6.8E-07	0.019	0.004	3.0E-08
rs732084	17	37834357	PGAP3	A	C	0.016	0.005	2.9E-03	-0.027	0.005	1.6E-08	-0.027	0.005	1.4E-08	-0.018	0.004	1.4E-05
rs10445373	17	41709281	MEOX1	T	A	0.036	0.007	1.6E-07	-0.03	0.006	6.2E-07	-0.031	0.006	5.3E-07	-0.032	0.005	4.2E-10
rs144520830	17	45097802	RPRML / GOSR2 / CDC27	A	G	-0.067	0.007	3.2E-22	0.055	0.007	9.9E-17	0.057	0.007	1.0E-17	0.067	0.008	2.2E-17
rs11870740	17	79389486	BAHCC1	G	A	0.03	0.006	1.2E-06	-0.027	0.006	1.1E-06	-0.028	0.006	6.3E-07	-0.024	0.004	4.2E-09
rs178007	18	20092966	CTAGE1	C	T	-0.043	0.006	2.5E-14	0.034	0.005	2.1E-12	0.035	0.005	1.1E-12	0.031	0.004	9.5E-15
rs116843064	19	8429323	ANGPTL4	G	A	-0.094	0.018	1.6E-07	0.086	0.015	3.0E-08	0.089	0.016	1.3E-08	0.092	0.012	1.1E-13
rs112107114	19	11190074	LDLR	G	A	-0.065	0.008	1.9E-17	0.047	0.007	7.2E-13	0.05	0.007	8.5E-14	0.049	0.006	2.8E-18
rs56086235	19	16426752	KLF2	T	C	0.034	0.006	6.0E-10	-0.029	0.005	2.9E-09	-0.03	0.005	1.8E-09	-0.024	0.004	1.4E-09
rs112009052	19	41099501	SHKBP1	T	A	0.105	0.022	2.6E-06	-0.111	0.02	1.3E-08	-0.111	0.02	2.0E-08	-0.098	0.015	1.5E-10
rs429358	19	45411941	APOE	T	C	0.028	0.007	7.5E-05	-0.035	0.006	9.6E-09	-0.036	0.006	8.2E-09	-0.024	0.004	4.8E-09
rs8101491	19	47642780	SAE1	G	A	-0.022	0.005	1.1E-05	0.02	0.004	3.6E-06	0.021	0.004	1.8E-06	0.018	0.003	3.9E-08
rs73608424	20	7663563	HAO1	G	A	0.031	0.007	1.8E-05	-0.028	0.006	9.3E-06	-0.028	0.006	9.3E-06	-0.036	0.006	3.6E-11
rs2207098	20	11208274	JAG1	A	G	-0.029	0.005	5.4E-09	0.022	0.004	5.4E-07	0.022	0.004	3.1E-07	0.022	0.003	2.2E-10
rs6141743	20	31107399	NOL4L	C	G	0.027	0.006	9.4E-06	-0.031	0.005	8.5E-09	-0.031	0.005	6.4E-09	-0.026	0.004	2.3E-10
rs57495888	20	33092724	ITCH / PIGU	T	C	-0.013	0.006	3.4E-02	0.033	0.005	5.8E-10	0.032	0.005	1.4E-09	0.021	0.004	2.5E-09
rs4809369	20	62470785	ZBTB46	G	A	-0.029	0.005	6.1E-09	0.017	0.004	4.7E-05	0.018	0.004	3.9E-05	0.016	0.003	4.2E-06
rs117870289	21	39983448	ERG	C	T	0.132	0.021	2.2E-10	-0.103	0.018	7.8E-09	-0.105	0.018	6.6E-09	-0.107	0.018	4.3E-09
rs4821116	22	21973319	UBE2L3	C	T	-0.031	0.006	1.6E-06	0.029	0.006	2.0E-07	0.03	0.006	1.4E-07	0.025	0.004	1.5E-08
rs5752639	22	28200176	MN1	A	G	-0.03	0.005	2.3E-08	0.031	0.005	1.4E-10	0.031	0.005	1.0E-10	0.022	0.004	6.4E-09
rs77464740	22	30602105	HORMAD2 / MTMR3C	T	T	0.066	0.016	3.6E-05	-0.087	0.014	3.5E-10	-0.086	0.014	8.1E-10	-0.073	0.011	4.5E-12
rs1006139	22	38118805	TRIOBP	A	C	0.032	0.005	2.9E-10	-0.03	0.004	6.6E-12	-0.031	0.004	2.6E-12	-0.027	0.003	2.2E-15
rs86708	22	41069773	MCHR1	G	A	0.027	0.005	1.7E-07	-0.021	0.005	2.2E-06	-0.022	0.005	1.9E-06	-0.021	0.004	1.2E-08
rs5906313	X	43827412	NDP	T	C	-0.031	0.006	1.0E-07	0.039	0.006	1.5E-12	0.041	0.006	4.5E-13	0.04	0.007	1.1E-09
rs1802288	X	99890204	TSPAN6	T	C	-0.049	0.008	2.5E-10	0.028	0.007	1.8E-04	0.03	0.007	6.9E-05	0.038	0.009	1.6E-05

SNP=single nucleotide polymorphism. CH=chromosome. POS=genomic position keyed to GRCh37. EA=effect allele to which the "beta" (effect estimate) is aligned. OA=other allele (non-effect allele). SE=standard deviation. P=P value from MTAG. Note that at the same locus, different traits will often have a different lead variant; however, to permit display, the SNP, chromosome, and position shown here are optimized for aortic stenosis; other traits may have more extreme P values that are not displayed because they are linked to different variants at the locus.

• Figure 3 – this also needs to be branded as MTAG result

Author response:

Thanks, we have clarified so in the title and figure legend.

Changes to the Manuscript:

Figure 3 is now:

Figure 3: GWAS for aortic stenosis after MTAG

Genetic associations for aortic stenosis after Multi-trait analysis of GWAS (MTAG). Loci with $P < 5E-8$ are colored and labeled with the name of the gene in closest proximity to the strongest associated variant. The color is red when the locus was not associated at $P < 5E-8$ with aortic valve measurements and blue otherwise. For visualization purposes, the Manhattan plot is truncated at $-\log_{10}(P)=30$. X-axis: chromosomal position. Y-axis: $-\log_{10}(P)$.

• Methods – multi -ancestry GWAS – is there any heterogeneity across ancestries?

Author response:

Thanks, in this analysis we focused on European ancestry. Due to the lack of large numbers of diverse ancestry in UK Biobank and FinnGen (where we had individual level data), we did not pursue multi-ancestry GWAS.

We realize this is, unfortunately, a serious limitation of our work and have highlighted so in the limitations section.

Changes to the Manuscript:

The **Discussion>Limitations** section now states:

While analyses were conducted across three major biobanks, the vast majority of participants had European ancestries, limiting generalizability to diverse populations.

Minor comments

- **Missing page numbering, please do this as standard.**
- **Suppl. Tables – there are no legends, these need to be added for knowledge of the column headers at least as many abbreviations, the reviewer needs to work these out which can lead to mis-interpretation**

Author response:

Thanks, we have included page numbering. We also now provide a legend for each table in the supplementary table section and apologize for the oversight.

References:

- ADHR Consortium. 2000. "Autosomal Dominant Hypophosphataemic Rickets Is Associated with Mutations in FGF23." *Nature Genetics* 26 (3): 345–48.
- Algül, Sila, Maike Schuldt, Emmy Manders, Valentijn Jansen, Saskia Schlossarek, Richard de Goeij-de Haas, Alex A. Henneman, et al. 2023. "EGFR/IGF1R Signaling Modulates Relaxation in Hypertrophic Cardiomyopathy." *Circulation Research* 133 (5): 387–99.
- Aragam, Krishna G., Tao Jiang, Anuj Goel, Stavroula Kanoni, Brooke N. Wolford, Deepak S. Atri, Elle M. Weeks, et al. 2022. "Discovery and Systematic Characterization of Risk Variants and Genes for Coronary Artery Disease in over a Million Participants." *Nature Genetics* 54 (12): 1803–15.
- Baumgartner, Helmut, Chair, Judy Hung Co-Chair, Javier Bermejo, John B. Chambers, Thor Edvardsen, Steven Goldstein, Patrizio Lancellotti, Melissa LeFevre, Fletcher Miller Jr, and Catherine M. Otto. 2017. "Recommendations on the Echocardiographic Assessment of Aortic Valve Stenosis: A Focused Update from the European Association of Cardiovascular Imaging and the American Society of Echocardiography." *European Heart Journal Cardiovascular Imaging* 18 (3): 254–75.
- Bycroft, Clare, Colin Freeman, Desislava Petkova, Gavin Band, Lloyd T. Elliott, Kevin Sharp, Allan Motyer, et al. 2018. "The UK Biobank Resource with Deep Phenotyping and Genomic Data." *Nature* 562 (7726): 203–9.
- Carlo, Anne-Sophie. 2013. "Sortilin, a Novel APOE Receptor Implicated in Alzheimer Disease." *Prion* 7 (5): 378–82.
- Chaffin, Mark, Irinna Papangeli, Bridget Simonson, Amer-Denis Akkad, Matthew C. Hill, Alessandro Arduini, Stephen J. Fleming, et al. 2022. "Single-Nucleus Profiling of Human Dilated and Hypertrophic Cardiomyopathy." *Nature* 608 (7921): 174–80.
- Córdova-Palomera, Aldo, Catherine Tcheandjieu, Jason A. Fries, Paroma Varma, Vincent S. Chen, Madalina Fiterau, Ke Xiao, et al. 2020. "Cardiac Imaging of Aortic Valve Area from 34 287 UK Biobank Participants Reveals Novel Genetic Associations and Shared Genetic Comorbidity with Multiple Disease Phenotypes." *Circulation. Genomic and Precision Medicine* 13 (6): e003014.
- Ercu, Maria, Michael B. Mücke, Tamara Pallien, Lajos Markó, Anastasiia Sholokh, Carolin Schächterle, Atakan Aydin, et al. 2022. "Mutant Phosphodiesterase 3A Protects From Hypertension-Induced Cardiac Damage." *Circulation* 146 (23): 1758–78.
- Erdmann, Jeanette, Thorsten Kessler, Loreto Munoz Venegas, and Heribert Schunkert. 2018. "A Decade of Genome-Wide Association Studies for Coronary Artery Disease: The Challenges Ahead." *Cardiovascular Research* 114 (9): 1241–57.
- Fazel, Shafie S., Hari R. Mallidi, Richard S. Lee, Michael P. Sheehan, David Liang, Dominik Fleischman, Robert Herfkens, R. Scott Mitchell, and D. Craig Miller. 2008. "The Aortopathy of Bicuspid Aortic Valve Disease Has Distinctive Patterns and Usually Involves the Transverse Aortic Arch." *The Journal of Thoracic and Cardiovascular Surgery* 135 (4): 901–7, 907.e1–2.
- Franaszczyk, Maria, Zofia T. Bilinska, Małgorzata Sobieszczańska-Małek, Ewa Michalak, Justyna Sleszycka, Agnieszka Sioma, Łukasz A. Małek, et al. 2014. "The BAG3 Gene Variants in Polish Patients with Dilated Cardiomyopathy: Four Novel Mutations and a Genotype-Phenotype Correlation." *Journal of Translational Medicine* 12 (July): 192.
- Gan, Lu, Demin Liu, Dina Xie, Wayne Bond Lau, Jing Liu, Theodore A. Christopher, Bernard Lopez, et al. 2022. "Ischemic Heart-Derived Small Extracellular Vesicles Impair Adipocyte Function." *Circulation Research* 130 (1): 48–66.
- Guo, Dong-Chuan, Megan L. Grove, Siddharth K. Prakash, Per Eriksson, Ellen M. Hostetler, Scott A. LeMaire, Simon C. Body, et al. 2016. "Genetic Variants in LRP1 and ULK4 Are

- Associated with Acute Aortic Dissections.” *American Journal of Human Genetics* 99 (3): 762–69.
- Hall, Duane D., Jessica M. Ponce, Biyi Chen, Kathryn M. Spitler, Adrienne Alexia, Gavin Y. Oudit, Long-Sheng Song, and Chad E. Grueter. 2017. “Ectopic Expression of Cdk8 Induces Eccentric Hypertrophy and Heart Failure.” *JCI Insight* 2 (15). <https://doi.org/10.1172/jci.insight.92476>.
- Hartiala, Jaana A., Yi Han, Qiong Jia, James R. Hilser, Pin Huang, Janet Gukasyan, William S. Schwartzman, et al. 2021. “Genome-Wide Analysis Identifies Novel Susceptibility Loci for Myocardial Infarction.” *European Heart Journal* 42 (9): 919–33.
- Helgadottir, Anna, Gudmar Thorleifsson, Solveig Gretarsdottir, Olafur A. Stefansson, Vinicius Tragante, Rosa B. Thorolfsson, Ingileif Jonsdottir, et al. 2018. “Genome-Wide Analysis Yields New Loci Associating with Aortic Valve Stenosis.” *Nature Communications* 9 (1): 987.
- Hinton, Robert B., Jennifer Adelman-Brown, Sandra Witt, Varun K. Krishnamurthy, Hanna Osinska, Bhuvaneshwari Sakthivel, Jeanne F. James, et al. 2010. “Elastin Haploinsufficiency Results in Progressive Aortic Valve Malformation and Latent Valve Disease in a Mouse Model.” *Circulation Research* 107 (4): 549–57.
- Horn, Berthold, Berthold Klaus, and Paul Horn. 1986. *Robot Vision*. MIT Press.
- Jin, Young-June, Ramesh Chennupati, Rui Li, Guozheng Liang, Shengpeng Wang, Andrés Iring, Johannes Graumann, Nina Wettschureck, and Stefan Offermanns. 2021. “Protein Kinase N2 Mediates Flow-Induced Endothelial NOS Activation and Vascular Tone Regulation.” *The Journal of Clinical Investigation* 131 (21). <https://doi.org/10.1172/JCI145734>.
- Kirillov, A., E. Mintun, N. Ravi, H. Mao, and C. Rolland. 2023. “Segment Anything.” *arXiv Preprint arXiv*. <https://arxiv.org/abs/2304.02643>.
- Klarin, Derek, Poornima Devineni, Anoop K. Sendamarai, Anthony R. Angueira, Sarah E. Graham, Ying H. Shen, Michael G. Levin, et al. 2023. “Genome-Wide Association Study of Thoracic Aortic Aneurysm and Dissection in the Million Veteran Program.” *Nature Genetics* 55 (7): 1106–15.
- Klarin, Derek, Shefali Setia Verma, Renae Judy, Ozan Dikilitas, Brooke N. Wolford, Ishan Paranjpe, Michael G. Levin, et al. 2020. “Genetic Architecture of Abdominal Aortic Aneurysm in the Million Veteran Program.” *Circulation* 142 (17): 1633–46.
- Köster, Anja, Y. Bernice Chao, Marian Mosior, Amy Ford, Patricia A. Gonzalez-DeWhitt, John E. Hale, Deshan Li, et al. 2005. “Transgenic Angiopoietin-like (angptl)4 Overexpression and Targeted Disruption of angptl4 and angptl3: Regulation of Triglyceride Metabolism.” *Endocrinology* 146 (11): 4943–50.
- Laar, Ingrid M. B. H. van de, Rogier A. Oldenburg, Gerard Pals, Jolien W. Roos-Hesselink, Bianca M. de Graaf, Judith M. A. Verhagen, Yvonne M. Hoedemaekers, et al. 2011. “Mutations in SMAD3 Cause a Syndromic Form of Aortic Aneurysms and Dissections with Early-Onset Osteoarthritis.” *Nature Genetics* 43 (2): 121–26.
- Larsen, Jakob Vejby, and Claus Munck Petersen. 2017. “SorLA in Interleukin-6 Signaling and Turnover.” *Molecular and Cellular Biology* 37 (11). <https://doi.org/10.1128/MCB.00641-16>.
- Lee, John S., Qing Yu, Jordan T. Shin, Eric Sebzda, Cara Bertozzi, Mei Chen, Patti Mericko, et al. 2006. “Klf2 Is an Essential Regulator of Vascular Hemodynamic Forces in Vivo.” *Developmental Cell* 11 (6): 845–57.
- Lee, Seung Hyun, Nayoung Kim, Minkyu Kim, Sang-Ho Woo, Inhee Han, Jisu Park, Kyeongdae Kim, et al. 2022. “Single-Cell Transcriptomics Reveal Cellular Diversity of Aortic Valve and the Immunomodulation by PPAR γ during Hyperlipidemia.” *Nature Communications* 13 (1): 5461.
- Leeuw, Christiaan A. de, Joris M. Mooij, Tom Heskes, and Danielle Posthuma. 2015. “MAGMA: Generalized Gene-Set Analysis of GWAS Data.” *PLoS Computational Biology* 11 (4):

e1004219.

- Leeuw, Christiaan de, Nancy Y. A. Sey, and Hyejung Won. 2020. "A Response to Yurko et Al: H-MAGMA, Inheriting a Shaky Statistical Foundation, Yields Excess False Positives." *bioRxiv*. <https://doi.org/10.1101/2020.09.25.310722>.
- Lindsay, Mark E., Dorien Schepers, Nikhita Ajit Bolar, Jefferson J. Doyle, Elena Gallo, Justyna Fert-Bober, Marlies J. E. Kempers, et al. 2012. "Loss-of-Function Mutations in TGFB2 Cause a Syndromic Presentation of Thoracic Aortic Aneurysm." *Nature Genetics* 44 (8): 922–27.
- Lin, Tsung-Yi, Priya Goyal, Ross Girshick, Kaiming He, and Piotr Dollar. 2020. "Focal Loss for Dense Object Detection." *IEEE Transactions on Pattern Analysis and Machine Intelligence* 42 (2): 318–27.
- Li, Yuan, Ping-Ping He, Da-Wei Zhang, Xi-Long Zheng, Francisco S. Cayabyab, Wei-Dong Yin, and Chao-Ke Tang. 2014. "Lipoprotein Lipase: From Gene to Atherosclerosis." *Atherosclerosis* 237 (2): 597–608.
- Lonsdale, John, Jeffrey Thomas, Mike Salvatore, Rebecca Phillips, Edmund Lo, Saboor Shad, Richard Hasz, et al. 2013. "The Genotype-Tissue Expression (GTEx) Project." *Nature Genetics* 45 (6): 580–85.
- Lynch, Sally Ann, Nicola Foulds, Ann-Charlotte Thuresson, Amanda L. Collins, Göran Annerén, Bernt-Oves Hedberg, Carol A. Delaney, et al. 2011. "The 12q14 Microdeletion Syndrome: Six New Cases Confirming the Role of HMGA2 in Growth." *European Journal of Human Genetics: EJHG* 19 (5): 534–39.
- MacFarlane, Elena Gallo, Sarah J. Parker, Joseph Y. Shin, Benjamin E. Kang, Shira G. Ziegler, Tyler J. Creamer, Rustam Bagirzadeh, et al. 2019. "Lineage-Specific Events Underlie Aortic Root Aneurysm Pathogenesis in Loeys-Dietz Syndrome." *The Journal of Clinical Investigation* 129 (2): 659–75.
- Ma, Lijiang, Nicole S. Bryce, Adam W. Turner, Antonio F. Di Narzo, Karishma Rahman, Yang Xu, Raili Ermel, et al. 2022. "The HDAC9-Associated Risk Locus Promotes Coronary Artery Disease by Governing TWIST1." *PLoS Genetics* 18 (6): e1010261.
- Monzen, Koshiro, Yuzuru Ito, Atsuhiko T. Naito, Hiroki Kasai, Yukio Hiroi, Doubun Hayashi, Ichiro Shiojima, et al. 2008. "A Crucial Role of a High Mobility Group Protein HMGA2 in Cardiogenesis." *Nature Cell Biology* 10 (5): 567–74.
- Musunuru, Kiran, James P. Pirruccello, Ron Do, Gina M. Peloso, Candace Guiducci, Carrie Sougnez, Kiran V. Garimella, et al. 2010. "Exome Sequencing, ANGPTL3 Mutations, and Familial Combined Hypolipidemia." *The New England Journal of Medicine* 363 (23): 2220–27.
- Nekoui, Mahan, James P. Pirruccello, Paolo Di Achille, Seung Hoan Choi, Samuel N. Friedman, Victor Nauffal, Kenney Ng, et al. 2022. "Spatially Distinct Genetic Determinants of Aortic Dimensions Influence Risks of Aneurysm and Stenosis." *Journal of the American College of Cardiology* 80 (5): 486–97.
- Nyul, L. G., J. K. Udupa, and Xuan Zhang. 2000. "New Variants of a Method of MRI Scale Standardization." *IEEE Transactions on Medical Imaging*. <https://doi.org/10.1109/42.836373>.
- Otto, Catherine M., Rick A. Nishimura, Robert O. Bonow, Blase A. Carabello, John P. Erwin 3rd, Federico Gentile, Hani Jneid, et al. 2021. "2020 ACC/AHA Guideline for the Management of Patients With Valvular Heart Disease: A Report of the American College of Cardiology/American Heart Association Joint Committee on Clinical Practice Guidelines." *Journal of the American College of Cardiology* 77 (4): e25–197.
- Petersen, Steffen E., Paul M. Matthews, Jane M. Francis, Matthew D. Robson, Filip Zemrak, Redha Boubertakh, Alistair A. Young, et al. 2015. "UK Biobank's Cardiovascular Magnetic Resonance Protocol." *Journal of Cardiovascular Magnetic Resonance: Official Journal of the Society for Cardiovascular Magnetic Resonance* 18 (1). <https://doi.org/10.1186/s12968->

016-0227-4.

- Pirruccello, James P., Mark D. Chaffin, Elizabeth L. Chou, Stephen J. Fleming, Honghuang Lin, Mahan Nekoui, Shaan Khurshid, et al. 2022. "Deep Learning Enables Genetic Analysis of the Human Thoracic Aorta." *Nature Genetics* 54 (1): 40–51.
- Pirruccello, James P., Paolo Di Achille, Victor Nauffal, Mahan Nekoui, Samuel F. Friedman, Marcus D. R. Klarqvist, Mark D. Chaffin, et al. 2022. "Genetic Analysis of Right Heart Structure and Function in 40,000 People." *Nature Genetics* 54 (6): 792–803.
- Pirruccello, James P., Joel T. Rämö, Seung Hoan Choi, Mark D. Chaffin, Shinwan Kany, Mahan Nekoui, Elizabeth L. Chou, et al. 2023. "The Genetic Determinants of Aortic Distention." *Journal of the American College of Cardiology* 81 (14): 1320–35.
- Raisi-Estabragh, Zahra, Nicholas C. Harvey, Stefan Neubauer, and Steffen E. Petersen. 2021. "Cardiovascular Magnetic Resonance Imaging in the UK Biobank: A Major International Health Research Resource." *European Heart Journal Cardiovascular Imaging* 22 (3): 251–58.
- Rasouli, Seyed Javad, Mohamed El-Brolosy, Ayele Taddese Tsedeke, Anabela Bensimon-Brito, Parisa Ghanbari, Hans-Martin Maischein, Carsten Kuenne, and Didier Y. Stainier. 2018. "The Flow Responsive Transcription Factor Klf2 Is Required for Myocardial Wall Integrity by Modulating Fgf Signaling." *eLife* 7 (December). <https://doi.org/10.7554/eLife.38889>.
- Riba, Edgar, Dmytro Mishkin, Daniel Ponsa, Ethan Rublee, and Gary Bradski. 2019. "Kornia: An Open Source Differentiable Computer Vision Library for PyTorch." *arXiv [cs.CV]*. arXiv. http://openaccess.thecvf.com/content_WACV_2020/html/Riba_Kornia_an_Open_Source_Differentiable_Computer_Vision_Library_for_PyTorch_WACV_2020_paper.html.
- Rogaeva, Ekaterina, Yan Meng, Joseph H. Lee, Yongjun Gu, Toshitaka Kawarai, Fanggeng Zou, Taiichi Katayama, et al. 2007. "The Neuronal Sortilin-Related Receptor SORL1 Is Genetically Associated with Alzheimer Disease." *Nature Genetics* 39 (2): 168–77.
- Shinohara, Russell T., Elizabeth M. Sweeney, Jeff Goldsmith, Navid Shiee, Farrah J. Mateen, Peter A. Calabresi, Samson Jarso, et al. 2014. "Statistical Normalization Techniques for Magnetic Resonance Imaging." *NeuroImage. Clinical* 6 (August): 9–19.
- Siu, Samuel C., and Candice K. Silversides. 2010. "Bicuspid Aortic Valve Disease." *Journal of the American College of Cardiology* 55 (25): 2789–2800.
- Skrbic, Biljana, Kristin V. T. Engebretsen, Mari E. Strand, Ida G. Lunde, Kate M. Herum, Henriette S. Marstein, Ivar Sjaastad, et al. 2015. "Lack of Collagen VIII Reduces Fibrosis and Promotes Early Mortality and Cardiac Dilatation in Pressure Overload in Mice." *Cardiovascular Research* 106 (1): 32–42.
- Small, Aeron M., Gina Peloso, Jason Linefsky, Jayashri Aragam, Ashley Galloway, Vidisha Tanukonda, Lu-Chen Wang, et al. 2023. "Multiancestry Genome-Wide Association Study of Aortic Stenosis Identifies Multiple Novel Loci in the Million Veteran Program." *Circulation*, February. <https://doi.org/10.1161/CIRCULATIONAHA.122.061451>.
- Smith, Leslie N., and Nicholay Topin. 2019. "Super-Convergence: Very Fast Training of Neural Networks Using Large Learning Rates." In *Artificial Intelligence and Machine Learning for Multi-Domain Operations Applications*, 11006:369–86. SPIE.
- Sudlow, Cathie, John Gallacher, Naomi Allen, Valerie Beral, Paul Burton, John Danesh, Paul Downey, et al. 2015. "UK Biobank: An Open Access Resource for Identifying the Causes of a Wide Range of Complex Diseases of Middle and Old Age." *PLoS Medicine* 12 (3): e1001779.
- Tadros, Rafik, Catherine Francis, Xiao Xu, Alexa M. C. Vermeer, Andrew R. Harper, Roy Huurman, Ken Kelu Bisabu, et al. 2021. "Shared Genetic Pathways Contribute to Risk of Hypertrophic and Dilated Cardiomyopathies with Opposite Directions of Effect." *Nature Genetics* 53 (2): 128–34.
- Takahashi, Mao, Hideaki Bujo, Meizi Jiang, Hirofumi Noike, Yasushi Saito, and Kohji Shirai. 2010. "Enhanced Circulating Soluble LR11 in Patients with Coronary Organic Stenosis."

Atherosclerosis 210 (2): 581–84.

- Thériault, Sébastien, Zhonglin Li, Erik Abner, Jian 'an Luan, Hasanga D. Manikpurage, Ursula Houessou, Pardis Zamani, et al. 2024. "Integrative Genomic Analyses Identify Candidate Causal Genes for Calcific Aortic Valve Stenosis Involving Tissue-Specific Regulation." *Nature Communications* 15 (1): 2407.
- Turley, Patrick, Raymond K. Walters, Omeed Maghjian, Aysu Okbay, James J. Lee, Mark Alan Fontana, Tuan Anh Nguyen-Viet, et al. 2018. "Multi-Trait Analysis of Genome-Wide Association Summary Statistics Using MTAG." *Nature Genetics* 50 (2): 229–37.
- Vahanian, Alec, Friedhelm Beyersdorf, Fabien Praz, Milan Milojevic, Stephan Baldus, Johann Bauersachs, Davide Capodanno, et al. 2022. "2021 ESC/EACTS Guidelines for the Management of Valvular Heart Disease: Developed by the Task Force for the Management of Valvular Heart Disease of the European Society of Cardiology (ESC) and the European Association for Cardio-Thoracic Surgery (EACTS)." *European Heart Journal* 43 (7): 561–632.
- Vijayaraj, Preethi, Alexandra Le Bras, Nora Mitchell, Maiko Kondo, Saul Juliao, Meredith Wasserman, David Beeler, et al. 2012. "Erg Is a Crucial Regulator of Endocardial-Mesenchymal Transformation during Cardiac Valve Morphogenesis." *Development* 139 (21): 3973–85.
- Wei, Wei-Qi, Xiaohui Li, Qiping Feng, Michiaki Kubo, Iftikhar J. Kullo, Peggy L. Peissig, Elizabeth W. Karlson, et al. 2018. "LPA Variants Are Associated With Residual Cardiovascular Risk in Patients Receiving Statins." *Circulation* 138 (17): 1839–49.
- Willer, Cristen J., Yun Li, and Gonçalo R. Abecasis. 2010. "METAL: Fast and Efficient Meta-Analysis of Genomewide Association Scans." *Bioinformatics* 26 (17): 2190–91.
- Yan, Andrew T., Maria Koh, Kelvin K. Chan, Helen Guo, David A. Alter, Peter C. Austin, Jack V. Tu, Harindra C. Wijeyesundera, and Dennis T. Ko. 2017. "Association Between Cardiovascular Risk Factors and Aortic Stenosis: The CANHEART Aortic Stenosis Study." *Journal of the American College of Cardiology* 69 (12): 1523–32.
- Yang, Bo, Wei Zhou, Jiao Jiao, Jonas B. Nielsen, Michael R. Mathis, Mahyar Heydarpour, Guillaume Lettre, et al. 2017. "Protein-Altering and Regulatory Genetic Variants near GATA4 Implicated in Bicuspid Aortic Valve." *Nature Communications* 8 (May): 15481.
- Yap, Sing-Chien, Robert-Jan van Geuns, Folkert J. Meijboom, Sharon W. Kirschbaum, Jackie S. McGhie, Maarten L. Simoons, Philip J. Kilner, and Jolien W. Roos-Hesselink. 2007. "A Simplified Continuity Equation Approach to the Quantification of Stenotic Bicuspid Aortic Valves Using Velocity-Encoded Cardiovascular Magnetic Resonance." *Journal of Cardiovascular Magnetic Resonance: Official Journal of the Society for Cardiovascular Magnetic Resonance* 9 (6): 899–906.
- Yu Chen, Hao, Christian Dina, Aeron M. Small, Christian M. Shaffer, Rebecca T. Levinson, Anna Helgadóttir, Romain Capoulade, et al. 2023. "Dyslipidemia, Inflammation, Calcification, and Adiposity in Aortic Stenosis: A Genome-Wide Study." *European Heart Journal* 44 (21): 1927–39.
- Yu, Wei, Xueyan Ma, Jinjin Xu, Andreas Wilhelm Heumüller, Zhaoliang Fei, Xue Feng, Xiaodong Wang, et al. 2019. "VGLL4 Plays a Critical Role in Heart Valve Development and Homeostasis." *PLoS Genetics* 15 (2): e1007977.
- Zakhary, D. R., C. S. Moravec, and M. Bond. 2000. "Regulation of PKA Binding to AKAPs in the Heart: Alterations in Human Heart Failure." *Circulation* 101 (12): 1459–64.
- Zhu, Wuqiang, Meng Zhao, Saidulu Mattapally, Sifeng Chen, and Jianyi Zhang. 2018. "CCND2 Overexpression Enhances the Regenerative Potency of Human Induced Pluripotent Stem Cell-Derived Cardiomyocytes." *Circulation Research* 122 (1): 88–96.

Response to Reviewers

Reviewer #1:

Remarks to the Author:

The authors have properly addressed my major concerns, and the manuscript now represents a major contribution to the field of genetics of aortic valve function and disease.

Author response: We appreciate the recognition of our efforts and the kind words. The helpful Reviewer comments significantly strengthened the manuscript.

The following remaining comments are minor:

1) Title: The genetic determinants of aortic valve function link 166 loci to aortic stenosis risk

The authors are correct that the MTAG identifies 166 loci associated with either AV function and/or AS. However, the title claiming that 166 loci are linked to AS risk is not supported by the presented data. If the authors want to keep this title, they need to provide evidence that all 166 loci are linked to AS risk. At minimum, lead variants at all 166 loci would need to be significantly associated with AS in the AS GWAS METAL meta-analysis (not MTAG) using a bonferroni corrected threshold ($P < 0.05/166$) or similar correction for multiple "targeted" testing.

Author response: We agree with the Reviewer. Our original title unhelpfully conflated the quantitative trait and AS locus counts. We have now followed the Editor's suggestion to re-title the manuscript to: "**Multi-trait analyses identify genetic variants associated with aortic valve function and aortic stenosis risk**"

2) Line 203: "...remaining loci had still had...". Please rephrase.

Author response: Thank you for catching this oversight. This has now been corrected in the new version of the sentence. We have noticed that the Figure itself was misleading in the presentation and have therefore reformatted it as outlined below. We also corrected the number of significant loci overlapping between aortic valve traits and aortic stenosis after MTAG.

Changes to the manuscript:

The **Results>Multitrait analysis of GWAS identifies 166 loci for aortic valve function and aortic stenosis** section now states:

Of the 166 distinct MTAG loci, 102 were associated with AS and at least one valve measurement with $P < 5 \times 10^{-08}$ for both (**Supplementary Figure 3**, raw non-MTAG P values and effect estimates shown for all MTAG loci in **Figure 4**).

3) Line 227: "All five polygenic scores were significantly associated ...". It is unclear what are these 5 scores, and Table S5 only lists 4 scores.

Author response: This is a good catch. We have corrected this mistake.

Changes to the manuscript:

The **Results>Polygenic scores predict aortic stenosis in All of Us and Mass General Brigham Biobank** section now states:

All four polygenic scores were significantly associated with incident AS among 496 incident cases and 243,954 controls.

4) It is unclear why there are supplemental tables in two different sequences (A-M, and 1-6). This is a little confusing. For each supplemental table, please provide a list of abbreviations used for clarity.

Author response: Thanks, we have now numbered all supplemental tables in one sequence. We also provide a list of abbreviations in either the table legend in the supplemental file or the manifest sheet in the excel file.

5) Figure 4, specify if the beta are those of the MTAG or GWAS. If beta differs between GWAS and MTAG, it is preferable to report that of the GWAS as these are less biased by the other correlated traits.

Author response: We have now modified Figure 4 to use both the raw P-values and the raw GWAS (or METAL) effect sizes, instead of the post-MTAG values. The site selection is unchanged, and is still based on the final list of MTAG loci. We believe this formulation is of greater interest to the reader. For example, it allows the reader to identify the loci that did not meet $P < 5E-08$ for any of the raw GWASes, but which appear on this plot because at least one trait did so after MTAG. We have retained the version using MTAG effect sizes in the supplementary material since we believe these effect sizes enable complementary information for interested readers.

Changes to the manuscript:

Figure 4 is now

Depiction of the loci significant for at least one trait in MTAG analysis. Effect estimates and P-values are taken from the original (non-MTAG) analysis. Effect direction is depicted with respect to the reference allele. Variants with a risk-decreasing reference allele for aortic stenosis are depicted in the left panel; those with a risk-increasing reference allele are depicted in the right panel. Within each panel, variants are sorted by effect size for aortic stenosis. Effect sizes have

units of standard deviation for AVA, peak velocity, mean gradient, and log odds for aortic stenosis. Variants are represented by nearest gene name and variant identifier; multiple variants at one locus are listed when the lead variant differs across phenotypes.

6) Figure 5, provide the numbers at risk below the KM curves (at time 0 and each year of follow-up). I also suggest to add the results from MGB as a second panel (using the same PRS). Provide the HR with 95%CI on the figure and/or legend.

Author response: Thanks for the suggestions, Figure 5 is now updated to include the numbers at risk. We have also added the results from MGB as a second panel using the same PRS.

Changes to the manuscript:

Figure 5 is now

Kaplan-Meier cumulative incidence of aortic stenosis in 496 cases and 243,954 controls and 95% - confidence interval in *All of Us* (496 cases, 243,954 controls, left panel) and Mass General Brigham (MGB) Biobank (680 cases, 42,328 controls, right panel). Red: the top 5% for the aortic stenosis polygenic score. Blue: all other participants. The plot is terminated at the 95th percentile of follow-up time (5.0 years and 9.0 years, respectively). For *All of US*, the hazard ratio for the top 5% of participants was 3.32 ($P=8.8 \times 10^{-22}$) compared to the remaining 95% of participants. For MGB Biobank, the hazard ratio for the top 5% of participants was 2.76 ($P=7.8 \times 10^{-15}$) compared to the remaining 95% of participants.

7) Figure S4, Units (mmHg, cm²) are not needed since only P values are presented. If the authors decide to include the units, please check if the unit for peak velocity should be cm/s or m/s.

Author response: Thanks for the suggestions, Figure S4 has been adjusted to remove the units.

Changes to the manuscript:
Figure S4 is now

Statistical gene set prioritization for each aortic valve trait by applying the MAGMA framework to the MTAG-augmented GWAS summary statistics.

8) Lines 356-358 "In comparison, the best-performing disease-based PGS for AS from a recent study based on MVP showed a ~1.4 increase in odds for AS." This sentence is confusing. The effect size in the Small et al refers to that per SD increase in PRS which is similar to what is reported by the authors using their PRS (1.6). The sentence placed here gives the impression that the "new" PRS developed by the authors performs better which is not supported by the presented data. Please remove sentence OR rephrase OR provide the performance of the Small et al PRS in AllOfUs to directly compare the "published" with the "new" PRS.

Author response: We thank the reviewer for highlighting this point and agree the original wording could cause confusion without further information. We have now expanded the section to reflect that the Small et al. PRS refers to an effect size per standard deviation. We would have welcomed the opportunity to directly compare the performance of the previously published PGS in our dataset, but unfortunately, this was not feasible due to the unavailability of the exact PGS weights. Apart from the differences in population (MVP being a mostly male cohort of older veterans vs the younger, more diverse cohort in AoU), the statistical models employed also do not permit a direct comparison. Small et al. used logistic regression for cases vs controls whereas we calculate hazard ratios in a Cox proportional hazard model. We believe this section is now clearer.

Changes to the manuscript:

The **Discussion** section now states:

The genetic contributions to normal variation in peak velocity and mean gradient were predictive of aortic stenosis diagnoses in three external cohorts: FinnGen (for the PGS from the aortic valve function GWASes before MTAG, since FinnGen data were used in the MTAG analyses) and MGB and *All of Us* (for the MTAG-augmented PGS). Additionally, in *All of Us*—a fully external biobank not used in the GWAS or the PGS derivation—each standard deviation increase in the MTAG-based aortic stenosis PGS conferred a hazard ratio of 1.64 for incident aortic stenosis. Compared to the remaining 95% of the population, the participants in the top 5% of the aortic stenosis PGS had a ~3.3-fold increased risk, which warrants further evaluation for clinical utility as part of a comprehensive screening algorithm in the future. The best-performing disease-based PGS for aortic stenosis from a recent study based on the Million Veteran Program showed an OR of 1.41 to 1.44 per standard deviation for aortic stenosis⁴⁹, but differences in study populations and analytic design preclude a direct comparison of performance.

Reviewer #2:

Remarks to the Author:

Thank you to the authors for their extensive revision of their manuscript. With the increase in sample size and application of MTAG, there has been a substantial uplift in

loci and performance of PRS and the reporting of new biological mechanisms. It is great the methodology is also now included in this manuscript. In response to reviewer 1 – many sensitivity analyses have been performed which is reassuring and supports the main observations.

Thank you for carefully addressing each of my comments and with the new analyses providing new figures and tables. Figure 1 is now very clear and the abstract very informative and clear.

Author response: Thank you very much for the kind words and the helpful comments during the review process. We believe that our manuscript is now much stronger thanks to suggestions from Reviewers and Editors.